# DreamCS: Geometry-Aware Text-to-3D Generation with Unpaired 3D Reward Supervision

**Xiandong Zou**[1], **Ruihao Xia**[2], **Hongsong Wang**[3], **Pan Zhou**[1]*
[1]Singapore Management University [2]East China University of Science and Technology
[3]Southeast University

## Abstract

While text-to-3D generation has attracted growing interest, existing methods often struggle to produce 3D assets that align well with human preferences. Current preference alignment techniques for 3D content typically rely on hardly-collected preference-paired multi-view 2D images to train 2D reward models, when then guide 3D generation — leading to geometric artifacts, such as the Janus face problem and geometric incompleteness, due to their inherent 2D bias. To address these limitations, we construct 3D-MeshPref, the first large-scale unpaired 3D preference dataset, featuring diverse 3D meshes annotated by a large language model and refined by human evaluators. We then develop RewardCS, the first reward model trained directly on unpaired 3D-MeshPref data using a novel Cauchy-Schwarz divergence objective, enabling effective learning of human-aligned 3D geometric preferences without requiring paired comparisons. Building on this, we propose DreamCS, a unified framework that integrates RewardCS into text-to-3D pipelines — enhancing both implicit and explicit 3D generation with human preference feedback. Extensive experiments show DreamCS outperforms prior methods, producing 3D assets that are geometrically faithful and human-preferred.

## 1 Introduction

Text-to-3D generation has emerged as a key technique for automating 3D asset creation in domains such as gaming, film, digital comics, and virtual reality. Recent advances include one-stage optimization-based 2D lifting methods (Shi et al., 2023b; Wang et al., 2023b; Chen et al., 2023; Lin et al., 2023; Wang et al., 2023a; Liu et al., 2023a; Shi et al., 2023a; Qian et al., 2023; Wu et al., 2024c; Zhu et al., 2023b; Kwak et al., 2024), two-stage approaches that separately model geometry and appearance (Lin et al., 2023; Chen et al., 2023), and end-to-end methods (Tsalicoglou et al., 2024; Jun & Nichol, 2023). Despite these impressive advancements, generated 3D assets often misalign with human preferences, highlighting the need for preference-aware generation frameworks.

To address this gap, reinforcement learning from human feedback (RLHF) (Christiano et al., 2017; Yang et al., 2024) has recently been incorporated into text-to-3D pipelines (Xie et al., 2024; Ye et al., 2024; Zhou et al., 2025), showing improvements in fidelity and realism. However, current approaches face two key challenges. First, they rely on costly paired preference-labeled 2D data: for each prompt, multiple 3D assets must be rendered from diverse viewpoints, and preferred/dispreferred image pairs must be manually annotated. This process is computationally intensive, time-consuming, and difficult to scale, especially since large preference gaps are rare in practice. Second, these methods provide only 2D view-dependent supervision. Rewards derived from rendered images (e.g., ImageReward (Xu et al., 2023)) may favor assets that appear plausible from some viewpoints while ignoring structural flaws from others, leading to artifacts such as Janus faces or incomplete geometry (Fig. 1). The root cause lies in the absence of 3D-aware reward signals: existing diffusion-based 2D lifting pipelines excel at semantic alignment but lack explicit global geometric supervision, limiting their ability to ensure consistent and plausible 3D structure.

**Contributions.** In this work, we take a step forward by proposing a novel 3D-reward guided framework for text-to-3D generation. To eliminate paired training data and provide geometry-level feedback during generation, we respectively resolve three associated key challenges by proposing the

---

*Corresponding author.

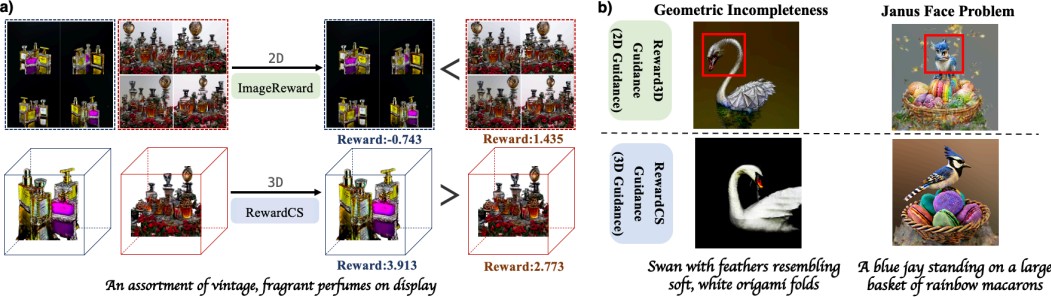

Figure 1: **Comparison of 2D- vs. 3D-based reward models. a)** 2D reward model ImageReward (Xu et al., 2023) assigns high scores to geometrically flawed 3D assets, while our 3D reward model RewardCS better aligns with human preference. **b)** while DreamFusion (Poole et al., 2022) guided by 2D Reward3D (Ye et al., 2024) produces 3D assets with geometric defects and the Janus problem, while DreamFusion with our RewardCS yields geometrically consistent 3D content.

first large-scale preference-unpaired 3D dataset, the first 3D reward model trained on our dataset with theoretical guarantees, and the first 3D-reward guided text-to-3D generative framework.

First, collecting paired preference-labeled 3D data is prohibitively expensive due to complex generation and annotation. To overcome this, we build **3D-MeshPref**, the first large-scale preference-unpaired 3D mesh dataset with over 30,000 samples, each containing a text prompt, 3D asset, and preference reward score. We curate meshes from Cap3D (Luo et al., 2023), evaluate them with Llama-Mesh (Wang et al., 2024) on geometric fidelity, semantic alignment, and structural plausibility, and refine scores with human verification. Assets with higher rewards are designated as preferred, while lower ones serve as dispreferred examples.

Second, prior preference alignment frameworks (Ouyang et al., 2022; Yang et al., 2024; Ye et al., 2024; Zhou et al., 2025) rely on paired comparisons and thus cannot operate in our unpaired settings. Accordingly, we introduce **RewardCS**, the first 3D geometry-aware reward model trained on unpaired data. To enable this, we propose a distribution-level training objective based on Cauchy–Schwarz (CS) divergence (Jenssen et al., 2006), treating preferred and dispreferred assets as samples from two distributions. Optimizing their CS divergence encourages higher rewards for geometrically and semantically superior assets. We also prove the equivalence between CS-divergence learning on unpaired data and vanilla paired preference supervision, providing theoretical guarantee.

Finally, existing text-to-3D frameworks lack native geometry-level feedback and face challenges in mesh representation, differentiability, and reward compatibility. So we develop **DreamCS**, the first 3D-reward guided text-to-3D framework. DreamCS integrates RewardCS into both implicit and explicit pipelines through three innovations: (1) differentiable meshization for end-to-end gradient flow, (2) adaptive mesh fusion for reward compatibility without sacrificing detail, and (3) progressive reward guidance balancing coarse structure refinement with fine-grained optimization.

Extensive experiments on the GPTEval3D benchmark (Wu et al., 2024a) show that DreamCS consistently improves geometric alignment and mesh quality for one-stage and two-stage pipelines like MVDream (Shi et al., 2023b), DreamFusion (Poole et al., 2022), and Magic3D (Lin et al., 2023). Moreover, our 3D reward guidance complements 2D-based methods, and can yield further gains.

## 2 RELATED WORK

**Text-to-3D Generation.** Recent advances in text-to-3D generation have largely been driven by the use of pretrained 2D diffusion models, given the scarcity and limited diversity of high-quality 3D datasets. A common strategy in works like DreamFusion (Poole et al., 2022) and SJC (Wang et al., 2023a) involves using Score Distillation Sampling (SDS) to optimize 3D representations like NeRF (Mildenhall et al., 2021)—by using gradients from a 2D diffusion model's denoising process as supervision to refine the 3D scene. Unfortunately, SDS suffers from issues like view inconsistency, the Janus problem, and poor structural fidelity (Wang et al., 2023b). To mitigate these, new methods adopt either one-stage pipelines (Shi et al., 2023b; Zhu et al., 2023b; Wang et al., 2023a; Wu et al., 2024c) that jointly optimize geometry and appearance, or two-stage pipelines (Lin et al., 2023; Chen et al., 2023; Wang et al., 2023b) that decouple them. However, most of them still struggle with alignment to human preferences, particularly in terms of global 3D geometry.

**Learning from Human Feedback.** Human feedback is increasingly vital for aligning generative models with user intent. In language, RLHF (Ouyang et al., 2022) and DPO (Rafailov et al., 2023) have successfully aligned LLMs to human values, and similar strategies have extended to images (Xu et al., 2023; Liang et al., 2024) and videos (Wu et al., 2024b; Liu et al., 2025a). In 3D generation, however, feedback has only recently been explored. DreamReward (Ye et al., 2024) trains a multi-view image-based reward model from annotated renderings, while DreamDPO (Zhou et al., 2025) applies DPO using human preferences from multi-view comparisons. Yet both approaches remain constrained to 2D supervision, aligning view-specific appearance rather than global 3D structure and often yielding inconsistent geometry. To address this gap, we introduce RewardCS, a 3D reward model for direct geometric supervision, and further design DreamCS to integrate this feedback into optimization, ensuring more consistent, plausible, and human-aligned 3D content.

## 3 METHODOLOGY

We first construct a large-scale preference-unpaired 3D mesh dataset, **3D-MeshPref**, in Section 3.1. Building on this, we develop the first 3D-geometry-aware reward model, **RewardCS**, trained on the 3D-MeshPref dataset, detailed in Section 3.2. To effectively integrate the 3D reward model, RewardCS, into text-to-3D generation pipelines, we propose **DreamCS** in Section 3.3 — a framework that aligns generated 3D content with human preferences through geometry-aware 3D supervision.

### 3.1 3D-MESHPREF: LARGE-SCALE PREFERENCE-UNPAIRED 3D MESH DATASET

A major challenge in 3D preference alignment is the lack of labeled datasets, as generating and annotating high-quality 3D assets is resource-intensive. Moreover, the complexity of explicit 3D representations hinders the construction of consistent paired preference data, limiting the development of effective 3D reward models. To overcome this, we propose **3D-MeshPref**, the first large-scale preference-unpaired 3D mesh dataset. It includes 30,000+ samples, each with a text prompt, 3D asset, and a preference reward score. Our pipeline combines automated LLM-based scoring with human refinement, enabling scalable annotation while preserving alignment with human judgment.

We build our dataset from annotated point clouds in Cap3D (Luo et al., 2023), sourced from 3D datasets like Objaverse (Deitke et al., 2023) and ABO (Collins et al., 2022). ABO contains high-resolution 3D household objects across 63 categories; Objaverse offers a large-scale and semantically rich 3D collection of over 21,000 categories. For diversity, we respectively collect 8,000 and 15,000 samples from ABO and Objaverse, which are then converted into high-quality mesh representations using MeshAnythingV2 (Chen et al., 2024), which consistently outperforms other meshilization methods in both mesh quality and computational efficiency. To ensure mesh quality and

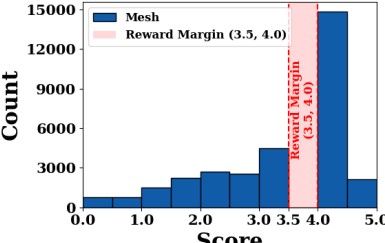

Figure 2: Annotated score distribution of meshes in **3D-MeshPref**.

compatibility with our 3D reward model, we apply QEM mesh simplification algorithm (Wu et al., 2004) to ensure the output with a maximum of 16,384 triangular faces. After filtering for mesh quality and completeness, we obtain 20,000+ 3D meshes. To adapt the 3D reward model to intermediate assets generated during SDS optimization, we augment the dataset with over 10,000+ meshes optimized using SDS-based methods, such as DreamFusion (Poole et al., 2022) and MVDream (Shi et al., 2023b), under text prompts in Objaverse. These meshes are sampled at early stages of the optimization process. (see Appendix B) This data augmentation exposes the reward model to both early- and late-stage geometries, improving robustness and generalization across the optimization.

We use Llama-Mesh (Wang et al., 2024), a capable open-source LLM fine-tuned for 3D evaluation, to score each mesh on a 0-5 Likert scale (Joshi et al., 2015) based on prompt alignment, structural realism, and visual fidelity. We observe that LLM-based ratings tend to overestimate quality relative to human perception. Accordingly, human raters verify and re-evaluate them. See annotation guidelines and human refinement process in Appendix E. We then partition the dataset using a reward threshold (3.5-4.0): meshes scoring $\geq 4.0$ show strong geometric integrity and semantic fidelity and are labeled as preferred (47% of all 3D asserts), while those $\leq 3.5$ typically indicate structural defects or poor alignment with the prompt and are dispreferred (53%). This clear margin excludes ambiguous cases, ensuring reliable partitioning—a distribution visualized in Fig. 2. This separation provides a strong supervisory signal for learning to distinguish relative quality in 3D meshes, and also provides the training of RewardCS balanced preference pairs—that is, each training instance

Figure 3: RewardCS is trained on 3D-MeshPref using Cauchy-Schwarz objective.

contains one preferred and one dispreferred mesh sample, regardless of their raw score values, ensuring that the learning signal remains unbiased toward either class and avoids reward skewing due to class imbalance. We provide examples and detailed rationale for 3D-MeshPref in Appendix B.

## 3.2 REWARDCS: 3D GEOMETRY-AWARE REWARD MODEL

Learning reward models for 3D assets is crucial for guiding high-quality text-to-3D generation (Xie et al., 2024; Ye et al., 2024), but it is hindered by the scarcity of paired preference annotations, which are costly and hard to scale due to 3D complexity. Prior preference alignment methods — both general (e.g., DPO (Rafailov et al., 2023) and its variants (Meng et al., 2024; Lai et al., 2024)) and 3D-specific (Zhou et al., 2025; Ye et al., 2024) — rely on paired data of the form $\{\mathbf{m}_i^+, \mathbf{m}_i^-, \mathbf{c}_i\}_{i=1}^m$, where each tuple includes a prompt $\mathbf{c}_i$ drawn from prompt distribution $p_{\mathbf{C}}$, and a pair of 3D meshes: a preferred one $\mathbf{m}_i^+$ and a dispreferred one $\mathbf{m}_i^-$. These samples are drawn from prompt-dependent distributions: $\mathbf{m}_i^+ \sim p^+(\cdot|\mathbf{c}_i)$ and $\mathbf{m}_i^- \sim p^-(\cdot|\mathbf{c}_i)$. The goal is to train reward models to assign higher rewards to preferred assets and guide text-to-3D generative models. However, this paradigm fails when such paired data are unavailable, which is common in real-world 3D settings.

### 3.2.1 REWARDCS: A NOVEL FRAMEWORK FOR UNPAIRED 3D PREFERENCE LEARNING

To overcome this limitation, we introduce **RewardCS**, the first geometry-aware 3D reward model that learns from unpaired preference data. Rather than relying on explicit preferred–dispreferred pairs, RewardCS leverages a distributional training objective based on Cauchy-Schwarz (CS) divergence (Jenssen et al., 2006), enabling effective preference modeling at scale. Our method is built upon the newly introduced 3D-MeshPref dataset, which contains only unpaired samples: high-quality (preferred) and low-quality (dispreferred) 3D meshes associated with diverse prompts.

Let $\{\mathbf{n}_i^+\}_{i=1}^m \sim p^+(\cdot|c_i)$ and $\{\mathbf{n}_j^-\}_{j=1}^n \sim p^-(\cdot|c_j)$ denote two sets of unpaired preferred and dispreferred 3D meshes, each conditioned on independently sampled prompts $\{c_i\}_{i=1}^m$ and $\{c_j\}_{j=1}^n$ from a distribution $p_{\mathbf{C}}$. Standard preference alignment methods (Ouyang et al., 2022; Christiano et al., 2017; Ye et al., 2024) are inapplicable in this unpaired setting because they require joint comparisons across samples from the same prompt. Instead, we formulate preference learning as a distribution matching problem. We define a 3D reward model as a function $r_{\boldsymbol{\theta}} : \mathcal{M} \mid \mathcal{C} \to \mathbb{R}$, parameterized by $\boldsymbol{\theta}$, which maps a 3D mesh $\mathbf{m} \in \mathcal{M}$ and the text condition $\mathbf{c} \in \mathcal{C}$ to a scalar reward. Our reward model, RewardCS, consists of an encoder $r_{\boldsymbol{\theta}_e}$ that transforms the input pair into a unified embedding $\mathbf{z} \in \mathcal{Z}$ and a projection head to generate a scalar reward. Crucially, the model is designed to learn the reward conditioned on the text prompt (see Section 3.2.2). To train the 3D reward model, we encode unpaired preferred and dispreferred samples into latent embeddings via the encoder $r_{\boldsymbol{\theta}_e}$:

$$\{\mathbf{x}_i\}_{i=1}^m = r_{\boldsymbol{\theta}_e}(\mathbf{n}_i^+, \mathbf{c}_i) \sim p(\mathbf{x}), \quad \{\mathbf{y}_j\}_{j=1}^n = r_{\boldsymbol{\theta}_e}(\mathbf{n}_j^-, \mathbf{c}_j) \sim p(\mathbf{y}). \quad (1)$$

We treat these embeddings as samples from two distinct distributions $p(\mathbf{x})$ and $p(\mathbf{y})$, representing the overall structure of high- and low-quality 3D meshes in the latent space. Then rather than comparing individual mesh pairs, we aim to separate the distributions $p(\mathbf{x})$ and $p(\mathbf{y})$ in the embedding space. It allows the reward model to generalize over the global statistical properties of high- and low-quality 3D assets, enabling robust learning even when direct pairwise supervision is unavailable. To this end, we introduce a distribution-level loss based on CS divergence (Yin et al., 2025), which quantifies the dissimilarity between two probability density functions $p$ and $q$ over the support $\omega$:

$$D_{CS}(p \| q) = -\log\left(\left(\int p(\omega)q(\omega)\,d\omega\right)^2 \Big/ \left(\int p(\omega)^2\,d\omega \int q(\omega)^2\,d\omega\right)\right). \quad (2)$$

Compared to the Kullback–Leibler divergence (Van Erven & Harremos, 2014), the CS divergence offers a tighter generalization bound (Yin et al., 2024) and is more robust than Jensen-Shannon divergence, which lacks a closed-form for Gaussians (Fuglede & Topsoe, 2004; Nielsen, 2019).

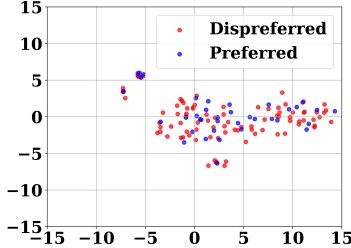

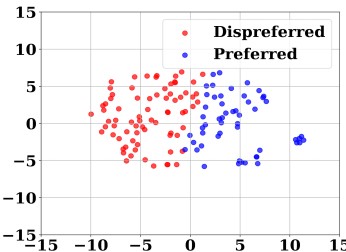

(a) RewardCS trained using Eq. (4) with $\lambda = 0$.  (b) RewardCS trained using Eq. (4) with $\lambda = 1$.

Figure 4: **t-SNE visualizations of class token embeddings based on different RewardCS in 3D-MeshPref data.** The addition of $\mathcal{L}_{\text{div}}$ enables RewardCS to separate the underlying data distributions of preferred and dispreferred mesh samples ($p(\mathbf{x})$ and $p(\mathbf{y})$) in the learned embedding space.

This makes CS divergence well-suited for unpaired 3D reward learning, as maximizing it helps the model capture semantic and geometric cues that differentiate preferred from dispreferred assets without explicit supervision for each prompt.

**Estimation of CS-Divergence.** Since true data distribution densities $p(\mathbf{x})$ and $p(\mathbf{y})$ are unknown, we estimate them via kernel density estimation (KDE) (Weglarczyk, 2018). Let $\{\mathbf{x}_i\}_{i=1}^m$ and $\{\mathbf{y}_j\}_{j=1}^n$ denote the embeddings of preferred and dispreferred samples. The empirical CS divergence is:

$$\hat{D}_{CS}(p(\mathbf{x}) \,\|\, p(\mathbf{y})) = \log\left(\frac{1}{m^2} \sum_{i,j=1}^m \kappa(\mathbf{x}_i, \mathbf{x}_j)\right) + \log\left(\frac{1}{m^2} \sum_{i,j=1}^n \kappa(\mathbf{y}_i, \mathbf{y}_j)\right) - 2\log\left(\frac{1}{mn} \sum_{i=1}^m \sum_{j=1}^n \kappa(\mathbf{x}_i, \mathbf{y}_j)\right), \quad (3)$$

where $\kappa(\cdot, \cdot)$ is a kernel function. This estimator possesses desirable properties: it is symmetric and differentiable. Importantly, it supports unequal numbers of preferred and dispreferred unpaired samples ($m \neq n$), enabling distributional alignment without requiring pairwise annotations.

**Overall Training Objective of RewardCS.** To train our RewardCS model $r_{\boldsymbol{\theta}}$, we combine a regression loss with the CS divergence loss. The final training objective is given by:

$$\mathcal{L}_{\text{RewardCS}}(\boldsymbol{\theta}) = \mathcal{L}_{\text{MSE}}(\boldsymbol{\theta}) + \lambda \mathcal{L}_{\text{div}}(\boldsymbol{\theta}), \quad (4)$$

where $\mathcal{L}_{\text{MSE}}$ is the mean squared error for reward prediction, $\mathcal{L}_{\text{div}} = -\hat{D}_{CS}(p(\mathbf{x}); p(\mathbf{y}))$ encourages separation between preferred and dispreferred mesh embeddings, and $\lambda$ is a hyperparameter.

Fig. 4 shows 2D t-SNE visualizations of mesh embeddings from RewardCS trained with $\lambda = 0$ and $\lambda = 1$. Adding $\mathcal{L}_{\text{div}}$ helps the reward model distinguishing preferred and dispreferred meshes in the embedding space, and enables RewardCS to align with human preferences from unpaired data. To further validate the CS divergence objective, we conduct an ablation study to examine its impact on clustering embedding quality and downstream performance (see Appendix C.2).

**Theoretical Justification.** We analyze CS divergence for training the 3D reward model on unpaired preference data by demonstrating its asymptotic equivalence to paired preference supervision. Optimizing the CS divergence is equivalent to optimizing a quantity in a kernel feature space (Jenssen et al., 2006; Yin et al., 2025):

$$\hat{D}_{\text{CS}}(p(\mathbf{x}) \,\|\, p(\mathbf{y})) = -2\log \frac{\langle \boldsymbol{\mu}_x, \boldsymbol{\mu}_y \rangle_{\mathcal{H}}}{\|\boldsymbol{\mu}_x\|_{\mathcal{H}} \|\boldsymbol{\mu}_y\|_{\mathcal{H}}}, \quad \text{with} \quad \boldsymbol{\mu}_x = \frac{1}{m} \sum_{i=1}^m \varphi(\mathbf{x}_i), \quad \boldsymbol{\mu}_y = \frac{1}{n} \sum_{i=1}^n \varphi(\mathbf{y}_i), \quad (5)$$

where a characteristic kernel $\kappa(\mathbf{x}, \mathbf{y})$ is defined as $\kappa(\mathbf{x}, \mathbf{y}) = \langle \varphi(\mathbf{x}), \varphi(\mathbf{y}) \rangle_{\mathcal{H}}$, and $\varphi$ maps samples to a Reproducing Kernel Hilbert Space (RKHS) $\mathcal{H}$ (Smola et al., 2007). In our case, we need to compute the means as

$$\begin{aligned}\boldsymbol{\mu}_x^{\text{paired}} &= \rho(\{(\mathbf{m}_i^+, \mathbf{c}_i)\}_{i=1}^m), \quad \boldsymbol{\mu}_y^{\text{paired}} = \rho(\{(\mathbf{m}_i^-, \mathbf{c}_i)\}_{i=1}^m), \\ \boldsymbol{\mu}_x^{\text{unpaired}} &= \rho(\{(\mathbf{n}_i^+, \mathbf{c}_i)\}_{i=1}^m), \quad \boldsymbol{\mu}_y^{\text{unpaired}} = \rho(\{(\mathbf{n}_j^-, \mathbf{c}_i)\}_{i=1}^n),\end{aligned} \quad (6)$$

where $\rho(\{(\mathbf{m}_i, \mathbf{c}_i)\}_{i=1}^k) = \frac{1}{k} \sum_{i=1}^k \varphi\left(r_{\boldsymbol{\theta}_e}(\mathbf{m}_i, \mathbf{c}_i)\right)$. Then we can define the CS divergence loss on unpaired data and paired data:

$$\hat{D}_{\text{CS}}^{\text{paired}} = -2\log \frac{\langle \boldsymbol{\mu}_x^{\text{paired}}, \boldsymbol{\mu}_y^{\text{paired}} \rangle_{\mathcal{H}}}{\|\boldsymbol{\mu}_x^{\text{paired}}\|_{\mathcal{H}} \|\boldsymbol{\mu}_y^{\text{paired}}\|_{\mathcal{H}}}, \quad \hat{D}_{\text{CS}}^{\text{unpaired}} = -2\log \frac{\langle \boldsymbol{\mu}_x^{\text{unpaired}}, \boldsymbol{\mu}_y^{\text{unpaired}} \rangle_{\mathcal{H}}}{\|\boldsymbol{\mu}_x^{\text{unpaired}}\|_{\mathcal{H}} \|\boldsymbol{\mu}_y^{\text{unpaired}}\|_{\mathcal{H}}}, \quad (7)$$

Then we pose necessary assumptions widely used in RLHF and divergence analysis (Zhu et al., 2023a; Zou et al., 2025; Kim et al., 2021).

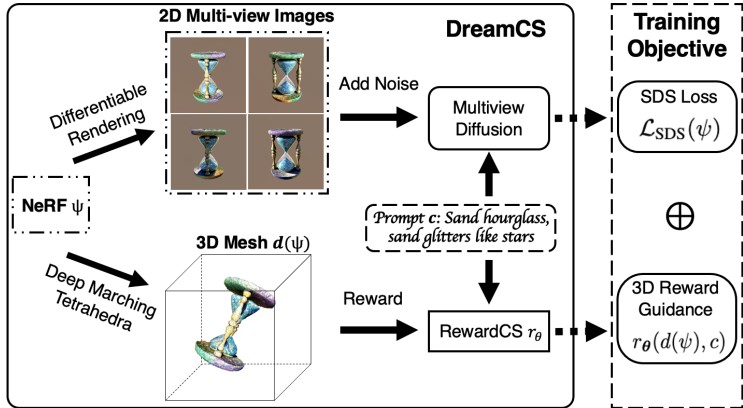

Figure 5: Framework of DreamCS: integrate RewardCS into the SDS model for NeRF optimization.

**Assumption 1.** *a) The kernel $\kappa(\cdot, \cdot)$ is bounded, i.e. $\sup_{\mathbf{x}, \mathbf{y} \in dom(\kappa)} \kappa(\mathbf{x}, \mathbf{y}) \leq K$, and characteristic, meaning the kernel mean embedding is injective. b) The population mean embeddings of the preferred and dispreferred samples satisfy:$\|\boldsymbol{\mu}_x^{\mathrm{paired}}\|_{\mathcal{H}}, \|\boldsymbol{\mu}_x^{\mathrm{unpaired}}\|_{\mathcal{H}}, \|\boldsymbol{\mu}_y^{\mathrm{paired}}\|_{\mathcal{H}}, \|\boldsymbol{\mu}_y^{\mathrm{unpaired}}\|_{\mathcal{H}} > 0$.*

Assumption 1 (a) and (b) are standard assumptions in kernel methods (Hofmann et al., 2008; Gretton et al., 2006; Muandet et al., 2017), and hold in our case. We use the Gaussian kernel $\kappa_\sigma(\mathbf{x}, \mathbf{y}) = \exp(-\|\mathbf{x} - \mathbf{y}\|_2^2/(2\sigma^2))$, which is bounded and characteristic, and our data distributions are well-separated, satisfying the non-zero mean embedding condition.

**Theorem 1** (Asymptotic Equivalence)**.** *Suppose Assumption 1 holds. With a constant $C > 0$, the empirical CS divergences $\hat{D}_{\mathrm{CS}}^{\mathrm{paired}}$ and $\hat{D}_{\mathrm{CS}}^{\mathrm{unpaired}}$ computed from paired and unpaired data satisfy:*

$$\left| \hat{D}_{\mathrm{CS}}^{\mathrm{paired}} - \hat{D}_{\mathrm{CS}}^{\mathrm{unpaired}} \right| \leq C \cdot \left( \frac{1}{\sqrt{m}} + \frac{1}{\sqrt{n}} \right) \xrightarrow{p} 0 \quad as \ m, n \to \infty, \tag{8}$$

See its proof in Appendix A. Thm.1 shows that the CS divergence computed from paired and unpaired preference data converges asymptotically at a convergence rate of $\mathcal{O}\left(\frac{1}{\sqrt{m}} + \frac{1}{\sqrt{n}}\right)$. Thus, our RewardCS can learn a latent embedding $\mathbf{z}$ such that maximizing CS divergence between embeddings of preferred and dispreferred samples induces reward separation even without paired data.

### 3.2.2 NETWORK ARCHITECTURE OF REWARDCS

As shown in Fig. 3, RewardCS contains two key components: an encoder $r_{\boldsymbol{\theta}_e}$ to produce a unified representations of 3D geometry and text, and a prediction head to generate a scalar reward score. The encoder $r_{\boldsymbol{\theta}_e}$ integrates three modules: a 3D mesh encoder for geometric feature extraction, a text encoder for prompt representation, and a cross-modal fusion module to integrate the two. 3D mesh encoder uses MeshMAE (Liang et al., 2022), a masked Vision Transformer (ViT) autoencoder for 3D meshes, which shows strong performance in capturing global geometric structure and its robustness to partial inputs and structural noise. Each input mesh is divided into 256 non-overlapping patches, with each patch consisting of 64 triangular faces. Each face is represented by a 10-dimensional feature vector capturing geometric attributes like area, angles, surface normals. This strategy enables 3D mesh encoder to capture both localized surface details and global geometric context. For text encoder, it adopts MeshCLIP (Song et al., 2023) to map input text prompt into a sequence of semantic embeddings. To fuse mesh and text information, we apply a cross-attention mechanism: mesh tokens with a learnable 128-dimensional class token act as queries, while text tokens serve as keys and values. During training, the pretrained text encoder is frozen. This design ensures that semantic information from the text condition is injected directly into the mesh representation, so that the resulting embeddings are contextually modulated by the prompt. The class token, which captures the global fused context, is then passed to an MLP-based projection head to produce the final reward score. More rationale of the network architecture is in Appendix C.1.

### 3.3 DREAMCS: 3D REWARD GUIDANCE FRAMEWORK

Integrating a 3D reward model into text-to-3D frameworks presents two key challenges. Firstly, existing reward methods are designed for 2D paired preferences and cannot directly handle preference-unpaired explicit 3D mesh representations, requiring a differentiable mechanism to convert implicit

fields into mesh form. Secondly, meshes generated during training may not meet the reward model's structural requirements due to face count constraints, necessitating adaptive topology alignment. To address these, we propose **DreamCS**, the first geometry-aware 3D reward guidance framework, integrating our RewardCS model into existing text-to-3D pipelines. As shown in Fig. 5, DreamCS enables end-to-end optimization via: (1) a differentiable meshization module for reward supervision, (2) an adaptive mesh fusion algorithm for topology alignment, and (3) a progressive reward guidance scheme balancing exploration and reward optimization.

**Differentiable Meshization.** Text-to-3D models often operate on implicit representations such as NeRF or SDFs, which must be converted into explicit mesh geometry to be compatible with our RewardCS model. We introduce a differentiable meshization module that enables smooth conversion from these fields to triangle meshes without breaking the optimization flow. For SDS pipelines (Poole et al., 2022), we use DMTet (Shen et al., 2021) to extract high-quality isosurfaces differentially. This preserves the gradient flow from the reward signal back to the implicit 3D parameters. Compared to Marching Cubes, DMTet enables high-fidelity geometry extraction. For pipelines that optimize explicit meshes, this conversion step is bypassed.

**Adaptive Mesh Fusion.** Meshes produced above are often structurally incompatible with the input requirements of our reward model, RewardCS which expects meshes partitioned into 256 non-overlapping patches, each with 64 faces. To address this, we propose an adaptive mesh fusion algorithm that simplifies and reorganizes mesh topology while preserving geometric fidelity.

The algorithm iteratively merges adjacent faces using two criteria: (1) similarity in face normals and (2) topological adjacency (shared vertex). When two faces share a vertex and exhibit high normal similarity, we construct a new face using the shared vertex and one additional vertex from each original face. When two faces share an edge and have highly aligned normals, a new face is formed using the edge and a third vertex from either face. This fusion process reduces mesh complexity while maintaining fine structural details. Importantly, the entire fusion algorithm is differentiable and integrated into the training loop, maintaining gradient flow and supporting end-to-end learning. We validate our proposed adaptive mesh fusion method on mesh geometric fidelity in Appendix D.2.

**Progressive Reward Guidance**. To ensure faithful 3D generation, DreamCS integrates RewardCS through progressive guidance. At each optimization step, the current differentiable mesh – produced via our differentiable meshization module – is evaluated by the RewardCS model $r_\theta$, which provides a gradient-based signal reflecting its geometric plausibility and alignment with input prompt.

Let the 3D generation be parameterized by an implicit representation $\psi_t$. At each optimization step $t$, the mesh $d(\psi_t)$ derived from $\psi_t$ via DMTet is evaluated by the reward model $r_\theta$ to assign a scalar score that reflects mesh's quality w.r.t. the input prompt $c$, which provides a gradient guidance signal based on the geometry of mesh vertices and flows back to the implicit field via mesh structure. Because RewardCS guidance derives from the high-level geometric properties of the mesh, it acts as a structural prior that promotes global geometric coherence and semantic alignment.

The optimization objective at optimization step $t$ is defined as:
$$\mathcal{L}(\psi_t) = \mathcal{L}_{\text{SDS}}(\psi_t) - \alpha(t) \cdot r_\theta(d(\psi_t)|c), \tag{9}$$
where $\mathcal{L}_{\text{SDS}}$ is vanilla score distillation loss (Poole et al., 2022), and $\alpha(t)$ is a weighting function.

To balance the SDS loss and the reward guidance, we adopt a progressive reward guidance schedule (Ye et al., 2024; Liu et al., 2025b). At the beginning of optimization, $\alpha$ is kept low so that the process is driven primarily by the SDS loss, enabling broad exploration of shape space. As optimization progresses, $\alpha$ increases linearly from $\alpha_{\min}$ to $\alpha_{\max}$:
$$\alpha(t) = \alpha_{\min} + (\alpha_{\max} - \alpha_{\min}) \cdot \tfrac{t}{T}. \tag{10}$$
This schedule delays strong reward influence until the geometry stabilizes, balancing the SDS loss and the reward guidance and promoting coarse-to-fine refinement.

## 4 EXPERIMENT

**Setup.** We test our DreamCS by integrating our reward model RewardCS into one-stage SDS-based text-to-3D generative models like MVDream (Shi et al., 2023b) and DreamFusion (Poole et al., 2022), and two-stage models like Magic3D (Lin et al., 2023) and Fantasia3D (Chen et al., 2023). Moreover, we compare with other 3D preference alignment approaches, DreamReward (Ye et al., 2024), and DreamDPO (Zhou et al., 2025). All baselines use their official implementations where

available, otherwise Threestudio's implementation (Liu et al., 2023b). Our DreamCS is initialized by pretrained ShapeNet (Chang et al., 2015) and fine-tuned on 3D-MeshPref by minimizing loss in Eq. (4). All methods run for 20,000 steps, with an additional 10,000 steps for refinement in texture/geometry-aware baselines. Rendering resolution follows a coarse-to-fine schedule ($64 \times 64$ for the first 5,000 steps, then $256 \times 256$), $\alpha_{\min} = 10$ and $\alpha_{\max} = 20$. We show that the trained MeshMAE and RewardCS can achieve strong alignment with the performance of LLaMA-Mesh, showing superiority in generalization and robustness (see details in Appendix C.3).

We use all 110 prompts in GPTEval3D (Wu et al., 2024a) for evaluation. We assess texture and geometry quality using CLIP (CP) (Radford et al., 2021) and VisionReward (VR) (Xu et al., 2024) on rendered 2D images. These image-text similarity metrics are computed by averaging the similarities between each view and the given prompt. We proposed a 3D Geometry-Asset Alignment Reward (GA) metric based on RewardCS trained with a different training configuration. It serves as an independent evaluator for 3D geometry-text alignment. MiniCPM-o (Yao et al., 2024), a multimodal vision-language model, rates text-asset alignment, 3D plausibility, and geometry-texture consistency on a 0–5 Likert scale. In addition, a user study with 30 participants on 60 prompts assesses these criteria. Finally, we report the proportion of 3D assets suffering from Janus artifacts (Proportion) by conducting an user study with 60 participants. See more details in Appendix E.

Table 1: Comparison of baselines using 110 GPTEval 3D prompts under **12 multi-view** images. Metrics: CP = CLIP, VR = VisionReward, GA = 3D Geometry-Asset Alignment Reward.

| Method | CP ↑ | VR ↑ | GA ↑ | Method | CP ↑ | VR ↑ | GA ↑ |
|---|---|---|---|---|---|---|---|
| DreamFusion | 0.22 | -3.21 | 2.53 | Magic3D Stage1 | 0.20 | -3.26 | 2.48 |
| +Reward3D | 0.23 | -3.11 | 2.77 | +Magic3D Stage2 | 0.21 | -3.03 | 2.65 |
| +DreamDPO | 0.23 | -2.98 | 2.79 | Magic3D Stage1+DreamDPO | 0.22 | -3.05 | 2.56 |
| +RewardCS | **0.25** | **-2.11** | 2.96 | +Magic3D Stage2+DreamDPO | 0.23 | -2.94 | 2.74 |
| +Reward3D+RewardCS | **0.25** | -2.77 | **3.22** | Magic3D Stage1+Reward3D | 0.23 | -3.09 | 2.63 |
| MVDream | 0.24 | -3.31 | 2.79 | +Magic3D Stage2+Reward3D | 0.23 | -3.32 | 2.55 |
| +Reward3D | 0.27 | -3.12 | 2.87 | Magic3D Stage1+RewardCS | 0.23 | -1.90 | 3.00 |
| +DreamDPO | 0.23 | -2.13 | 2.87 | +Magic3D Stage2+RewardCS | **0.24** | **-1.58** | **3.22** |
| +RewardCS | **0.29** | -2.11 | 2.96 | Fantasia3D | 0.22 | -3.03 | 2.95 |
| +Reward3D+RewardCS | **0.29** | **-1.92** | **3.08** | +RewardCS | **0.25** | **-1.01** | **3.33** |

**Quantitative Comparison.** Tab. 1 presents a comparative analysis of one-stage (DreamFusion, MVDream) and two-stage (Magic3D, Fantasia3D) pipelines, along with their respective 2D-guided (Reward3D, DreamDPO) and our proposed 3D-guided (RewardCS) variants. As Reward3D (Ye et al., 2024) is finetuned to optimize ImageReward score, methods with Reward3D guidance often overfit the ImageReward (IR) metric, but do not generate high-quality 3D assets as shown in Fig. 6.

Three key findings emerge from these results. **1)** RewardCS consistently improves upon all vanilla baselines and outperforms 2D-guided variants in CP, GA and VR scores. In Magic3D, RewardCS achieves the highest GA (3.22), significantly outperforming the baseline (2.65) and the variant with 2D guidance (2.55), while achieving the highest CP score (0.24) and VR score (-1.58). Similarly, in DreamFusion, RewardCS leads with a GA score of 2.96, surpassing both baseline (2.53) and the variant with 2D guidance (2.79), and the best CP (0.25) and VR score (-2.11). **2)** RewardCS exhibits strong compatibility with diverse text-to-3D backbones, including both implicit 3D representation optimization and explicit mesh-based rendering pipelines. In MVDream, it improves the vanilla baseline from -3.31 to -2.11 in VR score and from 2.79 to 2.96 in GA score, outperforming the Reward3D variant (-3.12 VR, 2.87 GA). Furthermore, in Fantasia3D's mesh-rendering-based pipeline, RewardCS achieves the highest CP and GA score and significantly improves the VR from -3.03 to -1.01. **3)** 3D and 2D guidance prove complementary: integrated guidance in DreamFusion achieves the peak GA (3.22) score while improving VR to -2.77, demonstrating that Reward3D enhances multi-view consistency while RewardCS optimizes geometric alignment. A discussion on inference computational cost is provided in Appendix F.4. Extended results on comparison with 3D controllable generation methods (Li et al., 2023; Huang et al., 2024) and advanced 3D generative backbones (Sun et al., 2023; Xiang et al., 2025) are provided in Appendix F.1 and F.2.

**Visualization Comparison.** Fig. 6 compares visual generation results across methods. Vanilla baselines such as frequently exhibit textural misalignment and the Janus face problem, as exemplified in Fantasia3D's example, where the baseline produces a two-headed geometry. Baselines augmented with 2D guidance via Reward3D or DreamDPO improve aesthetics but still display artifacts like ge-

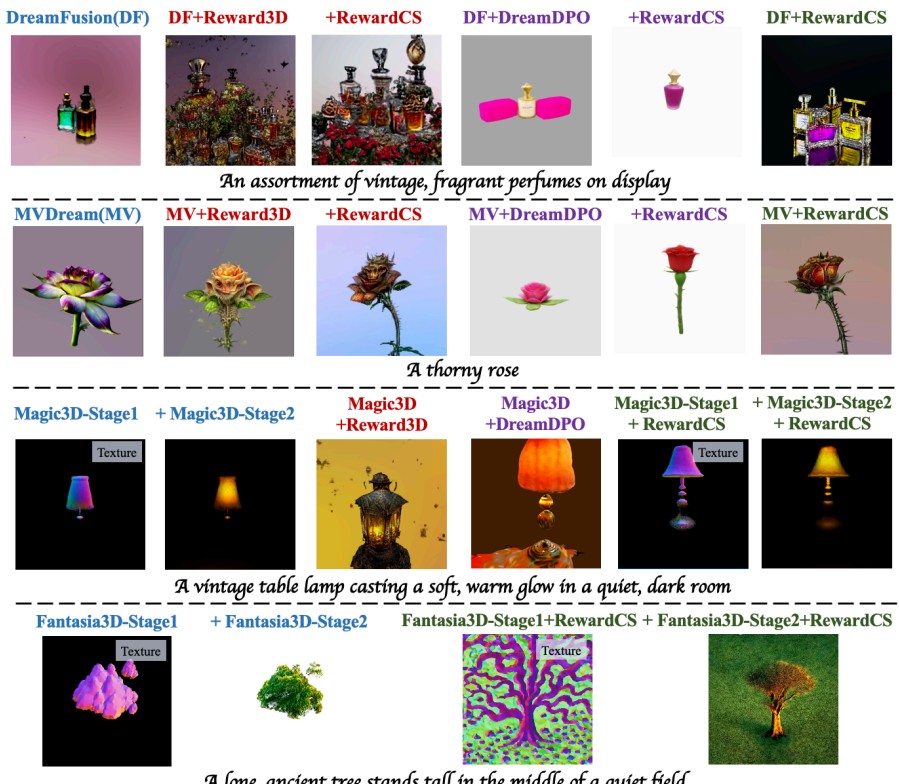

Figure 6: Comparisons with 1-stage generation pipelines (DreamFusion and MVDream) and 2-stage generation pipelines (Magic3D and Fantasia3D). More visualizations are provided in Appendix F.3.

Table 2: Comparison of methods on MiniCPM-o evaluation (left) and user study (right) using (a) 110 and (b) 30 representative GPTEval3D prompts, respectively. Metrics: T-A = Text-Asset Alignment, 3DP = 3D Plausibility, G-T = Geometry-Texture Alignment.

(a) MiniCPM-o evaluation on 110 prompts

| Method | T-A ↑ | 3DP ↑ | G-T ↑ |
|---|---|---|---|
| MVdream | 2.97 | 3.12 | 3.08 |
| +Reward3D | 3.38 | 3.35 | 3.22 |
| +DreamDPO | 3.42 | 3.30 | 3.11 |
| +RewardCS | 3.59 | **4.05** | **3.95** |
| +Reward3D+RewardCS | **3.61** | 3.98 | 3.81 |

(b) User study on 60 prompts

| Method | T-A ↑ | 3DP ↑ | G-T ↑ |
|---|---|---|---|
| MVdream | 2.90 | 2.87 | 2.91 |
| +Reward3D | 3.04 | 3.15 | 3.07 |
| +DreamDPO | 3.15 | 3.19 | 3.00 |
| +RewardCS | 3.21 | **3.72** | **3.59** |
| +Reward3D+RewardCS | **3.41** | 3.51 | 3.54 |

Table 3: Proportion of 3D assets suffering Janus artifacts using 60 GPTEval3D prompts.

(a) Evaluation on MVDream

| Method | Proportion ↓ |
|---|---|
| MVDream | 0.52 |
| +Reward3D | 0.44 |
| +DreamDPO | 0.43 |
| +RewardCS | 0.30 |
| +Reward3D+RewardCS | **0.28** |

(b) Evaluation on DreamFusion

| Method | Proportion ↓ |
|---|---|
| DreamFusion | 0.61 |
| +Reward3D | 0.53 |
| +DreamDPO | 0.50 |
| +RewardCS | 0.41 |
| +Reward3D+RewardCS | **0.39** |

ometric incompleteness and floaters, as seen in DreamFusion, Magic3D and MVDream's examples. In contrast, our DreamCS framework, which combines a baseline with RewardCS, demonstrates robust compatibility across all 3D generation backbones, consistently producing assets with superior text alignment, geometric completeness, and texture fidelity. Moreover, our RewardCS is also orthogonal to 2D guidance like Reward3D to further enhance text-to-3D generation.

**Evaluation via MiniCPM-o and Human.** Tab. 2 (a) reports evaluation results using MiniCPM-o, and Tab. 2 (b) reports findings from a user study. One can observe that RewardCS guidance outperforms 2D preference-based methods on geometry-related criteria while maintaining compet-

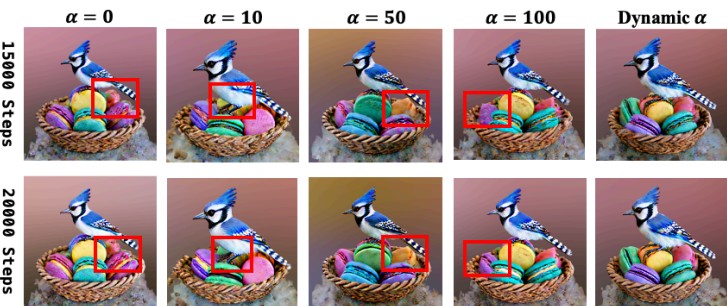

Figure 7: Ablation study on the weight of 3D guidance $\alpha$ in MVDream at 15000 and 20000 steps.

itive performance in text-asset alignment. In MiniCPM-o evaluations, MVDream with RewardCS achieves the highest scores in 3D Plausibility (4.05) and G-T Alignment (3.95), significantly exceeding both MVDream and MVDream with Reward3D. User studies confirm this trend, with RewardCS again leading in 3D Plausibility (3.72) and G-T Alignment (3.59).

**Evaluation via Janus-Artifact Proportion.** Regarding to the Janus problem, to the best of our knowledge, no established benchmark or quantitative metric currently exists for evaluating multi-view consistency or the Janus problem in text-to-3D generation. Consequently, we compare text-to-3D baselines and DreamCS using the proportion of generated 3D assets exhibiting Janus artifacts. As shown in Tab. 3, both MVDream and DreamFusion with RewardCS yields a lower rate of Janus artifacts (0.30 and 0.41) compared to both the vanilla baseline and 2D-guided variants.

**Ablation Study.** Fig. 7 shows the effect of varying guidance weight $\alpha$ in Eq. (9), balancing the 2D diffusion distillation score and DreamCS 3D guidance, on the quality of generated 3D assets using the MVDream backbone. Low values of $\alpha(0, 10)$ cause geometric ambiguity, while high $\alpha(50, 100)$ results in structural distortions. Our progressive strategy yields can improved results.

## 5 CONCLUSION

In this paper, we present DreamCS, the first 3D-guided text-to-3D generation framework explicitly designed for human preference alignment. Specifically, we construct a preference dataset on mesh data, 3D-MeshPref. Then we train a 3D reward model, RewardCS, on this dataset to encode human-aligned geometric priors. By incorporating RewardCS into the 3D generation pipeline, DreamCS produces geometrically plausible and texture-consistent 3D assets that are faithfully aligned with human preferences. Extensive experiments demonstrate that DreamCS consistently outperforms both vanilla and 2D-guided baselines across 3D geometry-oriented metrics and human evaluations.

**Limitations.** Due to limited budget, our evaluation of 3D generation relies on open-source LLMs like MiniCPM-o. Incorporating stronger models like GPT-4 (Wu et al., 2024a) may yield more reliable and fine-grained feedback, and will be considered when budget becomes available. In addition, the current absence of established benchmarks for assessing multi-view consistency and Janus artifacts limits the comprehensiveness of our evaluation. Developing standardized metrics related to the Janus problem would enable more robust comparisons in future work. Moreover, while DreamCS shows promising results, its performance is limited by text-to-3D generative backbone models like DreamFusion. In future work, we plan to explore advanced text-to-3D backbones.

## 6 DECLARATION OF LLM USAGE

We use Llama-Mesh to score each mesh in our 3D mesh dataset. Additionally, we employ MiniCPM-o to conduct the MLLM evaluation. All research ideas, methods, and experimental results presented are original contributions of the authors.

## ACKNOWLEDGEMENTS

This work was supported by the Singapore Ministry of Education (MOE) Academic Research Fund (AcRF) Tier 1 grant (Proposal ID: 23-SIS-SMU-070), the National Natural Science Foundation of China (62302093), and Jiangsu Province Natural Science Fund (BK20230833). Any opinions, findings and conclusions or recommendations expressed in this material are those of the author(s) and do not reflect the views of the Ministry of Education, Singapore.

## ETHICS STATEMENT

The text-to-3D base models employed in this research may reflect biases or generate sensitive or potentially offensive 3D content, intended solely for academic and scientific purposes. The opinions expressed within generated outputs do not represent the views of the authors. We remain committed to fostering the development of AI technologies which align with ethical standards and reflect societal values.

## REPRODUCIBILITY STATEMENT

We detail our work in the Methodology section (Section 3) and describe implementation details in Section 4 and Appendix E.

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

CONTENTS OF APPENDIX

## A  THEORETICAL ANALYSIS OF CS DIVERGENCE

**Theorem 2** (Asymptotic Equivalence). *Suppose Assumption 1 holds. With a constant $C > 0$, the empirical CS divergences $\hat{D}_{\text{CS}}^{\text{paired}}$ and $\hat{D}_{\text{CS}}^{\text{unpaired}}$ computed from paired and unpaired data satisfy:*

$$\left| \hat{D}_{\text{CS}}^{\text{paired}} - \hat{D}_{\text{CS}}^{\text{unpaired}} \right| \leq C \cdot \left( \frac{1}{\sqrt{m}} + \frac{1}{\sqrt{n}} \right) \xrightarrow{p} 0 \quad as \; m, n \to \infty, \tag{11}$$

Here we provide our proof for Theorem 1.

*Proof.* By Assumption 1(a), we know that the kernel $\kappa$ is bounded and we have $\sup_{\mathbf{x}} \kappa(\mathbf{x}, \mathbf{x}) \leq K < \infty$, so the canonical feature map $\varphi$ satisfies $\|\varphi(\mathbf{x})\|_{\mathcal{H}}^2 \leq K$ for all $\mathbf{x}$.

Next, we define the empirical mean embeddings for the paired and unpaired cases:

$$\boldsymbol{\mu}_x^{\text{paired}} = \frac{1}{m} \sum_{i=1}^{m} \varphi(r_{\boldsymbol{\theta}_{\text{e}}}(\mathbf{m}_i^+, \mathbf{c}_i)), \quad \boldsymbol{\mu}_y^{\text{paired}} = \frac{1}{m} \sum_{i=1}^{m} \varphi(r_{\boldsymbol{\theta}_{\text{e}}}(\mathbf{m}_i^-, \mathbf{c}_i)), \tag{12}$$

$$\boldsymbol{\mu}_x^{\text{unpaired}} = \frac{1}{m} \sum_{i=1}^{m} \varphi(r_{\boldsymbol{\theta}_{\text{e}}}(\mathbf{n}_i^+, \mathbf{c}_i)), \quad \boldsymbol{\mu}_y^{\text{unpaired}} = \frac{1}{n} \sum_{j=1}^{n} \varphi(r_{\boldsymbol{\theta}_{\text{e}}}(\mathbf{n}_j^-, \mathbf{c}_j)). \tag{13}$$

By the law of large numbers in Hilbert spaces, for i.i.d. samples, we obtain the weak convergence:

$$\frac{1}{m}\sum_{i=1}^{m}\varphi(r_{\boldsymbol{\theta}_e}(\mathbf{m}_i^+,\mathbf{c}_i)),\ \frac{1}{m}\sum_{i=1}^{m}\varphi(r_{\boldsymbol{\theta}_e}(\mathbf{n}_i^+,\mathbf{c}_i)) \xrightarrow{p} \mathbb{E}_{\mathbf{c}\sim p_{\mathbf{C}},\mathbf{m}^+\sim p^+(\cdot|\mathbf{c})}[\varphi(r_{\boldsymbol{\theta}_e}(\mathbf{m}^+,\mathbf{c}))], \tag{14}$$

$$\frac{1}{n}\sum_{j=1}^{n}\varphi(r_{\boldsymbol{\theta}_e}(\mathbf{m}_j^-,\mathbf{c}_j)),\ \frac{1}{n}\sum_{j=1}^{n}\varphi(r_{\boldsymbol{\theta}_e}(\mathbf{n}_j^-,\mathbf{c}_j)) \xrightarrow{p} \mathbb{E}_{\mathbf{c}\sim p_{\mathbf{C}},\mathbf{m}^-\sim p^-(\cdot|\mathbf{c})}[\varphi(r_{\boldsymbol{\theta}_e}(\mathbf{m}^-,\mathbf{c}))]. \tag{15}$$

Thus, we have the convergence of the empirical mean embeddings to their respective population mean embeddings in the RKHS:

$$\boldsymbol{\mu}_x^{\text{paired}},\ \boldsymbol{\mu}_x^{\text{unpaired}} \xrightarrow{p} \boldsymbol{\mu}_x^*,\quad \boldsymbol{\mu}_y^{\text{paired}},\ \boldsymbol{\mu}_y^{\text{unpaired}} \xrightarrow{p} \boldsymbol{\mu}_y^*, \tag{16}$$

where the population mean embeddings are defined as:

$$\boldsymbol{\mu}_x^* = \mathbb{E}_{\mathbf{c}\sim p_{\mathbf{C}},\,\mathbf{m}^+\sim p^+(\cdot|\mathbf{c})}[\varphi(r_e(\mathbf{m}^+,\mathbf{c}))],\quad \boldsymbol{\mu}_y^* = \mathbb{E}_{\mathbf{c}\sim p_{\mathbf{C}},\,\mathbf{m}^-\sim p^-(\cdot|\mathbf{c})}[\varphi(r_e(\mathbf{m}^-,\mathbf{c}))] \tag{17}$$

Applying the triangle inequality for the distance between the paired and unpaired empirical mean embeddings, we obtain:

$$\left\|\boldsymbol{\mu}_x^{\text{paired}}-\boldsymbol{\mu}_x^{\text{unpaired}}\right\|_{\mathcal{H}} \le \left\|\boldsymbol{\mu}_x^{\text{paired}}-\boldsymbol{\mu}_x\right\|_{\mathcal{H}} + \left\|\boldsymbol{\mu}_x^{\text{unpaired}}-\boldsymbol{\mu}_x\right\|_{\mathcal{H}} \xrightarrow{p} 0 \tag{18}$$

and similarly for $\boldsymbol{\mu}_y$.

Therefore, the sample means from the paired and unpaired cases converge to the same population mean in RKHS norm. Explicitly, we have:

$$\left\|\boldsymbol{\mu}_x^{\text{paired}}-\boldsymbol{\mu}_x^{\text{unpaired}}\right\|_{\mathcal{H}} \xrightarrow{p} 0,\quad \left\|\boldsymbol{\mu}_y^{\text{paired}}-\boldsymbol{\mu}_y^{\text{unpaired}}\right\|_{\mathcal{H}} \xrightarrow{p} 0. \tag{19}$$

To quantify the rate of convergence, we apply standard results from the theory of Hilbert space-valued random variables (Smola et al., 2007). For all $\varepsilon > 0$:

$$\mathbb{P}\left(\left\|\boldsymbol{\mu}_x^{\text{paired}}-\boldsymbol{\mu}_x\right\|_{\mathcal{H}} \ge \varepsilon\right) \le 2\exp\left(-\frac{m\varepsilon^2}{2K}\right), \tag{20}$$

and similarly for $\boldsymbol{\mu}_x^{\text{unpaired}}$.

Therefore, the convergence rate of the difference between the paired and unpaired empirical mean embeddings is given by:

$$\left\|\boldsymbol{\mu}_x^{\text{paired}}-\boldsymbol{\mu}_x^{\text{unpaired}}\right\|_{\mathcal{H}} = \mathcal{O}_p\left(\frac{1}{\sqrt{m}}\right). \tag{21}$$

$$\left\|\boldsymbol{\mu}_y^{\text{paired}}-\boldsymbol{\mu}_y^{\text{unpaired}}\right\|_{\mathcal{H}} = \mathcal{O}_p\left(\frac{1}{\sqrt{n}}\right). \tag{22}$$

We now consider the empirical CS divergence functional defined as:

$$f(\boldsymbol{\mu}_x,\boldsymbol{\mu}_y) = -2\log\left(\frac{\langle\boldsymbol{\mu}_x,\boldsymbol{\mu}_y\rangle_{\mathcal{H}}}{\|\boldsymbol{\mu}_x\|_{\mathcal{H}}\|\boldsymbol{\mu}_y\|_{\mathcal{H}}}\right) \tag{23}$$

which is continuous on $\mathcal{H}\setminus\{0\}\times\mathcal{H}\setminus\{0\}$. This condition is satisfied under Assumption 1, which ensures that the population mean embeddings are non-degenerate due to the boundedness and universality of the kernel.

Hence, based on the continuous mapping theorem, we yield:

$$\left|f(\boldsymbol{\mu}_x^{\text{paired}},\boldsymbol{\mu}_y^{\text{paired}})-f(\boldsymbol{\mu}_x^{\text{unpaired}},\boldsymbol{\mu}_y^{\text{unpaired}})\right| \xrightarrow{p} 0 \tag{24}$$

That is,

$$\left|\hat{D}_{\text{CS}}^{\text{paired}}-\hat{D}_{\text{CS}}^{\text{unpaired}}\right| \xrightarrow{p} 0 \tag{25}$$

as $m,n\to\infty$.

Furthermore, under Assumption 1(b), the CS divergence functional $f(\cdot, \cdot)$ is Lipschitz continuous, which allows us to quantify the rate of convergence. Using the previously established rates, we can conclude:

$$\left| f(\boldsymbol{\mu}_x^{\text{paired}}, \boldsymbol{\mu}_y^{\text{paired}}) - f(\boldsymbol{\mu}_x^{\text{unpaired}}, \boldsymbol{\mu}_y^{\text{unpaired}}) \right| = \mathcal{O}_p \left( \frac{1}{\sqrt{m}} + \frac{1}{\sqrt{n}} \right). \tag{26}$$

Thus, the convergence rate is bounded by:

$$\left| \hat{D}_{\text{CS}}^{\text{paired}} - \hat{D}_{\text{CS}}^{\text{unpaired}} \right| = \mathcal{O}_p \left( \frac{1}{\sqrt{m}} + \frac{1}{\sqrt{n}} \right) \leq C \cdot \left( \frac{1}{\sqrt{m}} + \frac{1}{\sqrt{n}} \right) \tag{27}$$

where $C$ is a constant that depends on the properties of the kernel function and the Lipschitz constant of the empirical CS divergence. $\qquad \square$

# B  DETAILS ABOUT 3D-MESHPREF

## B.1  DATASET SAMPLE VISUALIZATION

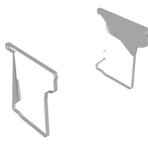

3D model of a two-seater sofa with a backrest and armrests.
Reward: 0.51

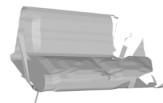

3D model of a two-seater sofa with a slanted backrest, curved armrests, and varying backrest heights.
Reward: 2.01

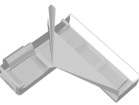

L-shaped sectional sofa with backrest and two armrests.
Reward: 1.61

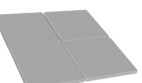

A tall, thin rectangular prism with a smooth surface, made of a light gray material.
Reward: 4.31

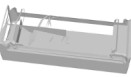

3D model of a daybed with a slatted base, curved backrest, seat, armrest, and footrest.
Reward: 2.13

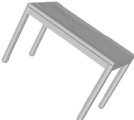

A 3D rectangular table with a rectangular top and base.
Reward: 4.88

Figure 8: Example instances from our large-scale preference-unpaired dataset 3D-MeshPref.

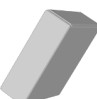

A 3D object featuring a cuboid with trapezoidal bases.
Reward: 4.81

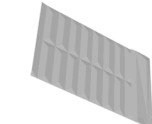

A flat, rectangular surface with slightly rounded edges, made of a smooth material.
Reward: 4.13

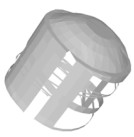

A cylinder with a spherical base and top.
Reward: 2.04

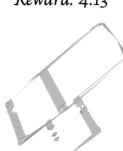

3D model of a three-tier cart with a shelf on each tier.
Reward: 0.24

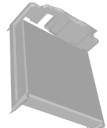

King size platform bed with a slatted base.
Reward: 3.63

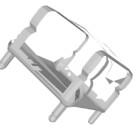

3D model of a cuboid footstool with a slanted top and bottom, royalty-free.
Reward: 1.33

Figure 9: Example instances from our large-scale preference-unpaired dataset 3D-MeshPref.

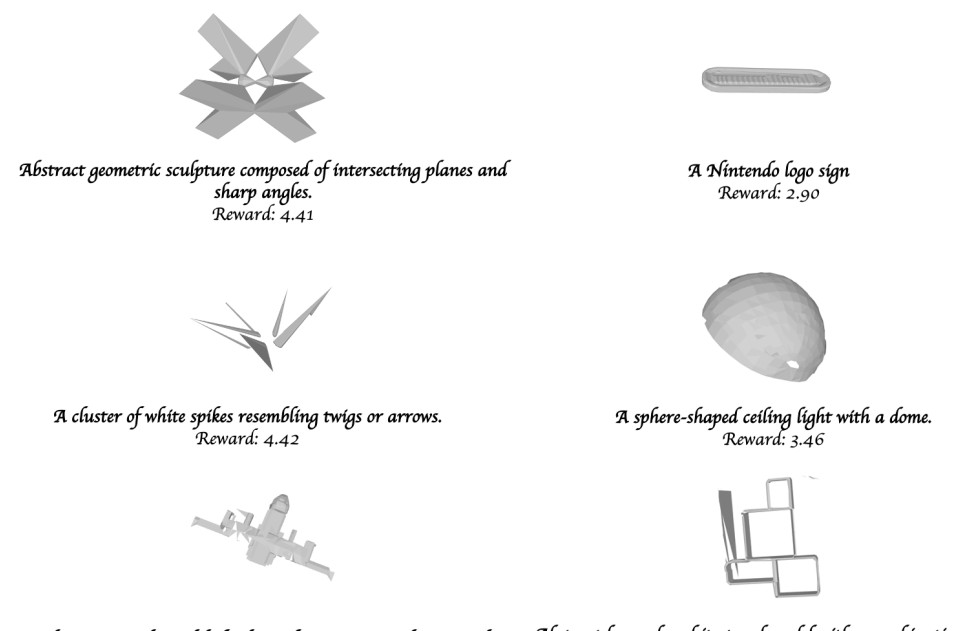

Abstract geometric sculpture composed of intersecting planes and sharp angles.
Reward: 4.41

A Nintendo logo sign
Reward: 2.90

A cluster of white spikes resembling twigs or arrows.
Reward: 4.42

A sphere-shaped ceiling light with a dome.
Reward: 3.46

Monochromatic scale model of a large, four-engine turboprop military transport aircraft with a high T-tail.
Reward: 2.16

Abstract layered architectural model with a combination of planes and textured blocks.
Reward: 3.10

Figure 10: Example instances from our large-scale preference-unpaired dataset 3D-MeshPref.

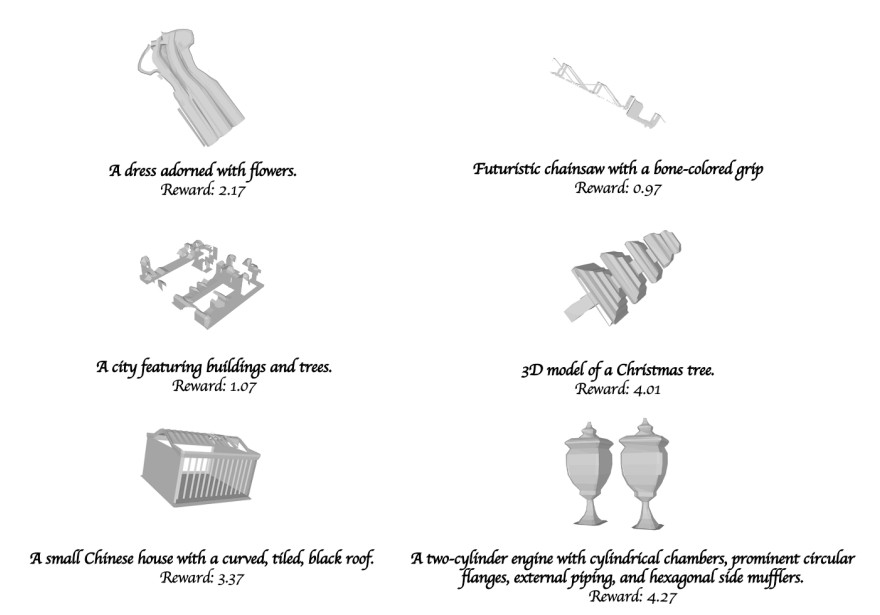

A dress adorned with flowers.
Reward: 2.17

Futuristic chainsaw with a bone-colored grip
Reward: 0.97

A city featuring buildings and trees.
Reward: 1.07

3D model of a Christmas tree.
Reward: 4.01

A small Chinese house with a curved, tiled, black roof.
Reward: 3.37

A two-cylinder engine with cylindrical chambers, prominent circular flanges, external piping, and hexagonal side mufflers.
Reward: 4.27

Figure 11: Example instances from our large-scale preference-unpaired dataset 3D-MeshPref.

As presented in Figs. 8, 9, 10, 11, we provide diverse 3D mesh instances in our curated 3D-MeshPref. 3D-MeshPref 3D assets are semantically diverse, high-quality. Each example instance contains a text prompt, 3D asset, and a preference reward score.

## B.2 DATASET CONSTRUCTION RATIONALE

We detail the rationale behind our design choices for each component and describe the measures we took to ensure data quality.

**(a) Meshilization Method.** We explored both traditional point cloud-to-mesh methods, such as Alpha Shapes (Edelsbrunner & Mücke, 1994), Ball Pivoting Algorithm (Bernardini et al., 2002), Poisson Reconstruction (Kazhdan et al., 2006), and deep learning-based approaches, including MeshAnythingV2 (Chen et al., 2024) and DeepMesh (Zhao et al., 2025). Through systematic comparisons, we found that MeshAnythingV2, particularly with its adjacent mesh tokenization design, consistently outperforms other alternatives in both mesh quality and computational efficiency. Based on these observations, we selected MeshAnythingV2 as it offers the best trade-off among current methods.

**(b) Scoring.** Given the current scarcity of 3D reward models, we adopted Llama-Mesh (Wang et al., 2024) — a large language model fine-tuned for 3D mesh evaluation — which, to our knowledge, is the most capable open-source solution for mesh scoring. Its strong mesh understanding and generalization ability derive from its large-scale supervised training data and its alignment with natural language instruction, making it well-suited for scoring prompt-conditioned meshes in our setting.

To further validate the scoring capability of Llama-Mesh, we follow the DeepMesh (Zhao et al., 2025) preference pair construction pipeline and construct a preference mesh dataset with 300 preference paired 3D meshes. Each pair includes a human-verified positive and negative example. Llama-Mesh evaluates all samples using the annotation criteria described in Appendix F.2 and produces a ranking for each pair. Comparing these rankings with the ground-truth preferences, we find that Llama-Mesh correctly identifies the preferred mesh in 93.6% of cases, demonstrating strong consistency with human judgment and supporting its suitability as an initial scorer for constructing 3D-MeshPref.

Due to the scarcity of large-scale human-annotated 3D data, prior work in 3D reward modeling (Ye et al., 2024) and 3D generation (Zhao et al., 2025) often relies on automated 3D asset pipelines with human verification. 3D-MeshPref adopts this practical approach, enabling scalable and effective training of RewardCS while maintaining alignment with human preference.

**(c) Human Refinement.** We recognize that LLM-based ratings may sometimes overestimate mesh quality relative to human perception. To address this, we incorporate manual verification and correction into the data pipeline to ensure the final dataset reflects human preferences more faithfully.

In addition, to adapt the 3D reward model to intermediate assets generated during SDS optimization, we augment the dataset with over 10,000+ meshes optimized using SDS-based methods, such as DreamFusion (Poole et al., 2022) and MVDream (Shi et al., 2023b), under text prompts in ABO and Objaverse. These meshes are sampled at early stages of the optimization process. Thus, we sample meshes not only at the final stages of SDS optimization but also at early stages of the process. By including these intermediate 3D geometries in 3D-MeshPref, the reward model is exposed to a wider spectrum of possible outputs, from coarse to refined structures.

In summary, we carefully selected the most capable mesh generation and scoring models currently available, and augmented them with human refinement to construct 3D-MeshPref. In addition, we include the intermediate 3D meshes across the SDS optimization in 3D-MeshPref to further extend the domain of our proposed RewardCS.

# C  DETAILS ABOUT REWARDCS

## C.1  DESIGN RATIONALE FOR REWARDCS ARCHITECTURE

Existing mesh encoders typically fall into two categories: graph-based and transformer-based. Transformer-based encoders like MeshMAE offer superior geometric representations given the mesh input compared to the graph-based encoders, while there is currently no mesh encoder that jointly models both high-resolution geometry and texture semantics due to lack of large-scale 3D mesh data. Thus, we adopt MeshMAE, a transformer-based encoder, in RewardCS due to its strong performance in capturing global geometric structure and its robustness to partial inputs and structural noise. MeshMAE is pretrained using masked autoencoding on large-scale 3D mesh data, making it well-suited for downstream tasks requiring shape understanding.

## C.2  ABLATION STUDY ON LATENT-SPACE SEPARATION LOSS

We provide empirical evidence demonstrating the necessity of the separation loss $\mathcal{L}_{div}$ in Equation. 4 in RewardCS. Our findings validate that this component improves embedding quality and text-to-3D performance.

Table 4: Clustering performance under different $\lambda$ values.

| $\lambda$ | SCI $\uparrow$ | CHI $\uparrow$ | DBI $\downarrow$ |
|---|---|---|---|
| 0 | -0.21 | 0.52 | 13.65 |
| 0.5 | 0.38 | 50.23 | 6.89 |
| 1 | 0.64 | 212.92 | 0.97 |
| 10 | 0.58 | 340.30 | 0.86 |

We conduct ablations with four different $\lambda$ values: 0, 0.5, 1, and 10. For each setting, we train a separate RewardCS model and evaluate the quality of the learned embeddings using three standard clustering metrics across 10 test batches from the 3D-MeshPref dataset: Silhouette Coefficient Index (SCI) (Rousseeuw, 1987), Calinski–Harabasz Index (CHI) (Caliński & Harabasz, 1974), and Davies–Bouldin Index (DBI) (Davies & Bouldin, 2009). Across all three metrics shown in Table 4, we observe consistent improvements in cluster separability between preferred and dispreferred samples as $\lambda$ increases, confirming that introducing $\mathcal{L}_{div}$ effectively enhances the structure of the latent space for reward learning.

Table 5: Performance on 3D Generation using RewardCS trained under different $\lambda$ values.

| $\lambda$ | IR | VR | GA | T-A(M) | 3DP(M) | G-T(M) | T-A(U) | 3DP(U) | G-T(U) |
|---|---|---|---|---|---|---|---|---|---|
| 0 | -0.61 | -4.10 | 1.97 | 2.91 | 2.80 | 2.62 | 2.25 | 2.48 | 2.23 |
| 0.5 | -0.56 | -3.28 | 2.88 | 3.18 | 3.31 | 3.72 | 3.17 | 3.23 | 3.01 |
| 1 | -0.43 | -2.10 | 2.95 | 3.21 | 4.09 | 4.02 | 3.27 | 3.69 | 3.65 |
| 10 | -1.48 | -4.28 | 1.26 | 2.38 | 2.23 | 2.07 | 2.21 | 2.06 | 2.07 |

Then We incorporate RewardCS trained with different $\lambda$ values into MVDream and evaluate different settings on GPTEval3D. We report results across the same metrics in our Tab. 1 and Tab. 2, where MLLM evaluation is denoted as M and user study is denoted as U. As shown in Tab. 5, RewardCS trained with $\lambda = 0.5$ and 1 consistently outperforms the $\lambda = 0$ variant across all metrics, with $\lambda = 1$ yielding the strongest performance overall. These results demonstrate the positive impact of on practical 3D asset quality.

These empirical findings support our theoretical results (Theorem 1), which show that optimizing CS divergence between unpaired preferred and dispreferred embeddings induces meaningful reward separation. Our experiments confirm that this theoretical formulation can enhance embedding quality and improve generation fidelity in practice.

## C.3 Ablation Study on RewardCS and MeshMAE

**Generalization Ability of RewardCS.** Our evaluation (Table 2) is based on the GPTEval3D benchmark, which consists of 110 test prompts that are out-of-distribution relative to the 3D-MeshPref dataset. These experiments demonstrate that RewardCS variants consistently outperform vanilla text-to-3D generation baselines, producing 3D assets with better 3DP and G-T, and thus it can generalize well.

we curated a novel evaluation setup to measure generalization of RewardCS. We sampled 200 distinct noun tags from the Recognize Anything (Zhang et al., 2024) tag set, and refined them into descriptive prompts using DeepSeek-R1. These prompts do not overlap with those in Cap3D, ensuring zero prompt-level data leakage. We used these prompts to generate 3D assets using two text-to-3D backbones: MVDream and DreamFusion, each generating 100 assets. These outputs are distinct from Cap3D assets and represent unseen real-world inference settings.

Then to assess how well RewardCS scores align with human-perceived quality, we used LLaMA-Mesh, a strong baseline scorer, to evaluate the generated assets based on three criteria E: (1) prompt alignment, (2) structural plausibility, and (3) geometric completeness. We then compared RewardCS scores against LLaMA-Mesh scores using: Mean Squared Error (MSE) to quantify absolute alignment between RewardCS and LLaMA-Mesh outputs; and Classification Accuracy (Acc) by discretizing the score interval [0,5] into 20 equal bins. A prediction is counted as correct if the RewardCS and LLaMA-Mesh scores fall in the same bin.

RewardCS achieves 0.06 and 83.10 in MSE and Acc, respectively, and thus show strong alignment with LLaMA-Mesh across both metrics, even on generated samples that lie far outside the training distribution. These results demonstrate that RewardCS effectively generalizes to high-quality 3D assets produced by text-to-3D pipelines like MVDream and DreamFusion, despite being trained on preference data from a different source.

**Robustness of MeshMAE.** Our evaluation is conducted on the GPTEval3D benchmark. The results show that RewardCS variants consistently outperform vanilla text-to-3D generation baselines, producing high-quality 3D assets. This performance indicates that MeshMAE—the mesh encoder used in RewardCS—is robust to geometric variation and is capable of effectively guiding the generation framework.

To further evaluate whether MeshMAE can effectively capture mesh quality, we constructed 100 paired prompts from 100 unique noun tags in the Recognize Anything (Zhang et al., 2024) dataset. Each pair consists of a positive prompt with detailed object descriptions and a negative prompt, derived from the positive one, but modified to include geometry-related defects (e.g., holes, cracks, asymmetry, and surface degradation). Then, using MVDream, we generated 3D assets from these prompt pairs, producing meshes with topology differences.

To establish reference scores, we used LLaMA-Mesh, a strong mesh-level quality assessor, to score each asset on three criteria: (1) prompt alignment, (2) realism and structural plausibility, and (3) geometric completeness. This evaluation process reflects the same preference criteria used in constructing the 3D-MeshPref dataset.

Finally, we used RewardCS (with MeshMAE as the encoder) to score the same 3D assets and compared the outputs to those from LLaMA-Mesh using: Mean Squared Error (MSE) between RewardCS and LLaMA-Mesh scores, and Classification Accuracy (Acc) by dividing the score range into 20 bins and checking score agreement within bins.

Reward CS achieves the MSE of 0.10 and Acc of 80.98. The results show strong alignment between RewardCS and LLaMA-Mesh across both metrics, indicating that MeshMAE captures meaningful distinctions in geometric quality and structural attributes across meshes with varying topology.

# D  DETAILS ABOUT REWARDCS

## D.1  SDS ALGORITHM

We consider a differentiable rendering function $g(\psi, m)$ which maps a 3D scene representation parameterized by $\psi \in \Psi$ and a camera parameter $m \in \mathcal{M}$ to a 2D image $x$.

Let $\phi(x_t \mid c)$ denote a pretrained 2D conditional diffusion model, where $x_t$ is the noisy image at diffusion timestep $t$ and $c$ is a conditioning text prompt. The corresponding noise prediction network is $\epsilon_\phi(x_t, t, c)$.

Following the SDS framework (Poole et al., 2022), we can compute the gradient of the SDS loss with respect to the 3D parameters $\psi$ via the chain rule at step $t$:

$$\nabla_{\psi_t} \mathcal{L}_{\text{SDS}}(\psi_t; m, c) = \mathbb{E}_{t,\epsilon,m} \left[ w(t) \big( \epsilon_\phi(x_t, t, c) - \epsilon \big) \frac{\partial x_t}{\partial \psi_t} \right], \tag{28}$$

where the latent $x_t$ is obtained by

$$x_t = \sqrt{\bar{\alpha}_t} \, g(\psi_t, m) + \sqrt{1 - \bar{\alpha}_t} \, \epsilon, \tag{29}$$

with $\bar{\alpha}_t$ the cumulative product of the diffusion schedule, $\epsilon \sim \mathcal{N}(0, I)$, and $w(t)$ a timestep-dependent weighting factor.

## D.2  DISCUSSION ON ADAPTIVE MESH FUSION

To assess the impact of the Adaptive Mesh Fusion (AMF) step on geometric fidelity, we evaluate two standard surface-level metrics that quantify the deviation between the mesh before ($A$) and after ($B$) applying AMF. All meshes are normalized by the diagonal length of their bounding box for scale-invariant evaluation.

**Chamfer Distance (CD)** measures the average distance from points on one surface to the closest points on the other surface, computed as:

$$\text{CD}(A, B) = \frac{1}{|A|} \sum_{x \in A} \min_{y \in B} \|x - y\|_1 + \frac{1}{|B|} \sum_{y \in B} \min_{x \in A} \|y - x\|_1, \tag{30}$$

where $|A|$ and $|B|$ denote the cardinality of points on meshes $A$ and $B$, respectively.

**95%-Hausdorff Distance (HD)** captures the worst-case deviation between the two surfaces while remaining robust to outliers:

$$\text{HD}(A, B) = \max \left( \text{Quantile}_{95} \left[ \min_{y \in B} \|x - y\|_2 \right]_{x \in A}, \ \text{Quantile}_{95} \left[ \min_{x \in A} \|y - x\|_2 \right]_{y \in B} \right). \tag{31}$$

We evaluate 40 prompts from GPTEval3D for three baselines: MVDream, DreamFusion, and DreamCraft3D. Measurements are reported for meshes before and after AMF at SDS steps 4000, 8000, 12000, 16000, and 20000. The averaged results are shown in Table 6.

Table 6: Geometric deviation of meshes before and after the Adaptive Mesh Fusion step in DreamCS. CD and HD denote Chamfer Distance and Hausdorff Distance (95%-quantile), respectively.

| Method | CD ($\times 10^{-3}$) | HD ($\times 10^{-2}$) |
|---|---|---|
| MVDream | 3.1 | 2.9 |
| DreamFusion | 4.0 | 3.2 |
| DreamCraft3D | 2.7 | 2.4 |

As shown Table 6, the geometric deviation introduced by AMF is negligible (less than 0.35% average error), confirming that AMF preserves geometric details throughout optimization.

# E  IMPLEMENTATION DETAILS

## E.1  EXPERIMENTAL SETUP

The conversion of point clouds to meshes was performed using the MeshAnythingV2 framework (Chen et al., 2024) across 8 L40-S GPUs. For dataset annotation, we employed 4 L40-S GPUs. The training of the reward model, DreamCS, was conducted on 8 x L40-S GPUs using our proposed 3D-MeshPref dataset. The reward model was fine-tuned based on MeshMAE checkpoint for 100 epochs using the AdamW optimizer with a learning rate of 1e-3 in total around 90 GPU hours. The CS divergence is computed at the batch level, and in our setup we fix the total batch size to $m + n = 256$. The Geometry-Asset Alignment Reward (GA) metric, derived from RewardCS, which is trained on approximately 10,000 annotated meshes from the Objaverse-XL subset of Cap3D (Luo et al., 2023). Since GA provides a model-agnostic and architecture-agnostic measure of geometric–semantic consistency, it offers a fair basis for comparing DreamCS with other text-to-3D generation approaches. This metric functions as an independent evaluator for assessing the alignment between 3D mesh geometry and textual descriptions.

## E.2  ANNOTATION PIPELINES

**Llama-Mesh Annotation.** To annotate the quality of the converted 3D meshes in our dataset, we employ a structured three-criterion annotation rubric. Each criterion is rated on a Likert scale from 0.00 to 5.00, where 5.00 denotes the highest level of performance. The final score is calculated as the average of the three individual scores:

- **Alignment with the provided description:** Measure the degree to which the 3D mesh accurately reflects the corresponding textual description. Deductions are applied for any semantic misalignment, omission of key elements, or inclusion of irrelevant features.
- **Plausibility and realism of the 3D structure:** Assess the physical plausibility and visual realism of the 3D mesh. Artifacts that violate basic physical principles or render the object visually implausible result in lower scores.
- **Completeness of geometry:** Evaluate the geometric and textural completeness of the 3D mesh. Penalties are applied for missing components, structural incompleteness, or absent/erroneous textures that compromise model integrity.

To control the potential biases in the automated assessments, rather than relying on a single annotation process, we score every mesh with three independent inference runs of Llama-Mesh and use the averaged score.

**Human Refinement.** Meshes with a Llama-Mesh score greater than 2.0 are subsequently re-evaluated by human annotators using the interactive annotation panel shown in Figure 12. The same three-criterion rubric is applied, ensuring consistency with the automated Llama-Mesh scoring while allowing humans to correct potential errors or biases. Human raters assign scores on the same 0.0–5.0 Likert scale, and the final score is computed as the average across the three criteria, providing a refined assessment of each mesh's quality.

- **Alignment with the provided description:** Measure the degree to which the 3D mesh accurately reflects the corresponding textual description. Deductions are applied for any semantic misalignment, omission of key elements, or inclusion of irrelevant features.
- **Plausibility and realism of the 3D structure:** Assess the physical plausibility and visual realism of the 3D mesh. Artifacts that violate basic physical principles or render the object visually implausible result in lower scores.
- **Completeness of geometry:** Evaluate the geometric and textural completeness of the 3D mesh. Penalties are applied for missing components, structural incompleteness, or absent/erroneous textures that compromise model integrity.

## E.3  EVALUATION PIPELINES

**MiniCPM-o Evaluation.** To evaluate the quality of the generated 3D assets across MVDream-based pipelines, we use MiniCPM-o based on a three-part evaluation rubric. Each criterion is rated

on a Likert scale from 0.00 to 5.00, where 5.00 denotes the highest performance. The final score is calculated as the average of the three individual scores:

- **Text Prompt and Asset Alignment:** Assess the semantic fidelity between the multi-view renderings and the original textual prompt. The images should depict all referenced entities and contextual elements with accurate visual attributes. Annotators are instructed to first describe the output of each model and then evaluate how comprehensively and accurately the visual content reflects the prompt.

- **3D Plausibility:** Evaluate the inferred 3D structure by examining the set of RGB and normal images. Reviewers consider the overall realism, solidity, and structural coherence implied by the views. Deductions are made for implausible geometry, unnatural proportions, duplicated or floating parts, and any visual artifacts that suggest physical inconsistency. High-quality outputs should convey convincing, well-formed 3D shapes from all angles.

- **Geometry-Texture Alignment:** Measure the local consistency between geometry and texture across views. Specific attention is paid to whether visual features, such as edges, patterns, or object parts, are coherently represented in both RGB and normal maps. Discrepancies between texture and underlying geometry reduce the score.

**User Study.** To assess the perceptual quality of the generated 3D meshes, we conduct a user study with 30 participants on 30 prompts based on three core criteria across MVDream-based pipelines, as shown in Figures 12 and 13. To avoid bias, models were anonymized, and the response order was randomized. Participants are asked to inspect the 3D meshes from multiple viewpoints and assign a score on a Likert scale from 0.00 to 5.00 for each criterion, where 5.00 denotes the highest performance. The final score is computed as the average of the three individual ratings:

- **Text-Asset Alignment:** Measure how accurately the 3D mesh represents the content described in the original text prompt. Participants assess whether all described objects, attributes, and spatial arrangements are present and correctly rendered in the mesh. Partial or incorrect representations lead to a lower score.

- **3D Plausibility:** Evaluate the realism and structural integrity of the mesh geometry. Participants judge whether the mesh appears physically plausible and free of distortions, unrealistic proportions, or non-functional geometry. 3D meshes with topological errors or unnatural forms are penalized.

- **Geometry-Texture Alignment:** Assess the coherence between the mesh geometry and its texture. Annotators examine whether surface textures appropriately follow the underlying shapes and contours of the geometry. Inconsistencies, such as textures that misalign with surface features or obscure geometric details, result in a lower score.

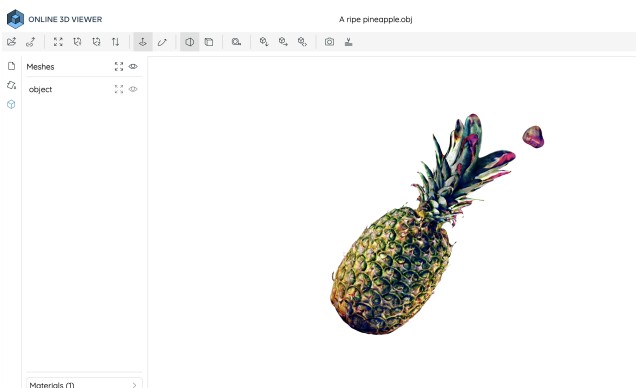

Figure 12: Interface built on Online 3D Viewer used in the user study for evaluating 3D meshes. Participants could interact with the mesh (e.g., rotate, zoom).

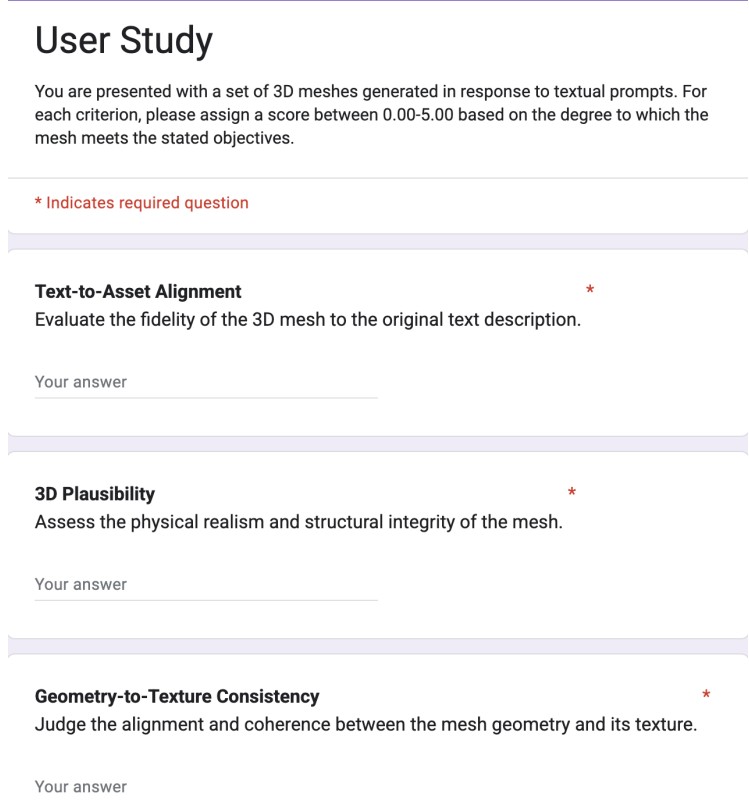

Figure 13: User study questionnaire built on Google Form for evaluating 3D mesh quality. Participants can evaluate each mesh based on (1) Text-to-Asset Alignment, (2) 3D Plausibility, and (3) Geometry-to-Texture Consistency.

# F EXTENDED EMPIRICAL RESULTS

## F.1 COMPARISON WITH 3D CONTROLLABLE GENERATION METHODS

3D controllable generation methods such as MVControl (Li et al., 2023), MT3D (Nath et al., 2025), and DreamControl (Huang et al., 2024) can provide guidance for optimization-based 3D generation, but they rely on strong 3D shape prior or external conditioning 2D inputs and are architecture-dependent, making them difficult to apply in real-world text-to-3D settings. Among them, MVControl is the most compatible with MVDream framework, so we adopt it for a comparison.

Under the MVDream backbone, we adopt the pretrained MVControl checkpoint. Given 30 evaluation prompts from GPTEval3D, we first generate 2D image prior using stable diffusion 3 and then use the MVcontrol method to generate the 3D assets. We report performance using the same metrics in Tables 1, 2, and 3 of our main paper, where T-A, 3DP, and G-T are derived from MLLM evaluation.

Table 7: Comparison with 3D controllable generation method MVControl.

| Model | CP (↑) | VR (↑) | GA (↑) | T-A (↑) | 3DP (↑) | G-T (↑) | Proportion (↓) |
|---|---|---|---|---|---|---|---|
| MVDream | 0.23 | -3.30 | 2.77 | 2.95 | 3.10 | 3.07 | 0.55 |
| MVControl | 0.27 | -3.09 | 2.86 | 3.42 | 3.47 | 3.51 | 0.40 |
| MVDream+Reward3D | 0.27 | -3.13 | 2.85 | 3.38 | 3.34 | 3.20 | 0.46 |
| MVDream+RewardCS | 0.29 | -2.15 | 2.96 | 3.58 | 4.04 | 3.93 | 0.31 |

As shown in Table 7, RewardCS outperforms both MVControl and Reward3D across all metrics, including CP, VR, GA, and especially in 3D plausibility and geometry–texture consistency. Moreover, RewardCS achieves the lowest Janus proportion (0.31), confirming that it effectively mitigates the Janus issue—without requiring any auxiliary conditioning input.

## F.2 COMPARISON WITH ADVANCED TEXT-TO-3D GENERATION PIPELINE

**Advanced mesh-based text-to-3D generation pipeline.** We extend our experiments to advanced multi-stage SDS-based pipelines: ProlificDreamer (Wang et al., 2023b) and DreamCraft3D (Sun et al., 2023) under 30 evaluation prompts from GPTEval3D. Following their default configurations, for ProlificDreamer, we adopt 20000, 10000, and 10000 optimization steps for the object, geometry, and texture stages, respectively. For DreamCraft3D, we use 5000 optimization steps for each of the NeRF, geometry, and texture stages. In addition, we use Stable Diffusion 3 to generate the 2D image prior. We adopt Zero123-XL as the backbone and use Omnidata for depth and normal estimation. For both methods, the rendering resolution is set to 128x128 in the first stage and 512x512 in the subsequent stages. We report performance using the same metrics in Tables 1, 2, and 3 of our main paper, where T-A, 3DP, and G-T are derived from MLLM evaluation.

Table 8: Comparison with multi-stage SDS-based text-to-3D generation pipelines.

| Model | CP (↑) | VR (↑) | GA (↑) | T-A (↑) | 3DP (↑) | G-T (↑) | Proportion (↓) |
|---|---|---|---|---|---|---|---|
| ProlificDreamer | 0.26 | -3.25 | 2.83 | 3.03 | 3.24 | 3.13 | 0.43 |
| +Reward3D | 0.28 | -3.18 | 2.89 | 3.41 | 3.48 | 3.39 | 0.37 |
| +RewardCS | 0.29 | -2.11 | 3.04 | 3.58 | 4.06 | 4.01 | 0.29 |
| DreamCraft3D | 0.28 | -3.20 | 2.84 | 3.11 | 3.26 | 3.16 | 0.41 |
| +Reward3D | 0.29 | -2.96 | 2.90 | 3.43 | 3.51 | 3.45 | 0.35 |
| +RewardCS | 0.30 | -2.08 | 3.13 | 3.61 | 4.11 | 4.07 | 0.24 |

As shown in Table 8, we observe that RewardCS consistently outperforms both the vanilla baselines and the 2D reward variant. In multi-stage pipelines, 2D reward signals are often unstable and require sensitive hyperparameter tuning; when optimization collapses in an early stage, the degradation propagates to later stages. This underscores the robustness of 3D reward guidance.

Additionally, RewardCS yields strong improvements on DreamCraft3D, which also adopts DMTet as its underlying 3D representation. This structural compatibility allows RewardCS to provide more accurate and geometry-aware feedback on-the-fly.

**Advanced mesh-based text-to-3D generation pipeline.** To further demonstrate its generality, we apply RewardCS to finetune the advanced end-to-end text-to-3D generation model TRELLIS-text-base (Xiang et al., 2025). We construct a finetuning dataset by randomly selecting 100K prompts from the TRELLIS-500K training set. For each prompt, a 3D asset is generated and differentially rendered to DMTet or 2D multi-view images. We then use either Reward3D or RewardCS as the reward guidance to finetune the TRELLIS backbone, following an objective similar to Equation 4.

Given 60 evaluation prompts from GPTEval3D, we report performance using the same metrics in Tables 1, 2, and 3 of our main paper, where T-A, 3DP, and G-T are derived from MLLM evaluation.

Table 9: Comparison of TRELLIS with reward-based enhancements.

| Model | CP (↑) | VR (↑) | GA (↑) | T-A (↑) | 3DP (↑) | G-T (↑) | Proportion (↓) |
|-------|--------|--------|--------|---------|---------|---------|----------------|
| TRELLIS | 0.29 | -1.53 | 2.86 | 3.13 | 3.30 | 3.51 | 0.43 |
| +Reward3D | 0.28 | -1.47 | 2.90 | 3.20 | 3.35 | 3.58 | 0.39 |
| +RewardCS | 0.30 | -1.39 | 3.02 | 3.25 | 3.43 | 3.60 | 0.35 |

As shown in Table 9, these results demonstrate that RewardCS consistently improves text–asset alignment, 3D plausibility, geometry–texture consistency, and reduces Janus artifacts, outperforming both the base TRELLIS model and the 2D reward variant. This highlights RewardCS's flexibility and effectiveness as a model-agnostic reward signal for advanced end-to-end 3D generation frameworks.

## F.3 QUALITATIVE VISUALIZATION

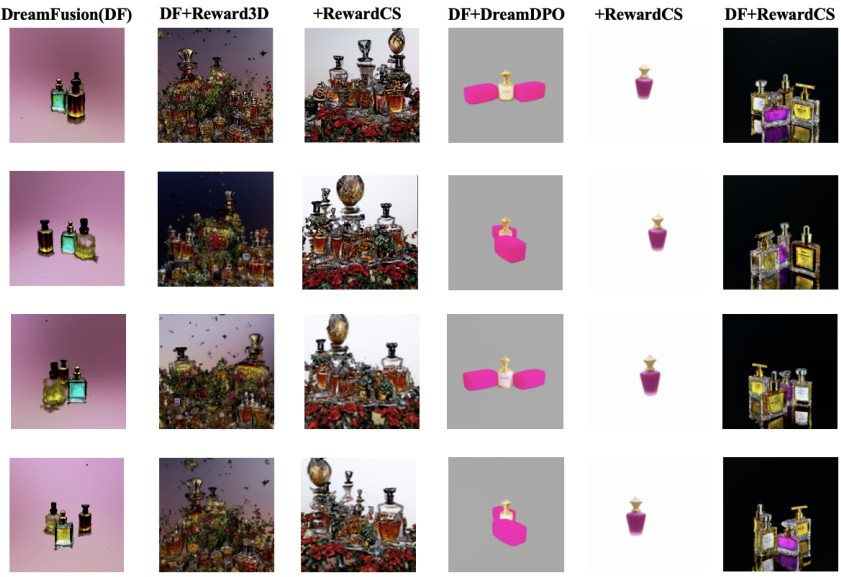

*An assortment of vintage, fragrant perfume on display*

Figure 14: Qualitative comparisons with 1-stage generation pipelines: DreamFusion.

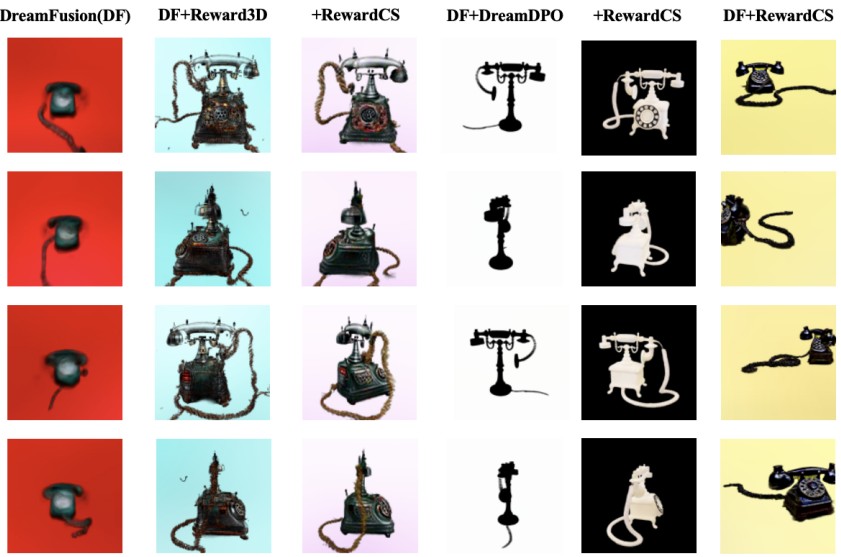

*An old-fashioned rotary phone with a tangled cord*

Figure 15: Qualitative comparisons with 1-stage generation pipelines: DreamFusion.

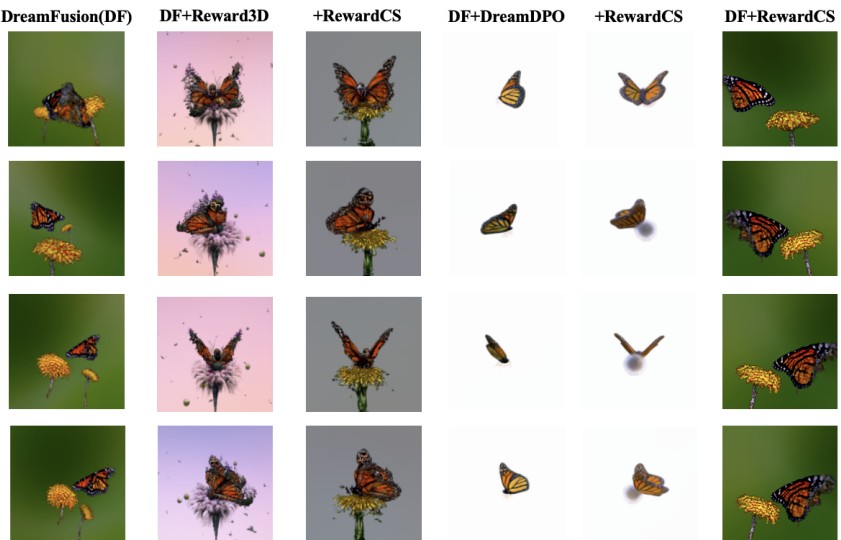

*Orange monarch butterfly resting on a dandelion*

Figure 16: Qualitative comparisons with 1-stage generation pipelines: DreamFusion.

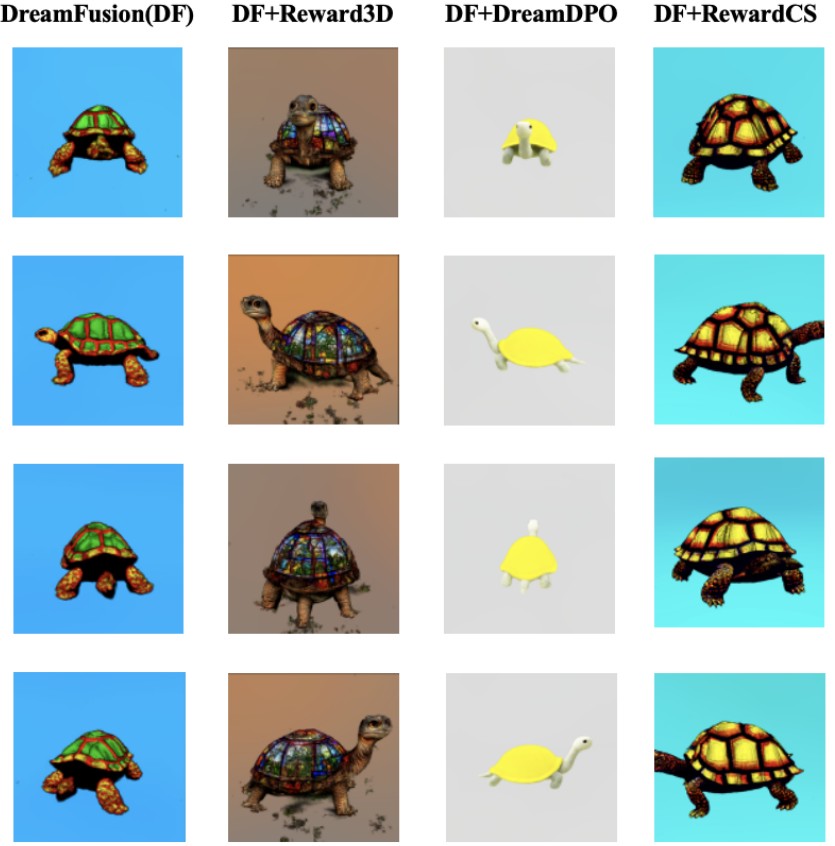

Figure 17: Qualitative comparisons with 1-stage generation pipelines: DreamFusion.

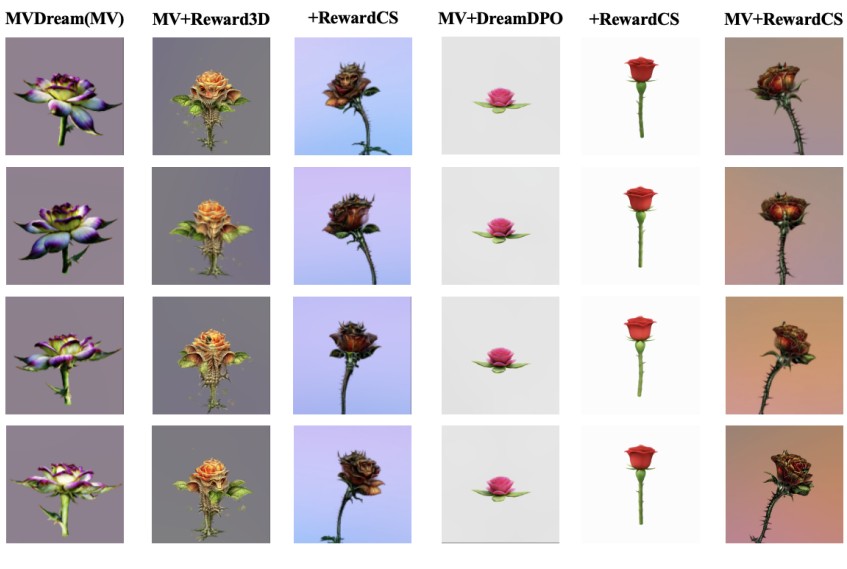

Figure 18: Qualitative comparisons with 1-stage generation pipelines: MVDream.

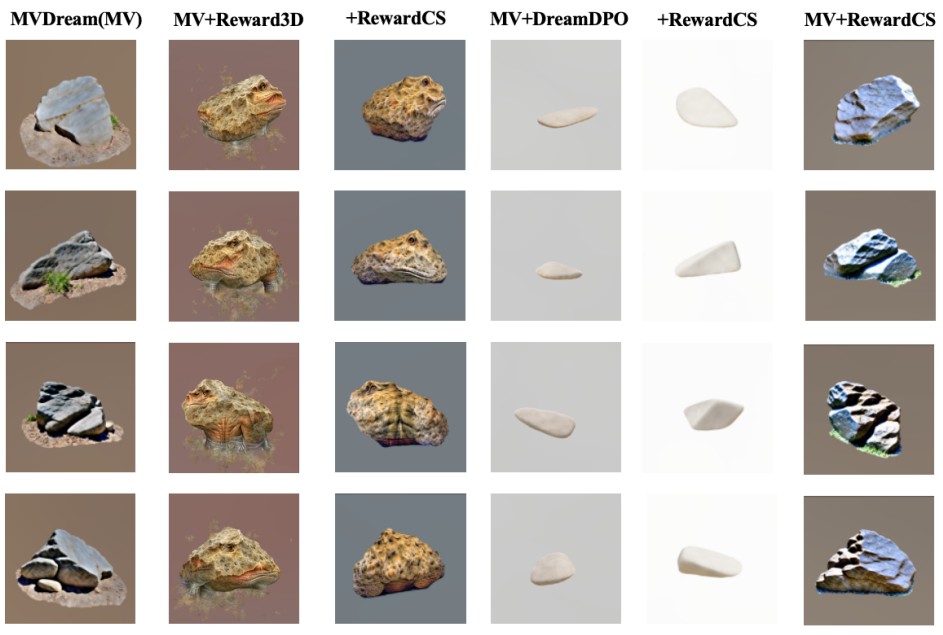

Figure 19: Qualitative comparisons with 1-stage generation pipelines: MVDream.

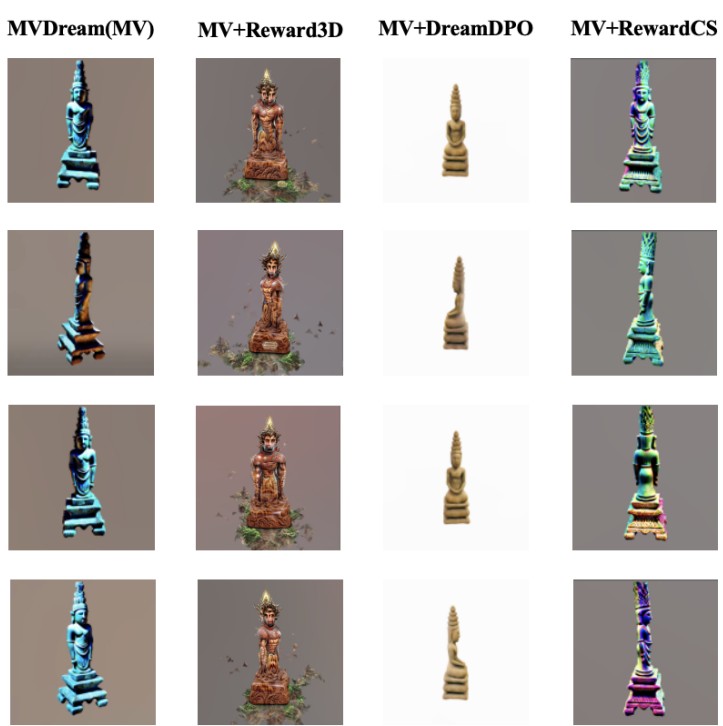

Figure 20: Qualitative comparisons with 1-stage generation pipelines: MVDream.

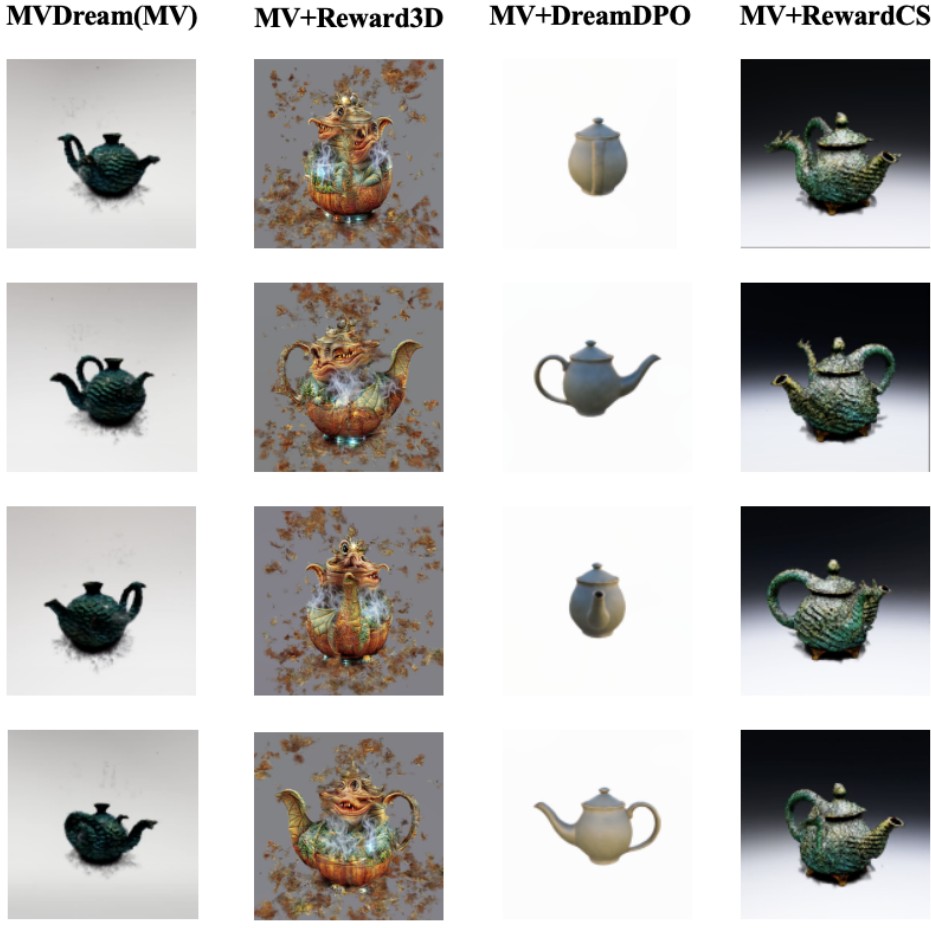

Figure 21: Qualitative comparisons with 1-stage generation pipelines: MVDream.

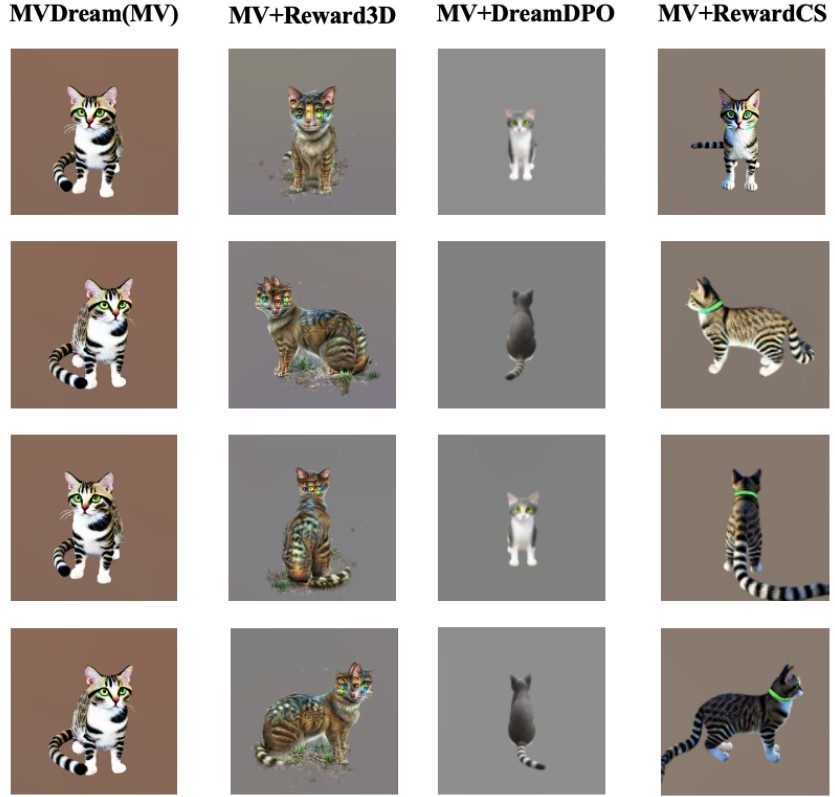

| MVDream(MV) | MV+Reward3D | MV+DreamDPO | MV+RewardCS |

*A cat in a sitting pose with two different colored eyes*

Figure 22: Qualitative comparisons with 1-stage generation pipelines: MVDream.

| MVDream(MV) | MV+Reward3D | MV+DreamDPO | MV+RewardCS |
| --- | --- | --- | --- |

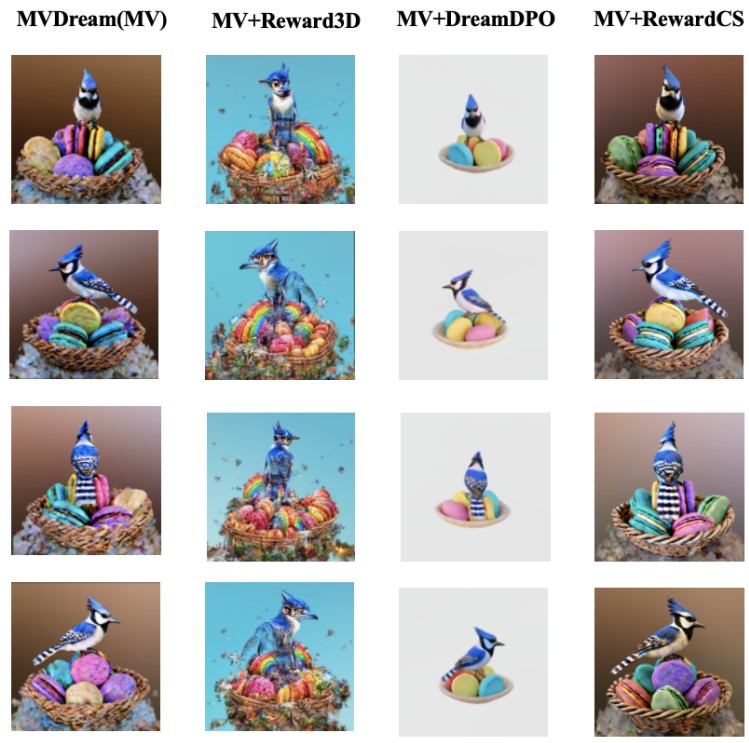

*A DSLR photo of a blue jay standing on a large basket of rainbow macarons*

Figure 23: Qualitative comparisons with 1-stage generation pipelines: MVDream.

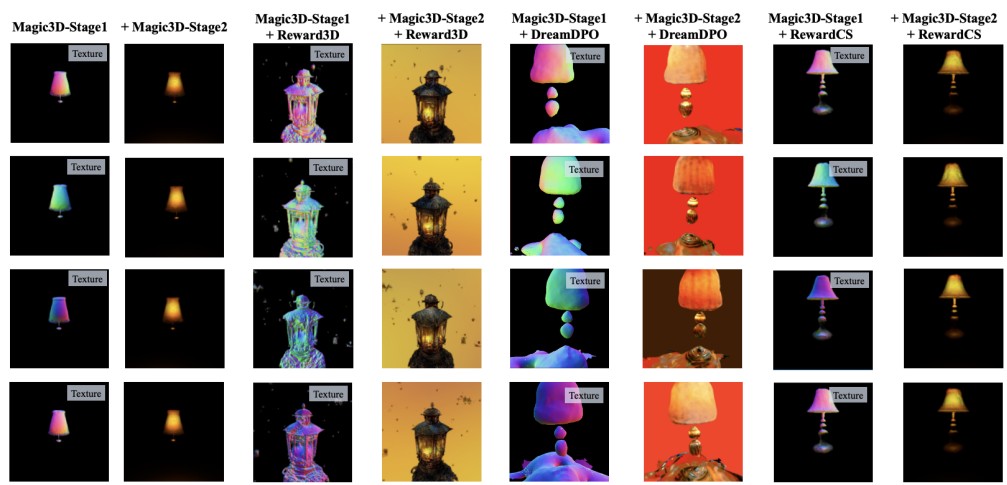

*A vintage table lamp casting a soft, warm glow in a quiet, dark room*

Figure 24: Qualitative comparisons with 2-stage generation pipelines: Magic3D.

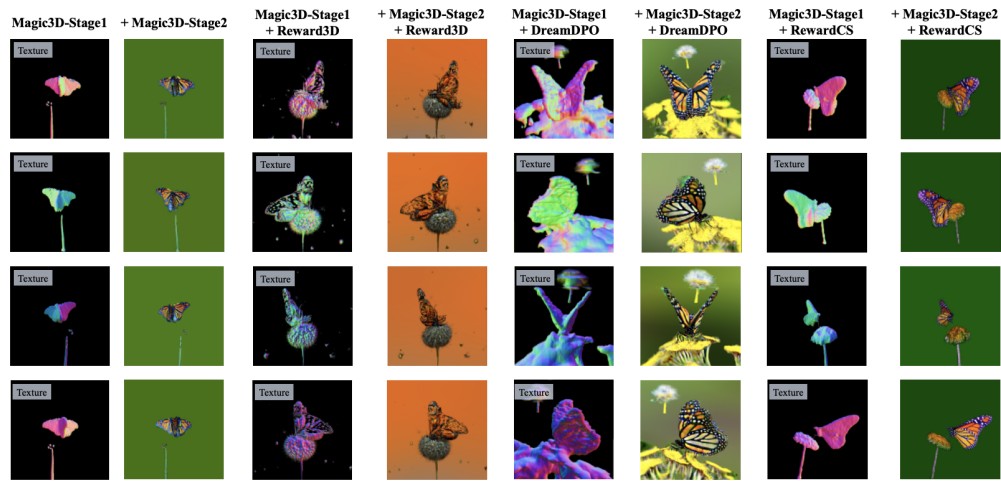

*Orange monarch butterfly resting on a dandelion*

Figure 25: Qualitative comparisons with 2-stage generation pipelines: Magic3D.

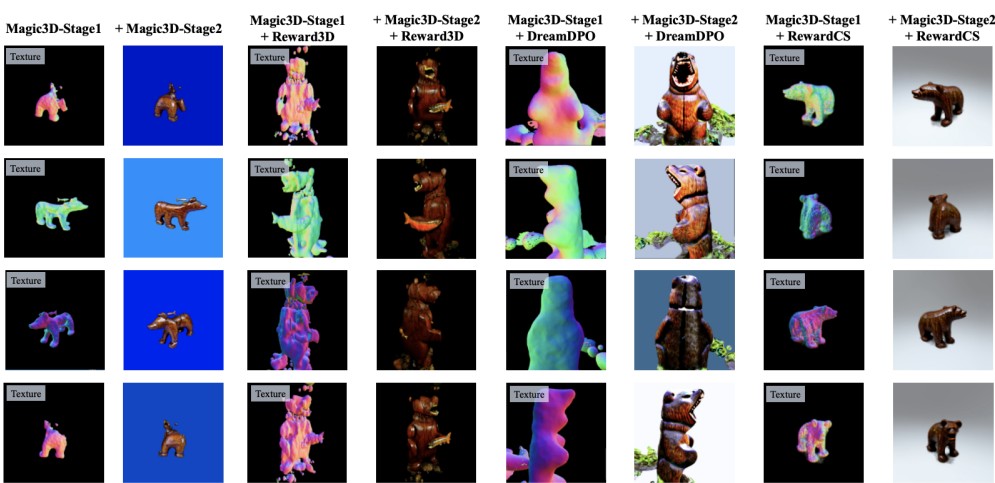

*A carved wooden bear in a preying pose*

Figure 26: Qualitative comparisons with 2-stage generation pipelines: Magic3D.

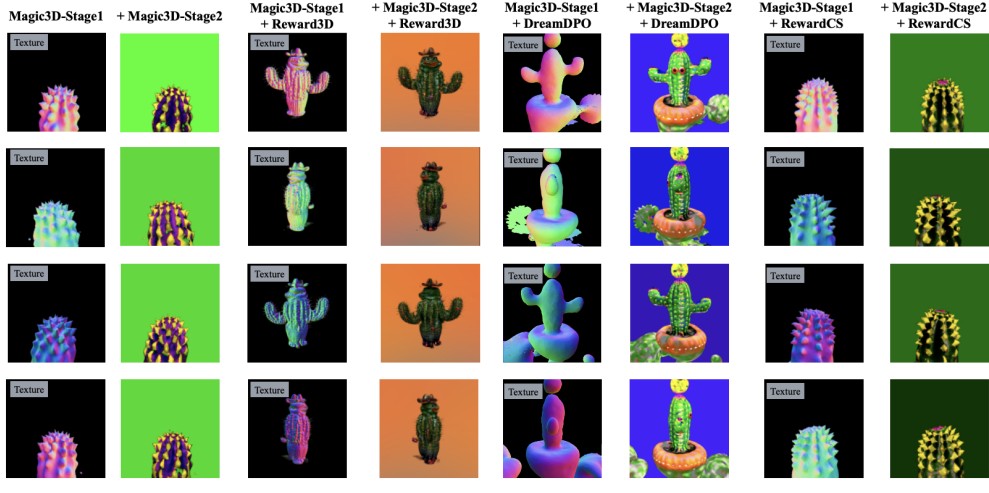

Figure 27: Qualitative comparisons with 2-stage generation pipelines: Magic3D.

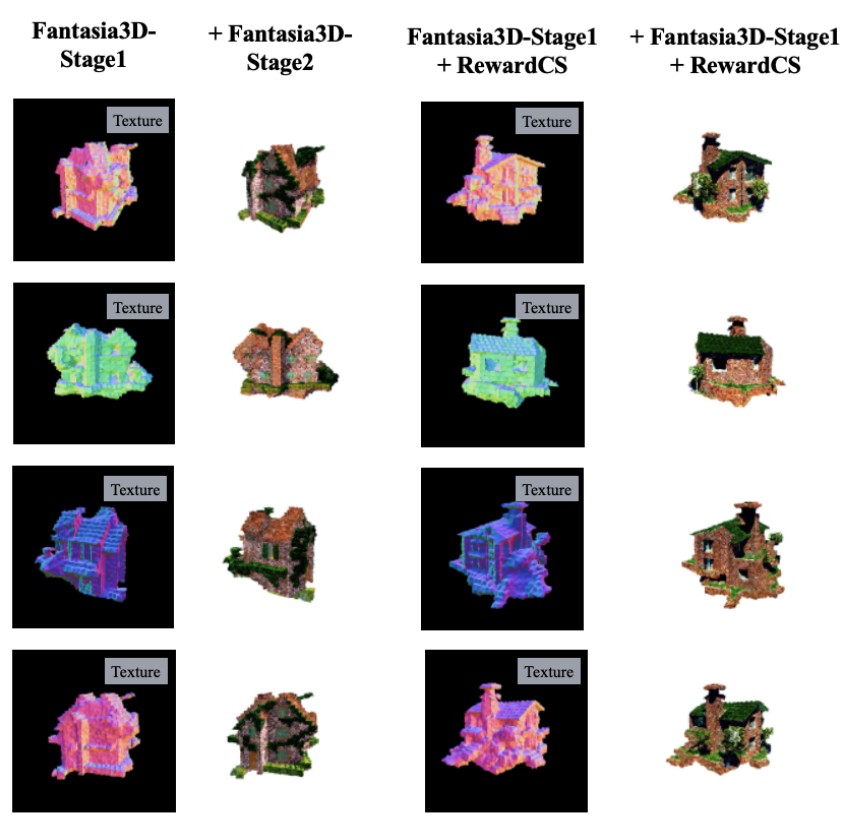

Figure 28: Qualitative comparisons with 2-stage generation pipelines: Fantasia3D.

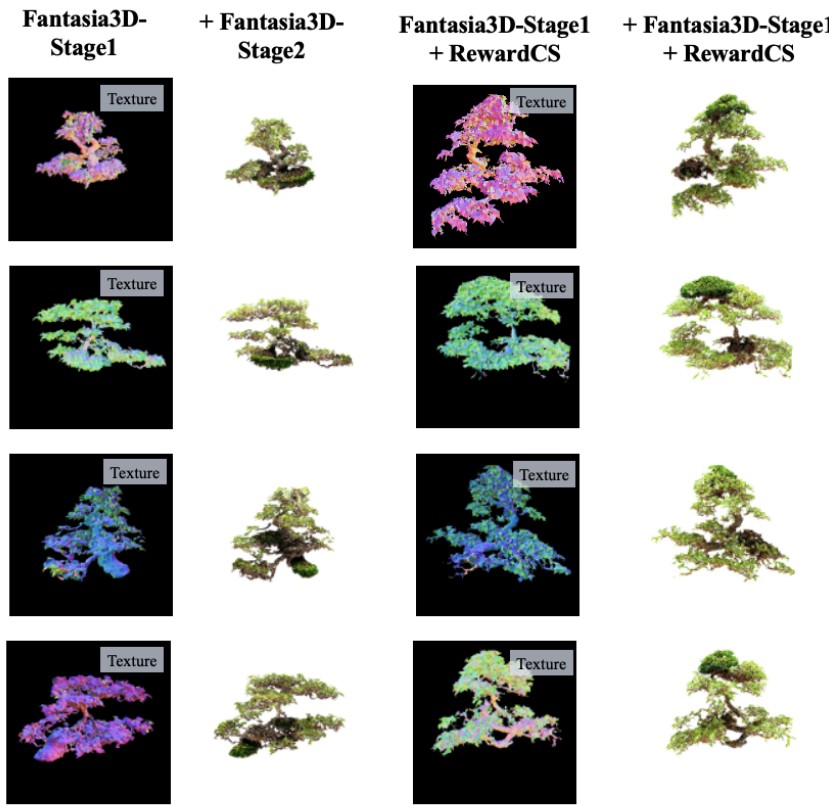

Figure 29: Qualitative comparisons with 2-stage generation pipelines: Fantasia3D.

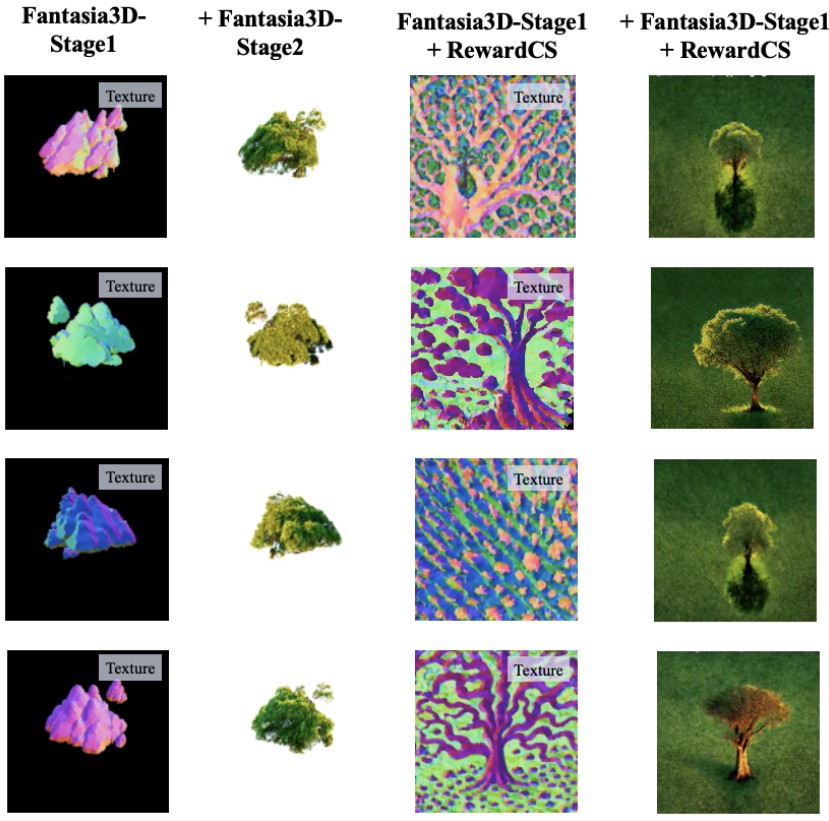

Figure 30: Qualitative comparisons with 2-stage generation pipelines: Fantasia3D.

**Trellis**     **Trellis+Reward3D**     **Trellis+RewardCS**

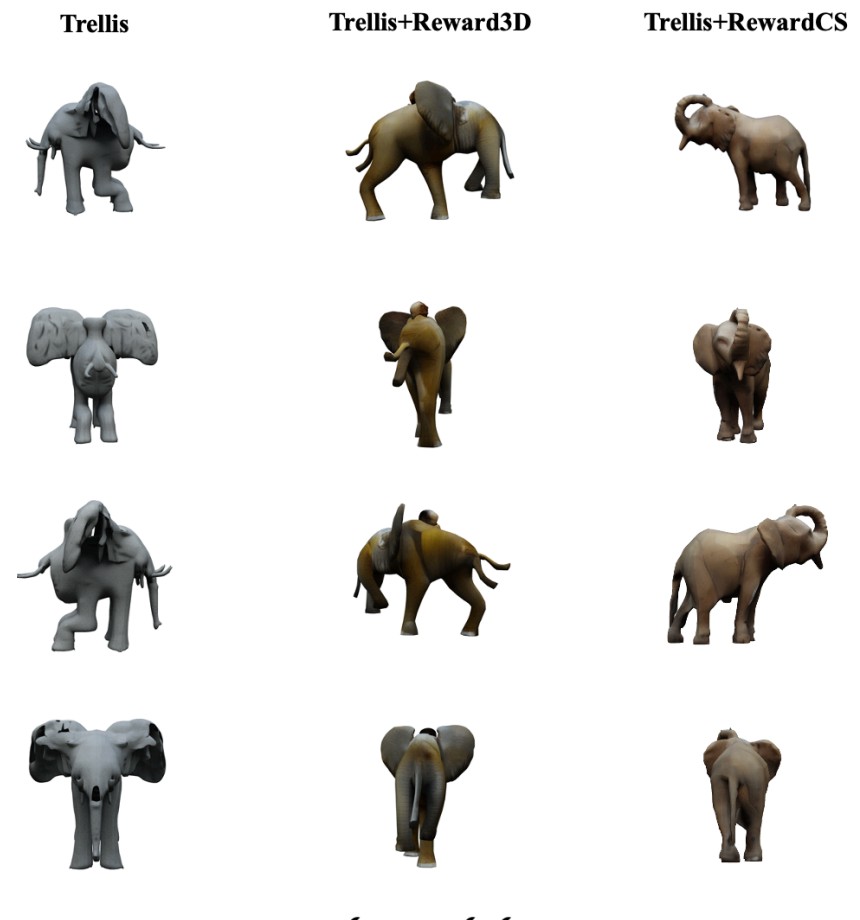

*A dancing elephant*

Figure 31: Qualitative comparisons with the end-to-end generation pipeline: Trellis.

**Trellis**     **Trellis+Reward3D**     **Trellis+RewardCS**

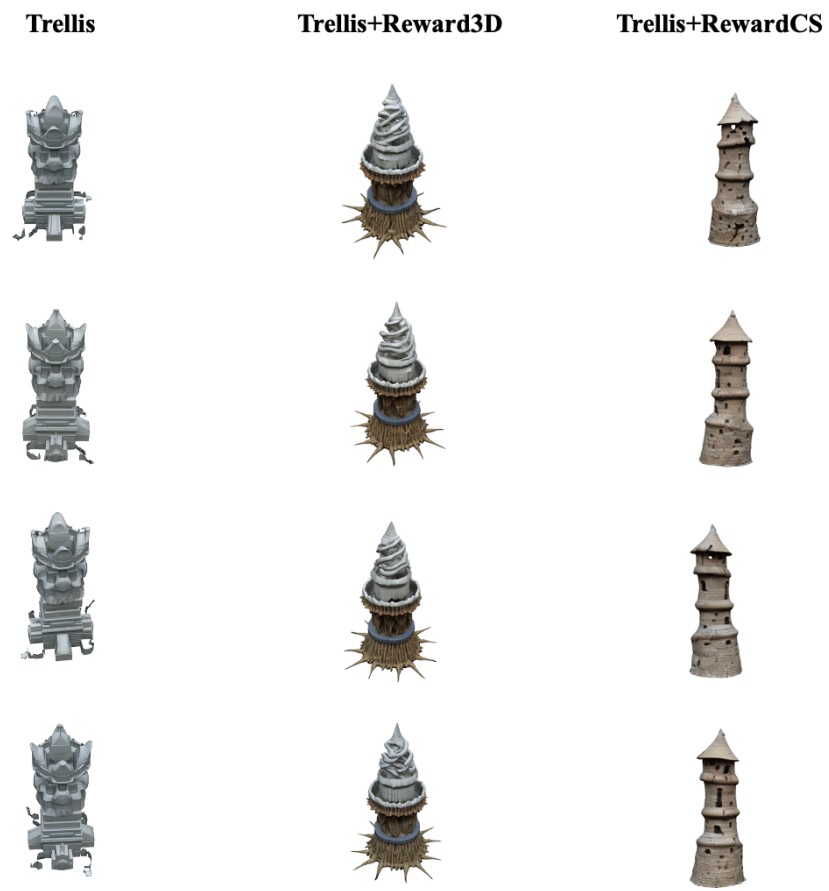

*An ancient, twisted, and unsymmetric tower.*

Figure 32: Qualitative comparisons with the end-to-end generation pipeline: Trellis.

**Trellis**     **Trellis+Reward3D**     **Trellis+RewardCS**

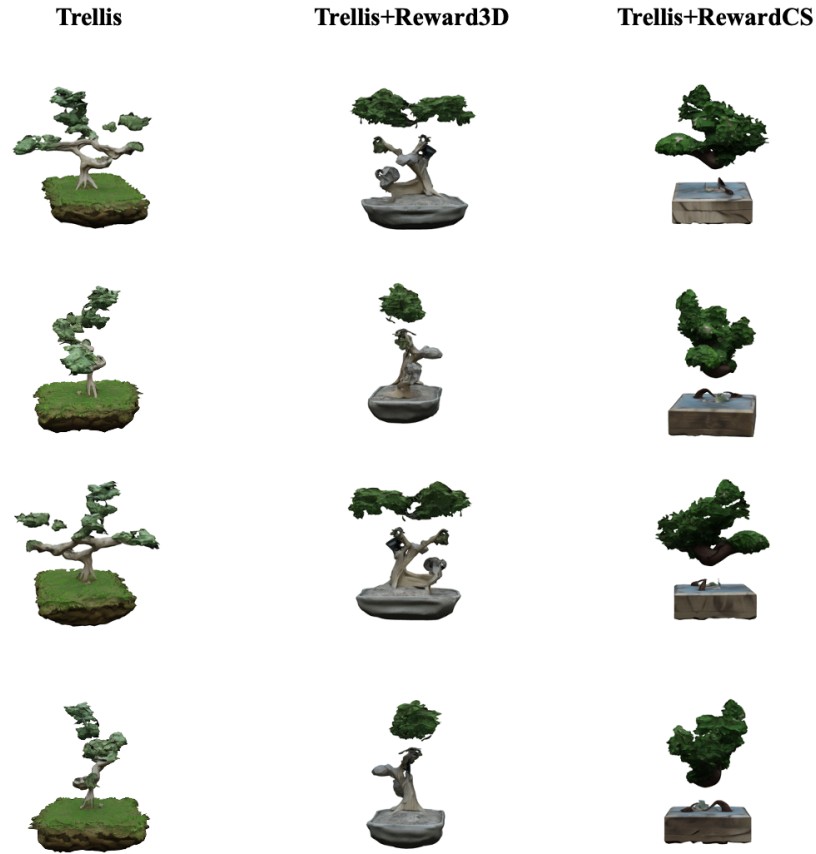

*Floating bonsai tree roots in mid-air.*

Figure 33: Qualitative comparisons with the end-to-end generation pipeline: Trellis.

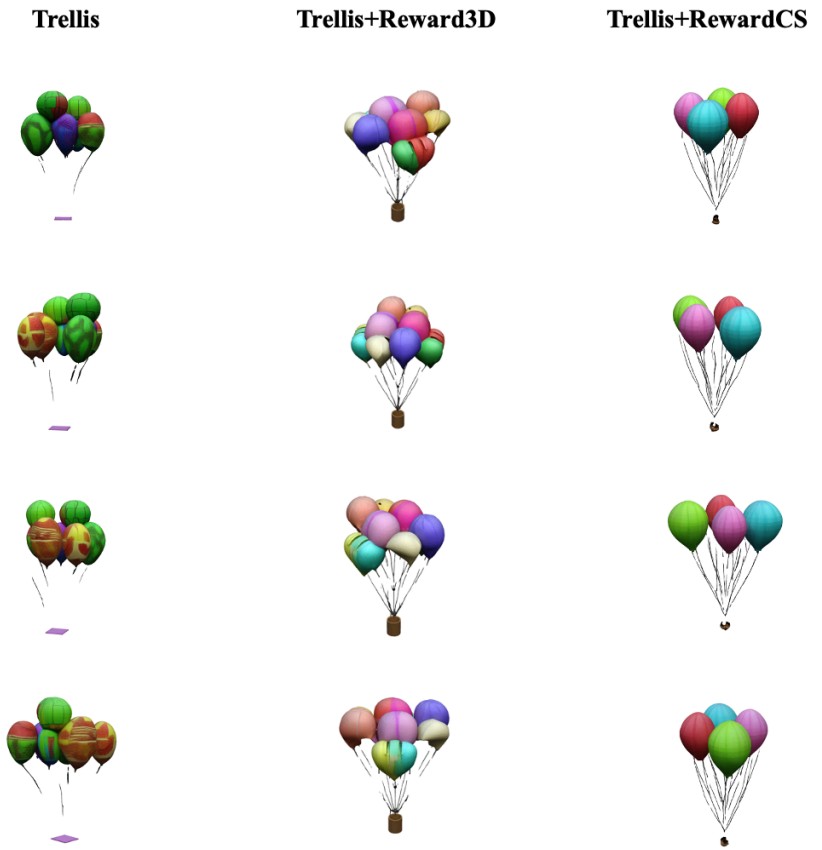

*Four colorful and vibrant balloons tied together.*

Figure 34: Qualitative comparisons with the end-to-end generation pipeline: Trellis.

| Trellis | Trellis+Reward3D | Trellis+RewardCS |
|---|---|---|

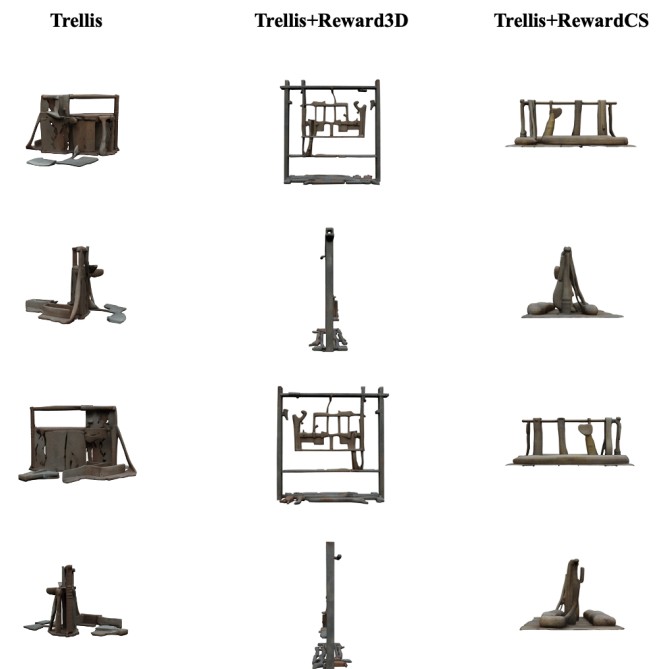

*A collection of solid, irregularly shaped hand tools, with wooden handles and metal ends, well-used and slightly rusty, hanging on a pegboard in a workshop.*

Figure 35: Qualitative comparisons with the end-to-end generation pipeline: Trellis.

These images present multi-view visualizations of text-to-3D results generated by one-stage and two-stage generation pipelines, respectively, across diverse text prompts. We find that text-to-3D vanilla baselines and variants enhanced with 2D guidance methods often produce 3D assets with geometric defects and the Janus problem, while 3D generation backbone with our RewardCS yields geometrically consistent 3D content. In conclusion, DreamCS outperforms text-to-3D vanilla baselines and variants enhanced with 2D guidance methods, 3D controllable generation methods, and advanced text-to-3D generative models by achieving superior geometric consistency and texture fidelity in 3D generation.

## F.4 ADDITIONAL COMPUTATIONAL COST FROM MESHIZATION GUIDANCE.

Regarding the inference cost, we would like to clarify that the computational overhead occurs only in the implicit NeRF-based SDS setting, since our 3D Reward Guidance framework employs DMTet for differentiable mesh extraction during optimization. For advanced mesh based text-to-3D generative pipeline and end-to-end 3D generative backbone, the computational cost incurred by RewardCS is negligible. We include these implicit NeRF-based SDS experiments to demonstrate that DreamCS is a model-agnostic guidance module compatible with general text-to-3D generative backbone.

**Implicit NeRF-based SDS-based Generative Pipeline.** While this introduces some overhead, we emphasize the following:

**a) Significant Quality Gains Justify the Cost.** Despite the added time, meshization-guided reward feedback significantly improves multi-view consistency, reduces geometric artifacts (e.g., floaters, Janus effects), and enhances 3D structural plausibility. These quality improvements are especially crucial in downstream applications like simulation, animation, and fabrication, where geometric plausibility is often more valuable than marginal runtime gains.

**b) Quality-Efficiency Trade-off via Resolution Control.** The meshization process is tunable via the isosurface resolution parameter. In our main experiments, we use a resolution of 128, which we find offers a good balance between efficiency and surface detail. Users can opt for lower resolutions

during fast prototyping or higher resolutions for final asset generation, providing practical flexibility in real-world workflows.

**c) Future Modular Design.** Our framework is modular and fully compatible with advances in mesh extraction techniques. Faster or more efficient meshization algorithms (e.g., learned meshing) can be easily integrated into our pipeline to further reduce runtime without compromising guidance quality.

We conducted the ablation study using MVDream and its RewardCS variants at three different isosurface resolution: 64, 128, and 256. Baseline refers to vanilla MVDream without meshization. The evaluation was performed across 15 diverse prompts from the GPTEval3D benchmark, with each 3D asset generated through optimization over 20000 iterations (it) on a single L40S GPU.

**Trade-off between Generation Speed and 3D Asset Quality.**

Table 10: Performance metrics for different isosurface resolutions.

| Resolution | Generation Speed ($\uparrow$) | IR ($\uparrow$) | VR ($\uparrow$) | GA ($\uparrow$) | T-A ($\uparrow$) | 3DP ($\uparrow$) | G-T ($\uparrow$) |
|---|---|---|---|---|---|---|---|
| Baseline | 0.64 it/s | -0.55 | -3.46 | 2.72 | 2.97 | 3.03 | 2.99 |
| 64 | 0.59 it/s | -0.49 | -2.96 | 2.81 | 3.04 | 3.37 | 3.32 |
| 128 | 0.50 it/s | -0.45 | -2.11 | 3.02 | 3.23 | 4.03 | 4.01 |
| 256 | 0.38 it/s | -0.43 | -1.84 | 3.18 | 3.34 | 4.10 | 4.09 |

Table 11: Computational cost metrics for different isosurface resolutions.

| Method | Wall-clock Time ($\downarrow$) | VRAM ($\downarrow$) |
|---|---|---|
| MVDream | 8.5 hours | 21872 MiB |
| +DreamDPO | 8.9 hours | 25787 MiB |
| +Reward3D | 8.8 hours | 26385 MiB |
| +RewardCS | 11.0 hours | 23784 MiB |

The results in Tab. 10 and 11 illustrate the trade-off between generation speed, computational cost and generated 3D asset quality. We report performance using the same metrics in the main paper, where T-A, 3DP, and G-T are derived from MLLM evaluation. We outline two key observations from these results:

i) Consistent improvement in mesh quality is observed across all metrics as the isosurface resolution increases.

ii) A diminishing return in quality improvement appears at higher isosurface resolution. This indicates that our 128-resolution setting achieves a balance between performance and efficiency.

**Reducing Overhead via 3D Guidance Frequency.** To reduce computational overhead, we conducted additional experiments under the 128-resolution setting, where we varied the RewardCS guidance frequency during 3D asset generation. In particular, we reduced the guidance frequency from the default of $1\text{it}^{-1}$(i.e., applying 3D guidance at every step). We evaluated two configurations with 3D guidance frequency of $\frac{1}{2}\text{it}^{-1}$ and $\frac{1}{3}\text{it}^{-1}$, corresponding to the guidance periods of 2 iterations and 3 iterations, respectively.

Table 12: Performance metrics for different 3D guidance frequencies.

| Frequency | Generation Speed ($\uparrow$) | IR ($\uparrow$) | VR ($\uparrow$) | GA ($\uparrow$) | T-A ($\uparrow$) | 3DP ($\uparrow$) | G-T ($\uparrow$) |
|---|---|---|---|---|---|---|---|
| $\frac{1}{2}\text{it}^{-1}$ | 0.54 it/s | -0.48 | -2.46 | 2.98 | 3.14 | 3.64 | 3.59 |
| $\frac{1}{3}\text{it}^{-1}$ | 0.56 it/s | -0.51 | -2.53 | 2.96 | 3.12 | 3.59 | 3.57 |

As shown in Tab. 12, despite the reduced guidance frequency, both settings still outperform the MVDream baseline across all quality metrics, while also achieving faster generation speed than the standard RewardCS guidance configuration

**Advanced Text-to-3D Generative Pipeline.** For advanced text-to-3D pipelines, where methods optimize an explicit mesh rather than an implicit NeRF field, RewardCS becomes lightweight. When integrated into mesh-based SDS systems such as Magic3D and DreamCraft3D, RewardCS operates directly on explicit geometry and adds $< 1\%$ inference-time cost, making it negligible. In addition, RewardCS can serve as a 3D prior to finetune current end-to-end text-to-3D models such as Trellis, which produces a 3D asset in 3 minutes. As shown in Table 13, RewardCS improves 3D fidelity and geometry metrics without introducing any extra inference cost in guiding DreamCraft3D and TRELLIS under the setting detailed in Appendix F.2, which are better than MVDream.

Table 13: Comparison of different text-to-3D generative backbones.

| Model | CP (↑) | VR (↑) | GA (↑) | T-A (↑) | 3DP (↑) | G-T (↑) | Proportion (↓) | Latency (↓) |
|---|---|---|---|---|---|---|---|---|
| MVDream | 0.23 | -3.30 | 2.77 | 2.95 | 3.10 | 3.07 | 0.55 | 8.5 hours |
| DreamCraft3D | 0.28 | -3.20 | 2.84 | 3.11 | 3.26 | 3.16 | 0.41 | 5.4 hours |
| +Reward3D | 0.29 | -2.96 | 2.90 | 3.43 | 3.51 | 3.45 | 0.35 | 5.5 hours |
| +RewardCS | 0.30 | -2.08 | 3.13 | 3.61 | 4.11 | 4.07 | 0.24 | 5.4 hours |
| TRELLIS | 0.29 | -1.53 | 2.86 | 3.03 | 3.30 | 3.51 | 0.43 | 2.91 mins |
| +Reward3D | 0.28 | -1.47 | 2.90 | 3.10 | 3.35 | 3.58 | 0.40 | 2.91 mins |
| +RewardCS | 0.30 | -1.39 | 3.02 | 3.15 | 3.43 | 3.60 | 0.35 | 2.91 mins |

These results indicate that advanced mesh-based and end-to-end generative backbone not only outperform implicit SDS-based, such as MVDream, in quality but also achieve significantly lower inference latency.

