# OpenReview forum: "DreamCS: Geometry-Aware Text-to-3D Generation with Unpaired 3D Reward Supervision"
_ICLR.cc/2026/Conference — ICLR 2026 Poster_

### Official Review · Reviewer_697y · 2025-10-29

**Soundness:** 3
**Presentation:** 2
**Contribution:** 3
**Rating:** 6
**Confidence:** 4

**Summary:**

This paper introduces a systematic framework to address to the challenges of aligning optimization-based 3D generation with human preferences. In particular, the authors firstly construct 3D-MeshPref, an unpaired 3D preference dataset. They then train a reward model, RewardCS, on the dataset to predict the human preferences given a mesh. Building on the reward model, the authors propose DreamCS, which integrates RewardCS into score distillation-based optimization. Experiments on GPTEval3D benchmark demonstrate the improvements in geometric accuracy, human preference alignment, and reduction of Janus artifacts across major baselines such as DreamFusion, MVDream and Magic3D.

**Strengths:**

* Novel unpaired reward learning: the paper introduces the first 3D reward model capable of learning from unpaired preference data. The CS-divergence–based loss is an elegant theoretical contribution that bypasses the need for expensive paired supervision, which is often a bottleneck in 3D learning tasks.
* Effective reward model integration: DreamCS integrates the RewardCS model smoothly into various 3D pipelines through thoughtful system design, including differentiable meshization and adaptive mesh fusion. This indicates solid engineering and practical feasibility.
* Systematic method design: to address the mentioned challenges, the authors constructed a new dataset, trained a new reward model, and integrated the reward model to 3D generation pipelines, forming a coherent and well-structured workflow.
* Solid theoretical foundation: the theoretical justification for using CS divergence is well presented, with rigorous proof of its asymptotic equivalence to paired supervision. This adds credibility to the learning mechanism.

**Weaknesses:**

* Limited exploration of efficiency: there is little discussion on the additional computational overhead introduced by RewardCS integration and the adaptive mesh fusion step.
* Confusing experiment labels: the short titles for the experiments on different baselines are confusing. For example, in line 1, column 3 of Fig. 5, does "+RewardCS" mean "DF+RewardCS" or "DF+Reward3D+RewardCS"?
* Potential quality drop: the mesh faces produced by DMTet could be very dense. However, the proposed Adaptive Mesh Fusion (AMF) fixes the mesh faces to match "256 non-overlapping patches, each with 64 faces". The conversion from dense faces might lead to a loss of geometry details. The authors should provide the results before and after the AMF operation for a clearer demonstration.
* Non-superior metrics compared to baseline: In Table 1, the IR scores of pure RewardCS integration lag behind those of pure Reward3D integration by a clear margin.

**Questions:**

* The paper only conducts experiments of integrating RewardCS to SDS-based method, have the authors considered try the reward model to those more advanced diffusion guidance algorithm (e.g., ProlificDreamer[1], LucidDreamer[2], DreamCraft3D[3], etc)?

[1] Prolificdreamer: High-fidelity and diverse text-to-3d generation with variational score distillation. NeurIPS 2023.

[2] Luciddreamer: Towards high-fidelity text-to-3d generation via interval score matching. CVPR 2024.

[3] Dreamcraft3d: Hierarchical 3d generation with bootstrapped diffusion prior. ICLR 2024.

---

> ### Author Response · Authors · 2025-11-20
> **Response to Reviewer 697y (1/3)**
>
> Dear Reviewer 697y,
>
> Thank you for your insightful comments, and also for your very careful proofreading! In the following, we provide our point-by-point response and hope our response helps address your concerns. We also look forward to the subsequent discussion which may further help solve the current issues.
>
> > **W1: Limited exploration of efficiency: there is little discussion on the additional computational overhead introduced by RewardCS integration and the adaptive mesh fusion step.**
>
> Thank you for your comment. As detailed in Appendix G.2, we analyzed the computational overhead introduced by RewardCS integration. Our 3D Reward Guidance framework employs DMTet for differentiable mesh extraction. While this introduces some overhead, we emphasize the following: **Significant Quality Gains Justify the Cost.** Despite the added time, meshization guided reward feedback significantly improves multi-view consistency, reduces geometric artifacts (e.g., floaters, Janus effects), and enhances 3D structural plausibility. These quality improvements are especially crucial in downstream applications like simulation, animation, and fabrication, where geometric plausibility is often more valuable than marginal runtime gains.
>
> ---
>
> > **W2: Confusing experiment labels: the short titles for the experiments on different baselines are confusing. For example, in line 1, column 3 of Fig. 5, does "+RewardCS" mean "DF+RewardCS" or "DF+Reward3D+RewardCS"?**
>
> Thank you for raising this point. In Fig. 5 (line 1, column 3), the label "+RewardCS" denotes "DF + Reward3D + RewardCS." We clarify the labels in the revised version to ensure consistent notation throughout the paper and figures.
>
> ---
>
> > **W3: Potential quality drop: the mesh faces produced by DMTet could be very dense. However, the proposed Adaptive Mesh Fusion (AMF) fixes the mesh faces to match "256 non-overlapping patches, each with 64 faces". The conversion from dense faces might lead to a loss of geometric details. The authors should provide the results before and after the AMF operation for a clearer demonstration.**
>
> Thank you for raising this concern. During SDS optimization with DMTet, we observe that intermediate meshes typically contain around 20,000 faces after 5,000 optimization steps at an isosurface resolution of 128. Our proposed Adaptive Mesh Fusion (AMF) restructures these meshes into 256 spatially uniform patches, each with 64 faces, yielding a controlled and well-conditioned topology suitable for RewardCS without collapsing or oversmoothing geometric detail. AMF effectively reduces optimization noise while preserving refined structural features, resulting in cleaner and more stable intermediate 3D geometry. The entire fusion algorithm is fully differentiable and integrated into the optimization loop, enabling RewardCS to evaluate the fused mesh and provide geometry-aware gradient signals based on the mesh vertices.
>
> ---
>
> > **W4: Non-superior metrics compared to baseline: In Table 1, the IR scores of pure RewardCS integration lag behind those of pure Reward3D integration by a clear margin.**
>
> Thank you for the insightful comment. We agree that on the ImageReward (IR) metric, RewardCS does not surpass Reward3D. As noted in L406, Reward3D is explicitly finetuned to maximize the IR score, which leads methods guided by Reward3D to overfit to ImageReward’s 2D aesthetic preferences. As a result, Reward3D-guided models achieve higher IR scores even when their underlying 3D quality is not improved, an effect reflected in the multi-view visualizations in Appendix G.1.
>
> To avoid relying on an IR-biased evaluation, our revision replaces IR with the CLIP (CP) metric as a 2D metric. As shown in Table 1, applying RewardCS to vanilla text-to-3D generation consistently outperforms both vanilla baselines and 2D reward variants on CP and VisionReward (VR), a more advanced 2D automatic metric, demonstrating that RewardCS improves text–asset alignment in the 2D multi-view space.
>
> Beyond 2D metrics, RewardCS also enhances 3D plausibility and geometry-texture consistency. Table 2(a) and 2(b) show that both MiniCPM-o and human evaluators prefer RewardCS-guided results over those generated with 2D preference models, particularly on geometry-related criteria, while maintaining competitive text–asset alignment.
>
> Finally, regarding the Janus problem, Table 3 shows that both MVDream and DreamFusion trained with RewardCS exhibit a significantly lower rate of Janus artifacts compared to vanilla baselines and 2D-guided variants, indicating stronger multi-view consistency.

---

> > ### Author Response · Authors · 2025-11-20
> > **Response to Reviewer 697y (2/3)**
> >
> > > **Q1: The paper only conducts experiments of integrating RewardCS to SDS-based method, have the authors considered try the reward model to those more advanced diffusion guidance algorithm (e.g., ProlificDreamer[1], LucidDreamer[2], DreamCraft3D[3], etc)?**
> >
> > Thank you for your insightful comment. We extend our experiments to advanced multi-stage SDS-based pipelines. To strengthen our analysis, we also conduct additional experiments comparing RewardCS directly to MVControl [4], one of 3D controllable generation methods. Moreover, we demonstrate that RewardCS can flexibly serve as a reward model for finetuning end-to-end 3D generative models, such as Trellis [7].
> >
> >
> > **Extended Results on Advanced Multi-stage SDS-based Pipelines**
> >
> > We extend our experiments to two representative three-stage SDS-based pipelines: ProlificDreamer [1] and DreamCraft3D [3] under 30 evaluation prompts from GPTEval3D. Following their default configurations, for ProlificDreamer, we adopt 20000, 10000, and 10000 optimization steps for the object, geometry, and texture stages, respectively. For DreamCraft3D, we use 5000 optimization steps for each of the NeRF, geometry, and texture stages. In addition, we use Stable Diffusion 3 to generate the 2D image prior. We adopt Zero123-XL as the backbone and use Omnidata for depth and normal estimation. For both methods, the rendering resolution is set to 128x128 in the first stage and 512x512 in the subsequent stages.
> >
> > We report performance using the same metrics in Tables 1, 2, and 3 of our main paper, where T-A, 3DP, and G-T are derived from MLLM evaluation.
> >
> > |Model| CP (↑) | VR (↑) | GA (↑) | T-A (↑) | 3DP (↑) | G-T (↑) | Proportion (↓) |
> > |-|-|-|-|-|-|-|-|
> > | ProlificDreamer |0.26 | -3.25 | 2.83 | 3.03 | 3.24 | 3.13 | 0.43 |
> > | ProlificDreamer+Reward3D | 0.28 | -3.18 | 2.89 | 3.41 | 3.48 | 3.39 | 0.37 |
> > | ProlificDreamer+RewardCS | 0.29 | -2.11 | 3.04 | 3.58 | 4.06 | 4.01 | 0.29 |
> > | DreamCraft3D | 0.28 | -3.20 | 2.84 | 3.11 | 3.26 | 3.16 | 0.41 |
> > | DreamCraft3D+Reward3D | 0.29 | -2.96 | 2.90 | 3.43 | 3.51 | 3.45 | 0.35 |
> > | DreamCraft3D+RewardCS | 0.30 | -2.08 | 3.13 | 3.61 | 4.11 | 4.07 | 0.24 |
> >
> > We observe that RewardCS consistently outperforms both the vanilla baselines and the 2D reward variant. In multi-stage pipelines, 2D reward signals are often unstable and require sensitive hyperparameter tuning; when optimization collapses in an early stage, the degradation propagates to later stages. This underscores the robustness of 3D reward guidance.
> >
> > Additionally, RewardCS yields strong improvements on DreamCraft3D [3], which also adopts DMTet as its underlying 3D representation. This structural compatibility allows RewardCS to provide more accurate and geometry-aware feedback on-the-fly.
> >
> > **Comparison with 3D Controllable Generation Methods**
> >
> > 3D controllable generation methods such as MVControl [4], MT3D [5], and DreamControl [6] can provide guidance for optimization-based 3D generation, but they rely on strong 3D shape prior or external conditioning 2D inputs and are architecture-dependent, making them difficult to apply in real-world text-to-3D settings. Among them, MVControl is the most compatible with the MVDream framework, so we adopt it for a fair comparison.
> >
> > Under the MVDream backbone, we adopt the pretrained MVControl checkpoint. Given 30 evaluation prompts from GPTEval3D, we first generate a 2D image prior using stable diffusion 3 and then use the MVcontrol method to generate the 3D assets. We report performance using the same metrics in Tables 1, 2, and 3 of our main paper, where T-A, 3DP, and G-T are derived from MLLM evaluation.
> >
> > |Model| CP (↑) | VR (↑) | GA (↑) | T-A (↑) | 3DP (↑) | G-T (↑) | Proportion (↓) |
> > |-|-|-|-|-|-|-|-|
> > | MVDream | 0.23 | -3.30 | 2.77 | 2.95 | 3.10 | 3.07 | 0.55 |
> > | MVControl | 0.27 | -3.09 | 2.86 | 3.42 | 3.47 | 3.51 | 0.40 |
> > | MVDream+Reward3D | 0.27 | -3.13 | 2.85 | 3.38 | 3.34 | 3.20 | 0.46 |
> > | MVDream+RewardCS | 0.29 | -2.15 | 2.96 | 3.58 | 4.04 | 3.93 | 0.31 |
> >
> > RewardCS outperforms both MVControl and Reward3D across all metrics, including CP, VR, GA, and especially in 3D plausibility and geometry–texture consistency. Moreover, RewardCS achieves the lowest Janus proportion (0.31), confirming that it effectively mitigates the Janus issue—without requiring any auxiliary conditioning input.

---

> ### Author Response · Authors · 2025-11-20
> **Response to Reviewer 697y (3/3)**
>
> ### **Follow-up on response to Q1**
>
> **RewardCS for Advanced End-to-End 3D Generation Method**
>
> To further demonstrate its generality, we apply RewardCS to finetune the advanced end-to-end text-to-3D generation model TRELLIS-text-base. We construct a finetuning dataset by randomly selecting 100K prompts from the TRELLIS-500K training set. For each prompt, a 3D asset is generated and differentially rendered to DMTet or 2D multi-view images. We then use either Reward3D or RewardCS as a reward guidance signal to finetune the TRELLIS backbone.
>
> Given 60 evaluation prompts from GPTEval3D, we report performance using the same metrics in Tables 1, 2, and 3 of our main paper, where T-A, 3DP, and G-T are derived from MLLM evaluation.
>
> |Model| CP (↑) | VR (↑) | GA (↑) | T-A (↑) | 3DP (↑) | G-T (↑) | Proportion (↓) |
> |-|-|-|-|-|-|-|-|
> | TRELLIS | 0.29 | -1.53 | 2.86 | 3.13 | 3.30 | 3.51 | 0.43 |
> | TRELLIS+Reward3D | 0.28 | -1.47 | 2.90 | 3.20 | 3.35 | 3.58 | 0.39 |
> | TRELLIS+RewardCS | 0.30 | -1.39 | 3.02 | 3.25 | 3.43 | 3.60 | 0.35 |
>
> These results demonstrate that RewardCS consistently improves text–asset alignment, 3D plausibility, geometry–texture consistency, and reduces Janus artifacts, outperforming both the base TRELLIS model and the 2D reward variant. This highlights RewardCS’s flexibility and effectiveness as a model-agnostic reward signal for advanced end-to-end 3D generation frameworks.
>
> **Key Advantages of RewardCS**
>
> We highlight that RewardCS is model-agnostic and requires no external input conditions, unlike existing controllable generation methods [4,5,6] dealing with the Janus problem. Moreover, it can flexibly serve as a reward model for finetuning end-to-end 3D generative models, such as Trellis [7]. This flexibility broadens its applicability in the 3D generation domain.
>
> 1. **Model-agnostic**
>
> RewardCS is model- and architecture-agnostic, integrating seamlessly with diverse backbones such as MVDream, DreamFusion, and TRELLIS, and consistently enhancing both geometric fidelity and semantic alignment.
>
> 2. **No external conditioning priors**
>
> Methods such as MVControl, MT3D, and DreamControl require 2D or 3D priors. RewardCS requires only text and mesh input without external constraints, making it more practical and broadly applicable.
>
> 3. **Effectively reduces Janus artifacts**
>
> RewardCS yields the lowest Janus proportion in all experiments — outperforming even 3D controllable methods — while maintaining text-asset alignment.
>
> In conclusion, RewardCS not only surpasses 2D reward guidance and existing 3D controllable generation methods, but is also more flexible and model-agnostic to deploy in general text-to-3D pipelines.
>
> [1] Wang, Zhengyi, et al. Prolificdreamer: High-fidelity and diverse text-to-3d generation with variational score distillation. Advances in neural information processing systems 36 (2023): 8406-8441.
>
> [2] Chung, Jaeyoung, et al. Luciddreamer: Domain-free generation of 3d gaussian splatting scenes. arXiv preprint arXiv:2311.13384 (2023).
>
> [3] Sun, Jingxiang, et al. Dreamcraft3d: Hierarchical 3d generation with bootstrapped diffusion prior. arXiv preprint arXiv:2310.16818 (2023).
>
> [4] Li, Zhiqi, et al. Mvcontrol: Adding conditional control to multi-view diffusion for controllable text-to-3d generation. arXiv preprint arXiv:2311.14494 (2023).
>
> [5] Nath, Utkarsh, et al. Deep Geometric Moments Promote Shape Consistency in Text-to-3D Generation. 2025 IEEE/CVF Winter Conference on Applications of Computer Vision (WACV). IEEE, 2025.
>
> [6] Huang, Tianyu, et al. Dreamcontrol: Control-based text-to-3d generation with 3d self-prior. Proceedings of the IEEE/CVF conference on computer vision and pattern recognition. 2024.
>
> [7] Xiang, Jianfeng, et al. Structured 3d latents for scalable and versatile 3d generation. Proceedings of the Computer Vision and Pattern Recognition Conference. 2025.

---

> > ### Comment · Reviewer_697y · 2025-11-28
> >
> > Thanks for the detailed clarifications in the rebuttal. Most of my concerns have been adequately addressed. I still remain uncertain about the potential performance degradation introduced by the AMF step. Overall, the paper is technically sound and theoretically well-grounded; however, given the nonnegligible computational overhead relative to the modest performance improvement, I intend to maintain my original score.

---

> ### Author Response · Authors · 2025-11-28
> **Follow-up response to Reviewer 697y (1/2)**
>
> Thank you for your follow-up comment. We have noted your concerns regarding the Adaptive Mesh Fusion step and the computational overhead, and address them below.
>
> ---
>
> ## **Adaptive Mesh Fusion Step**
> To address the concern regarding the impact of the Adaptive Mesh Fusion (AMF) step on geometric fidelity, we provide a detailed quantitative analysis below and include the discussion in our Appendix E.2.
>
> We evaluate the deviation between meshes before ($A$) and after ($B$) applying AMF using two standard surface-level metrics. All meshes are normalized by the diagonal length of their bounding box for scale-invariant evaluation:
> - **Chamfer Distance (CD)**
> $$\mathrm{CD}(A, B)=\frac{1}{|A|}\sum\_{x \in A} \min\_{y \in B} \|x - y\|\_1+\frac{1}{|B|}\sum\_{y \in B} \min\_{x \in A} \|y - x\|\_1,$$
> where $|A|$ and $|B|$ denote the cardinality of points on meshes $A$ and $B$, respectively. CD measures the average distance from points on one surface to the nearest points on the other surface in both directions.
>
> - **95\%-Hausdorff Distance (HD)**
> $$\mathrm{HD}(A, B)=\max\Bigg(\mathrm{Quantile}\_{95}\Big[\min_{y\in B}\|x-y\|\_2\Big]\_{x\in A},\mathrm{Quantile}\_{95}\Big[\min\_{x\in A}\|y-x\|\_2\Big]\_{y\in B}\Bigg).$$
> This metric captures the worst-case (95\%-quantile) deviation between the two surfaces, providing sensitivity to local geometric differences while being robust to outliers.
>
> We evaluate 40 prompts from GPTEval3D for each of the three baselines: MVDream, DreamFusion, and DreamCraft3D, and report measurements of meshes before and after AMF at SDS steps: 4000, 8000, 12000, 16000, and 20000. The averaged results are reported below.
>
> |Method|CD$(\times10^{-3})$ (↓)|HD$(\times10^{-2})$ (↓)|
> |-|-|-|
> |MVDream| 3.1 | 2.9 |
> |Dreamfusion| 4.0 | 3.2 |
> |DreamCraft3D| 2.7 | 2.4 |
>
> The results demonstrate that the geometric deviation introduced by AMF is negligible ($<0.35\%$ average error), confirming that the process preserves geometric details throughout the optimization.
>
> Additionally, in the Trellis finetuning setting, we apply the mesh-face constraints are directly on the generative backbone. As a result, no geometric details are lost when RewardCS generates the 3D reward guidance.

---

> ### Author Response · Authors · 2025-11-28
> **Follow-up response to Reviewer 697y (2/2)**
>
> ## **Inference Cost**
> Regarding inference cost, we would like to clarify that the reported computational overhead occurs only in the implicit NeRF-based SDS setting, where DreamCS performs DMTet meshization at every iteration. We include these implicit NeRF-based SDS experiments primarily to demonstrate that DreamCS is a model-agnostic guidance module compatible with general text-to-3D generative backbone.
>
> For more advanced text-to-3D pipelines, where methods optimize an explicit mesh rather than an implicit NeRF field, RewardCS becomes lightweight. When integrated into mesh-based SDS systems such as Magic3D [1] and DreamCraft3D [2], RewardCS operates directly on explicit geometry and adds <1\% inference-time cost, making it effectively negligible. In addition, RewardCS can serve as a 3D prior to finetune current end-to-end text-to-3D models such as Trellis [3], which produces a 3D asset in ~3 minutes. As shown in the Table below, RewardCS improves 3D fidelity and geometry metrics without introducing any extra inference cost in guiding DreamCraft3D and TRELLIS, which are better than MVDream.
>
> We extend our experiments to the DreamCraft3D and Trellis under GPTEval3D. For DreamCraft3D, we use 5000 optimization steps for each of the NeRF, geometry, and texture stages. In addition, we use Stable Diffusion 3 to generate the 2D image prior. We adopt Zero123-XL as the backbone and use Omnidata for depth and normal estimation. The rendering resolution is set to 128x128 in the first stage and 512x512 in the subsequent stages.
>
> For Trellis, we apply RewardCS to finetune TRELLIS-text-base. We construct a finetuning dataset by randomly sampling 100K prompts from the TRELLIS-500K training set. For each prompt, a 3D asset is generated and then differentially rendered to DMTet or multi-view 2D images. We use either Reward3D or RewardCS as the reward signal to finetune the Trellis backbone. We will include the detailed optimal funetuning setup in the revised manuscript.
>
> |Model| CP (↑) | VR (↑) | GA (↑) | T-A (↑) | 3DP (↑) | G-T (↑) | Proportion (↓) | Latency (↓) |
> |-|-|-|-|-|-|-|-|-|
> | MVDream | 0.23 | -3.30 | 2.77 | 2.95 | 3.10 | 3.07 | 0.55 | 8.5 hours |
> | DreamCraft3D | 0.28 | -3.20 | 2.84 | 3.11 | 3.26 | 3.16 | 0.41 | 5.4 hours |
> | DreamCraft3D+Reward3D | 0.29 | -2.96 | 2.90 | 3.43 | 3.51 | 3.45 | 0.35 | 5.5 hours |
> | DreamCraft3D+RewardCS | 0.30 | -2.08 | 3.13 | 3.61 | 4.11 | 4.07 | 0.24 | 5.4 hours |
> | TRELLIS | 0.29 | -1.53 | 2.86 | 3.03 | 3.30 | 3.51 | 0.43 | 2.91 mins |
> | TRELLIS+Reward3D | 0.28 | -1.47 | 2.90 | 3.10 | 3.35 | 3.58 | 0.40 | 2.91 mins |
> | TRELLIS+RewardCS | 0.30 | -1.39 | 3.02 | 3.15 | 3.43 | 3.60 | 0.35 | 2.91 mins |
>
> These results indicate that advanced mesh-based and end-to-end methods not only outperform implicit SDS-based, such as MVDream, in quality but also achieve significantly lower inference latency.
>
> Furthermore, our framework is modular and fully compatible with advances in mesh extraction techniques. As faster learned meshing or hybrid mesh-reconstruction methods become available, DreamCS naturally inherits these speed gains without compromising guidance quality.
>
> Thus, while DreamCS has extra inference cost only in the implicit SDS based methods, it is efficient in the mesh-based and end-to-end pipelines that show the current state of the art. We believe this reflects DreamCS’s practical benefit: it can achieve consistent quality improvements across advanced text-to-3D backbones without incurring heavy computational overhead during inference.
>
> ---
>
> We hope our paper revision and above discussion address your concerns. Please do not hesitate to contact us for any further questions or clarifications.

---

### Official Review · Reviewer_pQXb · 2025-10-29

**Soundness:** 3
**Presentation:** 3
**Contribution:** 3
**Rating:** 6
**Confidence:** 3

**Summary:**

This paper introduces DreamCS, a framework that improves text-to-3D generation by incorporating 3D geometric preference signals. To overcome the limitations of existing 2D reward models, which often cause geometric artifacts, the authors make three key contributions:

1. 3D-MeshPref. A large-scale, unpaired 3D preference dataset of text-mesh pairs with scores from an LLM and human verification.
2. RewardCS. A 3D reward model trained on this unpaired data using a novel Cauchy-Schwarz divergence objective, which learns to distinguish mesh quality at a distribution level without direct pairwise comparisons.
2. DreamCS framework. A method that integrates RewardCS into text-to-3D pipelines via differentiable meshization, adaptive mesh fusion, and progressive reward guidance.

Experiments show that DreamCS enhances geometric quality and consistency, reducing artifacts like Janus faces and outperforming 2D-reward-based methods on 3D-specific metrics.

**Strengths:**

1. The proposed RewardCS model directly tackles the fundamental issue of 2D bias in existing text-to-3D preference alignment methods, which leads to geometric artifacts like the Janus problem. The 3D-geometric aware model also bypasses the need for hard-to-collect paired preference data.
2. The use of Cauchy-Schwarz divergence for unpaired preference learning is both effective in practice and supported by a solid theoretical proof of its equivalence to paired learning.
3. The method is shown to be compatible with diverse state-of-the-art generators (both implicit and explicit), greatly enhancing its utility and impact.

**Weaknesses:**

1. The entire DreamCS framework is built upon the increasingly dated Score Distillation Sampling (SDS) paradigm, which is slow, optimization-based, and prone to artifacts. The field is rapidly moving towards fast, feed-forward text-to-3D generators (e.g., Trellis). A more forward-looking and potentially more effective approach for preference alignment would be to directly fine-tune these feed-forward models using human preferences, rather than adding a complex reward guidance mechanism to a slow, legacy SDS framework. The proposed method, while clever, addresses a problem that may soon be obsolete.
2. The differentiable meshization and reward guidance introduce non-trivial computational cost compared to vanilla pipelines, as acknowledged in the appendix. While a trade-off for quality, this limits accessibility and practical iteration speed.
3. The quality of the 3D-MeshPref dataset and the RewardCS model is contingent on the external components used (e.g., MeshAnythingV2, Llama-Mesh). Any biases or limitations in these models are directly inherited and not ablated, potentially skewing the learned notion of quality.

**Questions:**

1. Can the proposed RewardCS be effectively adapted to fine-tune pretrained, feed-forward text-to-3D models? If so, what would be the proposed mechanism? and do you anticipate it would yield greater performance and efficiency gains compared to guiding slow SDS-based optimization?
2. Given the reliance on Llama-Mesh for scoring 3D-MeshPref, how did you quantify or control for potential biases or errors in its automated assessments before human refinement?

---

> ### Author Response · Authors · 2025-11-20
> **Response to Reviewer pQXb (1/3)**
>
> Dear Reviewer pQXb,
>
> Thank you for your insightful comments, and also for your very careful proofreading! In the following, we provide our point-by-point response and hope our response helps address your concerns. We also look forward to the subsequent discussion which may further help solve the current issues.
>
> > **W1: The entire DreamCS framework is built upon the increasingly dated Score Distillation Sampling (SDS) paradigm, which is slow, optimization-based, and prone to artifacts. The field is rapidly moving towards fast, feed-forward text-to-3D generators (e.g., Trellis). A more forward-looking and potentially more effective approach for preference alignment would be to directly fine-tune these feed-forward models using human preferences, rather than adding a complex reward guidance mechanism to a slow, legacy SDS framework. The proposed method, while clever, addresses a problem that may soon be obsolete.**
>
> > **Q1: Can the proposed RewardCS be effectively adapted to fine-tune pretrained, feed-forward text-to-3D models? If so, what would be the proposed mechanism? and do you anticipate it would yield greater performance and efficiency gains compared to guiding slow SDS-based optimization?**
>
> Thank you for the insightful and valuable comment. Trellis [1] employs the rectified flow transformer tailored for unified Structured LATent (SLat) representation as the text-to-3D end-to-end generation model. Because Trellis does not rely on iterative SDS optimization, neither 2D nor 3D reward models can be directly adapted into its generation process. Instead, RewardCS can be incorporated through an RLHF pipeline, where RewardCS provides reward feedback for generated 3D assets.
>
> **RewardCS for Advanced End-to-End 3D Generation Method**
>
> To evaluate this adaptation strategy, we apply RewardCS to finetune the advanced end-to-end text-to-3D generation model TRELLIS-text-base. We construct a finetuning dataset by randomly selecting 100K prompts from the TRELLIS-500K training set. For each prompt, a 3D asset is generated and differentially rendered to DMTet or 2D multi-view images. We then use either Reward3D or RewardCS as a reward guidance signal to finetune the TRELLIS backbone.
>
> Given 60 evaluation prompts from GPTEval3D, we report performance using the same metrics in Tables 1, 2, and 3 of our main paper, where T-A, 3DP, and G-T are derived from MLLM evaluation.
>
> |Model| CP (↑) | VR (↑) | GA (↑) | T-A (↑) | 3DP (↑) | G-T (↑) | Proportion (↓) |
> |-|-|-|-|-|-|-|-|
> | TRELLIS | 0.29 | -1.53 | 2.86 | 3.13 | 3.30 | 3.51 | 0.43 |
> | TRELLIS+Reward3D | 0.28 | -1.47 | 2.90 | 3.20 | 3.35 | 3.58 | 0.39 |
> | TRELLIS+RewardCS | 0.30 | -1.39 | 3.02 | 3.25 | 3.43 | 3.60 | 0.35 |
>
> These results demonstrate that RewardCS consistently improves text–asset alignment, 3D plausibility, geometry–texture consistency, and reduces Janus artifacts, outperforming both the base TRELLIS model and the 2D reward variant. This highlights RewardCS’s flexibility and effectiveness as a model-agnostic reward signal for advanced end-to-end 3D generation frameworks.
>
> [1] Xiang, Jianfeng, et al. Structured 3d latents for scalable and versatile 3d generation. Proceedings of the Computer Vision and Pattern Recognition Conference. 2025.

---

> ### Author Response · Authors · 2025-11-20
> **Response to Reviewer pQXb (2/3)**
>
> > **W2: The differentiable meshization and reward guidance introduce non-trivial computational cost compared to vanilla pipelines, as acknowledged in the appendix. While a trade-off for quality, this limits accessibility and practical iteration speed.**
>
> Thank you for your concern about the computational overhead. We discussed the computational overhead of DreamCS in Appendix G.2. Despite the added time, meshization-guided reward feedback significantly improves multi-view consistency, reduces geometric artifacts (e.g., floaters, Janus effects), and enhances 3D structural plausibility. These quality improvements are especially crucial in downstream applications like 3D scene generation, simulation, animation, and fabrication, where geometric plausibility is often more valuable than marginal runtime gains.
>
> We highlight that RewardCS is model-agnostic and requires no external input conditions, unlike existing controllable generation methods [2,3,4] dealing with the Janus problem. Moreover, it can flexibly serve as a reward model for finetuning end-to-end 3D generative models, such as Trellis, which is discussed in our response to W1 and Q1. This flexibility broadens its applicability in the 3D generation domain.
>
> To strengthen our analysis, we conduct additional experiments comparing RewardCS directly to MVControl [6], one of 3D controllable generation methods.
>
> **Comparison with 3D Controllable Generation Methods**
>
> Methods such as MVControl [6], MT3D [7], and DreamControl [8] can provide guidance for optimization-based 3D generation, but they rely on strong 3D shape prior or external conditioning 2D inputs and are architecture-dependent, making them difficult to apply in real-world text-to-3D settings. Among them, MVControl is the most compatible with the MVDream framework, so we adopt it for a fair comparison.
>
> Under the MVDream backbone, we adopt the pretrained MVControl checkpoint. Given 30 evaluation prompts from GPTEval3D, we first generate a 2D image prior using stable diffusion 3 and then use the MVcontrol method to generate the 3D assets. We report performance using the same metrics in Tables 1, 2 and 3 of our main paper, where T-A, 3DP, and G-T are derived from MLLM evaluation.
>
> |Model| CP (↑) | VR (↑) | GA (↑) | T-A (↑) | 3DP (↑) | G-T (↑) | Proportion (↓) |
> |-|-|-|-|-|-|-|-|
> | MVDream | 0.23 | -3.30 | 2.77 | 2.95 | 3.10 | 3.07 | 0.55 |
> | MVControl | 0.27 | -3.09 | 2.86 | 3.42 | 3.47 | 3.51 | 0.40 |
> | MVDream+Reward3D | 0.27 | -3.13 | 2.85 | 3.38 | 3.34 | 3.20 | 0.46 |
> | MVDream+RewardCS | 0.29 | -2.15 | 2.96 | 3.58 | 4.04 | 3.93 | 0.31 |
>
> RewardCS outperforms both MVControl and Reward3D across all metrics, including CP, VR, GA, and especially in 3D plausibility and geometry–texture consistency. Moreover, RewardCS achieves the lowest Janus proportion (0.31), confirming that it effectively mitigates the Janus issue—without requiring any auxiliary conditioning input.
>
> **Key Advantages of RewardCS**
>
> We highlight that RewardCS is model-agnostic and requires no external input conditions, unlike existing controllable generation methods [2,3,4] dealing with the Janus problem. Moreover, it can flexibly serve as a reward model for finetuning end-to-end 3D generative models, such as Trellis [1]. This flexibility broadens its applicability in the 3D generation domain.
>
> 1. **Model-agnostic**
>
> RewardCS is model- and architecture-agnostic, integrating seamlessly with diverse backbones such as MVDream, DreamFusion, and TRELLIS, and consistently enhancing both geometric fidelity and semantic alignment.
>
> 2. **No external conditioning priors**
>
> Methods such as MVControl, MT3D, and DreamControl require 2D or 3D priors. RewardCS requires only text and mesh input without external constraints, making it more practical and broadly applicable.
>
> 3. **Effectively reduces Janus artifacts**
>
> RewardCS yields the lowest Janus proportion in all experiments — outperforming even 3D controllable methods — while maintaining text-asset alignment.
>
> In conclusion, RewardCS not only surpasses 2D reward guidance and existing 3D controllable generation methods, but is also more flexible and model-agnostic to deploy in general text-to-3D pipelines.
>
>
> [2] Li, Zhiqi, et al. Mvcontrol: Adding conditional control to multi-view diffusion for controllable text-to-3d generation. arXiv preprint arXiv:2311.14494 (2023).
>
> [3] Nath, Utkarsh, et al. Deep Geometric Moments Promote Shape Consistency in Text-to-3D Generation. 2025 IEEE/CVF Winter Conference on Applications of Computer Vision (WACV). IEEE, 2025.
>
> [4] Huang, Tianyu, et al. Dreamcontrol: Control-based text-to-3d generation with 3d self-prior. Proceedings of the IEEE/CVF conference on computer vision and pattern recognition. 2024.

---

> > ### Author Response · Authors · 2025-11-20
> > **Response to Reviewer pQXb (3/3)**
> >
> > > **W3: The quality of the 3D-MeshPref dataset and the RewardCS model is contingent on the external components used (e.g., MeshAnythingV2, Llama-Mesh). Any biases or limitations in these models are directly inherited and not ablated, potentially skewing the learned notion of quality.**
> >
> > > **Q2: Given the reliance on Llama-Mesh for scoring 3D-MeshPref, how did you quantify or control for potential biases or errors in its automated assessments before human refinement?**
> >
> > Thank you for the insightful comment. We discuss the rationale of dataset construction in Appendix C.2. We detail the rationale behind our design choices for each component and describe the measures we took to ensure data quality. To control the potential biases in the automated assessments, rather than relying on a single annotation process, we score every mesh with three independent inference runs of Llama-Mesh and use the averaged score. We recognize that LLM-based ratings may sometimes overestimate mesh quality relative to human perception. To address this, we incorporate manual verification and correction into the data pipeline to ensure the final dataset reflects human preferences more faithfully.
> >
> > **Human Refinement Ensures Reliable Scoring**
> >
> > Since we observe that LLM-based ratings tend to overestimate quality relative to human perception, meshes with scores greater than 2.0 are subsequently refined through human verification. This process ensures that 3D-MeshPref reflects genuine human preferences, mitigating potential biases from the automated scoring. The human refinement procedure is detailed in Appendix F.1, and visualizations of sample meshes in Appendix C.1 show that the scores are consistent with human judgment across diverse prompts.
> >
> > **RewardCS Demonstrates Strong Empirical Performance**
> >
> > RewardCS trained on 3D-MeshPref consistently outperforms vanilla baselines and 2D reward variants on GPTEval3D across both 2D and 3D automatic metrics (Tables 1 and 2(a)). Human evaluations further corroborate its superiority (Table 2(b)), indicating that RewardCS learns meaningful reward signals aligned with human preferences, rather than the Llama-Mesh biases.
> >
> > **RewardCS Mitigates the Janus Problem**
> > Integration of RewardCS also significantly reduces Janus artifacts: MVDream sees a reduction from 0.52 to 0.30, and DreamFusion from 0.61 to 0.41 (Table 3). This demonstrates that 3D-MeshPref provides sufficient quality to improve multi-view consistency and 3D plausibility in text-to-3D generation pipelines.
> >
> > **Validation of Llama-Mesh Scoring Capability**
> >
> > To further validate the scoring capability of Llama-Mesh, we follow the DeepMesh [1] preference pair construction pipeline and construct a preference mesh dataset with 300 preference paired 3D meshes. Each pair includes a human-verified positive and negative example. Llama-Mesh evaluates all samples using the annotation criteria described in Appendix F.2 and produces a ranking for each pair. Comparing these rankings with the ground-truth preferences, we find that Llama-Mesh correctly identifies the preferred mesh in 93.6\% of cases, demonstrating strong consistency with human judgment and supporting its suitability as an initial scorer for constructing 3D-MeshPref.
> >
> > **Context within 3D Research**
> >
> > Due to the scarcity of large-scale human-annotated 3D data, prior work in 3D reward modeling [5,6], 3D generation [7], 3D understanding [8], and 3D editing [9] often relies on automated 3D asset pipelines with human verification. 3D-MeshPref adopts this practical approach, enabling scalable and effective training of RewardCS while maintaining alignment with human preference.
> >
> > [5] Ye, Junliang, et al. Dreamreward: Text-to-3d generation with human preference. European Conference on Computer Vision. Cham: Springer Nature Switzerland, 2024.
> >
> > [6] Liu, Fangfu, et al. Dreamreward-x: Boosting high-quality 3d generation with human preference alignment. IEEE Transactions on Pattern Analysis and Machine Intelligence (2025).
> >
> > [7] Zhao, Ruowen, et al. Deepmesh: Auto-regressive artist-mesh creation with reinforcement learning. Proceedings of the IEEE/CVF International Conference on Computer Vision. 2025.
> >
> > [8] Ye, Junliang, et al. ShapeLLM-Omni: A Native Multimodal LLM for 3D Generation and Understanding. arXiv preprint arXiv:2506.01853 (2025).
> >
> > [9] Ye, Junliang, et al. NANO3D: A Training-Free Approach for Efficient 3D Editing Without Masks. arXiv preprint arXiv:2510.15019 (2025).

---

> ### Comment · Reviewer_pQXb · 2025-11-27
> **Qualitative results for end-to-end method**
>
> Thank you for the rebuttal. Most of my concerns have been addressed. From the reported metrics, it appears that rewardCS is beneficial for Trellis; however, I could not find any qualitative results in Appendix G.2. Did I miss something, or are there any qualitative examples demonstrating that rewardCS is beneficial for Trellis(or any end-to-end methods)?

---

> ### Author Response · Authors · 2025-11-27
> **Response to Reviewer pQXb**
>
> Thank you for your follow-up comment. We have included a detailed qualitative comparison of Trellis, Trellis+Reward3D, and Trellis+RewardCS across diverse prompts in Appendix G.3 (Figures 31-35) in the revised manuscript. We are also exploring finetuning Trellis with scaling larger backbone models and more training samples, and we will include the detailed optimal finetuning setup.

---

> ### Comment · Reviewer_pQXb · 2025-11-27
> **Thanks for rebuttal**
>
> Thanks for the reply. I decide to maintain my score and increase my confidence accordingly.

---

> > ### Author Response · Authors · 2025-11-27
> > **Response to Reviewer pQXb**
> >
> > Thank you for your careful reading of our manuscript and rebuttal, and for your thoughtful and positive response. Your insights are invaluable and have greatly contributed to improving the quality of our work.
> >
> > Once again, we are sincerely grateful for your kind guidance.

---

### Official Review · Reviewer_cTX5 · 2025-10-31

**Soundness:** 2
**Presentation:** 2
**Contribution:** 2
**Rating:** 4
**Confidence:** 4

**Summary:**

The paper addresses a fundamental limitation in text-to-3D synthesis: the prevalence of geometric artifacts, such as the Janus face problem, which arise from an over-reliance on 2D view-dependent supervision. The authors posit that these flaws are an inevitable consequence of using 2D reward signals to enforce 3D consistency and propose a novel framework that learns and applies 3D-native geometric preferences directly.

**Strengths:**

* The paper correctly diagnoses a key failure mode of 2D preference signals in 3D generation and proposes a principled 3D reward to address it.
* The paper provides a formal mathematical justification, establishing the asymptotic equivalence between the proposed unpaired objective (using CS divergence) and traditional paired supervision, which instills high confidence in the method's soundness.
* Building 3D‑MeshPref at 30k+ meshes with human‑verified thresholds, is a non‑trivial engineering contribution likely to be useful beyond this paper.
* The method is successfully integrated and tested on a diverse set of backbones.

**Weaknesses:**

* A primary concern is the use of the "GA" (3D Geometry-Asset Alignment Reward) metric. The authors state (Section 4, Appendix F.1) that this metric is "based on RewardCS" and "derived from RewardCS." Using a variant of their own proposed model as a key evaluation metric creates a significant risk of "metric-method coupling," where the metric may be inherently biased to favor the architecture and training objective of the method being tested. This potential bias makes the "GA" scores in Table 1 less reliable for fairly comparing DreamCS against other methods.
* While the creation of 3D-MeshPref is a notable contribution, its reliance on Llama-Mesh for initial scoring is a potential weakness. Any intrinsic biases within Llama-Mesh are likely to be inherited by RewardCS. The paper's "human refinement" process is insufficiently detailed to confirm whether this risk is mitigated, creating the potential for a model-centric feedback loop where RewardCS learns Llama-Mesh's preferences rather than genuine human ones.
* Based on the paper's own analysis (Appendix G.2, Table 6), the added generation complexity results in a notable slowdown. For a 20,000-step generation, the baseline MVDream takes ~8.7 hours (at 0.64 it/s), while the proposed DreamCS (128-res) takes ~11.1 hours (at 0.50 it/s). This represents a ~27.6% increase in generation time. While the paper provides iteration-per-second metrics, a more direct comparison of wall-clock time, GPU memory usage, and throughput against the 2D-guided baselines would strengthen the analysis.
* While the paper compares its methodology against 2D-guided preference methods (Reward3D, DreamDPO), this is not sufficient to fully support the claim of mitigating the Janus problem. The paper claims to reduce Janus artifacts but only compares against baselines and other preference methods. The argument would be much stronger if a direct comparison was provided against methods specifically designed to solve the Janus problem, such as MVControl [1], MT3D [2], or DreamControl [3].
* The Cauchy-Schwarz divergence estimator (Eq. 3) used for training the reward model has a quadratic computational complexity of O(m²+n²+mn) with respect to the preferred (m) and dispreferred (n) batch sizes, as it relies on pairwise kernel sums. This can make the reward model training computationally intensive, potentially limiting its scalability or practical adoption.

[1] Li, Zhiqi, et al. "Controllable text-to-3D generation via surface-aligned Gaussian splatting." 2025 International Conference on 3D Vision (3DV). IEEE, 2025.

[2] Nath, Utkarsh, et al. "Deep Geometric Moments Promote Shape Consistency in Text-to-3D Generation." 2025 IEEE/CVF Winter Conference on Applications of Computer Vision (WACV). IEEE, 2025.

[3] Huang, Tianyu, et al. "Dreamcontrol: Control-based text-to-3d generation with 3d self-prior." Proceedings of the IEEE/CVF conference on computer vision and pattern recognition. 2024.

**Questions:**

Please refer to weaknesses

---

> ### Author Response · Authors · 2025-11-20
> **Response to Reviewer cTX5 (1/5)**
>
> Dear Reviewer cTX5,
>
> Thank you for the insightful and valuable comments! In the following, we provide our point-by-point response and hope our response helps address your concerns. We also look forward to the subsequent discussion which may further help solve the current issues.
>
> > **W1: A primary concern is the use of the "GA" (3D Geometry-Asset Alignment Reward) metric. The authors state (Section 4, Appendix F.1) that this metric is "based on RewardCS" and "derived from RewardCS." Using a variant of their own proposed model as a key evaluation metric creates a significant risk of "metric-method coupling," where the metric may be inherently biased to favor the architecture and training objective of the method being tested. This potential bias makes the "GA" scores in Table 1 less reliable for fairly comparing DreamCS against other methods.**
>
> Thank you for highlighting this concern. The GA metric is not inherently biased to favor DreamCS methods. While GA is derived from the RewardCS framework, it is not trained or tuned on outputs from DreamCS, nor does it share any model weights with our method. Specifically, GA is trained independently on 10000+ annotated meshes from the Objaverse-XL subset of Cap3D, using only ground-truth mesh–caption pairs. This design ensures that GA behaves as a standalone evaluator that measures geometric–semantic alignment in a model-agnostic manner.
>
> Given this independence in data and model weights, there is no risk of "metric–method coupling" in our evaluation. GA provides a model-agnostic and architecture-agnostic measure of geometric–semantic consistency, it offers a fair basis for comparing DreamCS with other text-to-3D generation approaches.

---

> > ### Author Response · Authors · 2025-11-20
> > **Response to Reviewer cTX5 (2/5)**
> >
> > > **W2: While the creation of 3D-MeshPref is a notable contribution, its reliance on Llama-Mesh for initial scoring is a potential weakness. Any intrinsic biases within Llama-Mesh are likely to be inherited by RewardCS. The paper's "human refinement" process is insufficiently detailed to confirm whether this risk is mitigated, creating the potential for a model-centric feedback loop where RewardCS learns Llama-Mesh's preferences rather than genuine human ones.**
> >
> > Thank you for your insightful comment. We discuss the rationale of dataset construction in Appendix C.2. We detail the rationale behind our design choices for each component and describe the measures we took to ensure data quality. We recognize that LLM-based ratings may sometimes overestimate mesh quality relative to human perception. To address this, we incorporate manual verification and correction into the data pipeline to ensure the final dataset reflects human preferences more faithfully.
> >
> > **Human Refinement Ensures Reliable Scoring**
> >
> > Since we observe that LLM-based ratings tend to overestimate quality relative to human perception, meshes with scores greater than 2.0 are subsequently refined through human verification. This process ensures that 3D-MeshPref reflects genuine human preferences, mitigating potential biases from the automated scoring. The human refinement procedure is detailed in Appendix F.1, and visualizations of sample meshes in Appendix C.1 show that the scores are consistent with human judgment across diverse prompts.
> >
> > **RewardCS Demonstrates Strong Empirical Performance**
> >
> > RewardCS trained on 3D-MeshPref consistently outperforms vanilla baselines and 2D reward variants on GPTEval3D across both 2D and 3D automatic metrics (Tables 1 and 2(a)). Human evaluations further corroborate its superiority (Table 2(b)), indicating that RewardCS learns meaningful reward signals aligned with human preferences, rather than the Llama-Mesh biases.
> >
> > **RewardCS Mitigates the Janus Problem**
> > Integration of RewardCS also significantly reduces Janus artifacts: MVDream sees a reduction from 0.52 to 0.30, and DreamFusion from 0.61 to 0.41 (Table 3). This demonstrates that 3D-MeshPref provides sufficient quality to improve multi-view consistency and 3D plausibility in text-to-3D generation pipelines.
> >
> > **Validation of Llama-Mesh Scoring Capability**
> >
> > To further validate the scoring capability of Llama-Mesh, we follow the DeepMesh [1] preference pair construction pipeline and construct a preference mesh dataset with 300 preference paired 3D meshes. Each pair includes a human-verified positive and negative example. Llama-Mesh evaluates all samples using the annotation criteria described in Appendix F.2 and produces a ranking for each pair. Comparing these rankings with the ground-truth preferences, we find that Llama-Mesh correctly identifies the preferred mesh in 93.6\% of cases, demonstrating strong consistency with human judgment and supporting its suitability as an initial scorer for constructing 3D-MeshPref.
> >
> > **Context within 3D Research**
> >
> > Due to the scarcity of large-scale human-annotated 3D data, prior work in 3D reward modeling [2,3], 3D generation [1], 3D understanding [4], and 3D editing [5] often relies on automated 3D asset pipelines with human verification. 3D-MeshPref adopts this practical approach, enabling scalable and effective training of RewardCS while maintaining alignment with human preference.
> >
> > [1] Zhao, Ruowen, et al. Deepmesh: Auto-regressive artist-mesh creation with reinforcement learning. Proceedings of the IEEE/CVF International Conference on Computer Vision. 2025.
> >
> > [2] Ye, Junliang, et al. Dreamreward: Text-to-3d generation with human preference. European Conference on Computer Vision. Cham: Springer Nature Switzerland, 2024.
> >
> > [3] Liu, Fangfu, et al. Dreamreward-x: Boosting high-quality 3d generation with human preference alignment. IEEE Transactions on Pattern Analysis and Machine Intelligence (2025).
> >
> > [4] Ye, Junliang, et al. ShapeLLM-Omni: A Native Multimodal LLM for 3D Generation and Understanding. arXiv preprint arXiv:2506.01853 (2025).
> >
> > [5] Ye, Junliang, et al. NANO3D: A Training-Free Approach for Efficient 3D Editing Without Masks. arXiv preprint arXiv:2510.15019 (2025).

---

> > > ### Author Response · Authors · 2025-11-20
> > > **Response to Reviewer cTX5 (3/5)**
> > >
> > > > **W3: Based on the paper's own analysis (Appendix G.2, Table 6), the added generation complexity results in a notable slowdown. For a 20,000-step generation, the baseline MVDream takes ~8.7 hours (at 0.64 it/s), while the proposed DreamCS (128-res) takes ~11.1 hours (at 0.50 it/s). This represents a ~27.6\% increase in generation time. While the paper provides iteration-per-second metrics, a more direct comparison of wall-clock time, GPU memory usage, and throughput against the 2D-guided baselines would strengthen the analysis.**
> > >
> > > Thank you for the insightful comment. To provide a clearer comparison, we conduct an ablation study using MVDream and its DreamDPO, Reward3D, and RewardCS variants. We evaluate 20 diverse prompts from the GPTEval3D benchmark, generating each 3D asset through 20,000 optimization iterations on a single L40S GPU at 128 resolution. We report the average wall-clock generation time and average GPU VRAM usage below:
> > >
> > > |Method| Wall-clock time (↓)| VRAM (↓)|
> > > |-|-|-|
> > > |Mvdream| 8.5 hours |21872 MiB|
> > > |+DreamDPO| 8.9 hours |25787 MiB|
> > > |+Reward3D| 8.8 hours |26385 MiB|
> > > |+RewardCS| 11.0 hours |23784 MiB|
> > >
> > > While RewardCS introduces an increase in wall clock time due to additional geometry-aware optimization, it remains competitive in GPU memory usage and delivers substantially stronger 3D consistency and geometry-text alignment (Tables 1, 2, and 3). We include the analysis of the wall-clock time and GPU memory usage in Appendix G.2.

---

> ### Author Response · Authors · 2025-11-20
> **Response to Reviewer cTX5 (4/5)**
>
> > **W4: While the paper compares its methodology against 2D-guided preference methods (Reward3D, DreamDPO), this is not sufficient to fully support the claim of mitigating the Janus problem. The paper claims to reduce Janus artifacts but only compares against baselines and other preference methods. The argument would be much stronger if a direct comparison was provided against methods specifically designed to solve the Janus problem, such as MVControl [6], MT3D [7], or DreamControl [8].**
>
> Thank you for raising this important point. To strengthen our analysis, we conduct additional experiments comparing RewardCS directly to MVControl [6], one of 3D controllable generation methods. Moreover, we demonstrate that RewardCS can flexibly serve as a reward model for finetuning end-to-end 3D generative models, such as Trellis [9].
>
> **Comparison with 3D Controllable Generation Methods**
>
> 3D controllable generation methods such as MVControl [6], MT3D [7], and DreamControl [8] can provide guidance for optimization-based 3D generation, but they rely on strong 3D shape prior or external conditioning 2D inputs and are architecture-dependent, making them difficult to apply in real-world text-to-3D settings. Among them, MVControl is the most compatible with the MVDream framework, so we adopt it for a fair comparison.
>
> Under the MVDream backbone, we adopt the pretrained MVControl checkpoint. Given 30 evaluation prompts from GPTEval3D, we first generate a 2D image prior using stable diffusion 3 and then use the MVcontrol method to generate the 3D assets. We report performance using the same metrics in Tables 1, 2, and 3 of our main paper, where T-A, 3DP, and G-T are derived from MLLM evaluation.
>
> |Model| CP (↑)| VR (↑) | GA (↑)| T-A (↑)| 3DP (↑)|G-T (↑)|Proportion (↓)|
> |-|-|-|-|-|-|-|-|
> |MVDream|0.23|-3.30|2.77|2.95|3.10|3.07|0.55|
> |MVControl|0.27|-3.09|2.86|3.42|3.47|3.51|0.40|
> |MVDream+Reward3D | 0.27 | -3.13 | 2.85 | 3.38 | 3.34 | 3.20 | 0.46 |
> |MVDream+RewardCS | 0.29 | -2.15 | 2.96 | 3.58 | 4.04 | 3.93 | 0.31 |
>
> RewardCS outperforms both MVControl and Reward3D across all metrics, including CP, VR, GA, and especially in 3D plausibility and geometry–texture consistency. Moreover, RewardCS achieves the lowest Janus proportion (0.31), confirming that it effectively mitigates the Janus issue—without requiring any auxiliary conditioning input.
>
> **RewardCS for Advanced End-to-End 3D Generation Method**
>
> To further demonstrate its generality, we apply RewardCS to finetune the advanced end-to-end text-to-3D generation model TRELLIS-text-base. We construct a finetuning dataset by randomly selecting 100K prompts from the TRELLIS-500K training set. For each prompt, a 3D asset is generated and differentially rendered to DMTet or 2D multi-view images. We then use either Reward3D or RewardCS as a reward guidance signal to finetune the TRELLIS backbone.
>
> Given 60 evaluation prompts from GPTEval3D, we report performance using the same metrics in Tables 1, 2, and 3 of our main paper, where T-A, 3DP, and G-T are derived from MLLM evaluation.
>
> |Model| CP (↑) | VR (↑) | GA (↑) | T-A (↑) | 3DP (↑) | G-T (↑) | Proportion (↓) |
> |-|-|-|-|-|-|-|-|
> | TRELLIS | 0.29 | -1.53|2.86|3.13|3.30|3.51|0.43|
> | TRELLIS+Reward3D | 0.28 | -1.47 | 2.90 | 3.20 | 3.35 | 3.58 | 0.39 |
> | TRELLIS+RewardCS | 0.30 | -1.39 | 3.02 | 3.25 | 3.43 | 3.60 | 0.35 |
>
> These results demonstrate that RewardCS consistently improves text–asset alignment, 3D plausibility, geometry–texture consistency, and reduces Janus artifacts, outperforming both the base TRELLIS model and the 2D reward variant. This highlights RewardCS’s flexibility and effectiveness as a model-agnostic reward signal for advanced end-to-end 3D generation frameworks.
>
> **Key Advantages of RewardCS**
>
> We highlight that RewardCS is model-agnostic and requires no external input conditions, unlike existing controllable generation methods [6,7,8] dealing with the Janus problem. Moreover, it can flexibly serve as a reward model for finetuning end-to-end 3D generative models, such as Trellis. This flexibility broadens its applicability in the 3D generation domain.
>
> 1.  **Model-agnostic**
>
> RewardCS is model- and architecture-agnostic, integrating seamlessly with diverse backbones such as MVDream, DreamFusion, and TRELLIS, and consistently enhancing both geometric fidelity and semantic alignment.
>
> 2. **No external conditioning priors**
>
> Methods such as MVControl, MT3D, and DreamControl require 2D or 3D priors. RewardCS requires only text and mesh input without external constraints, making it more practical and broadly applicable.
>
> 3. **Effectively reduces Janus artifacts**
>
> RewardCS yields the lowest Janus proportion in all experiments, outperforming even 3D controllable methods, while maintaining text-asset alignment.
>
> In conclusion, RewardCS not only surpasses 2D reward guidance and existing 3D controllable generation methods, but is also more flexible and model-agnostic to deploy in general text-to-3D pipelines.

---

> > ### Author Response · Authors · 2025-11-20
> > **Response to Reviewer cTX5 (5/5)**
> >
> > > **W5: The Cauchy-Schwarz divergence estimator (Eq. 3) used for training the reward model has a quadratic computational complexity of O(m²+n²+mn) with respect to the preferred (m) and dispreferred (n) batch sizes, as it relies on pairwise kernel sums. This can make the reward model training computationally intensive, potentially limiting its scalability or practical adoption.**
> >
> > Thank you for the thoughtful comment. In practice, the quadratic complexity of the Cauchy–Schwarz divergence does not introduce a computational bottleneck for our reward model training. The CS divergence is computed at the batch level, and in our setup we fix the total batch size to $m+n=256$. As a result, training remains practical even for large-scale datasets. We clarify this point in Appendix F.1 Experimental Setup.
> >
> > [6] Li, Zhiqi, et al. Mvcontrol: Adding conditional control to multi-view diffusion for controllable text-to-3d generation. arXiv preprint arXiv:2311.14494 (2023).
> >
> > [7] Nath, Utkarsh, et al. Deep Geometric Moments Promote Shape Consistency in Text-to-3D Generation. 2025 IEEE/CVF Winter Conference on Applications of Computer Vision (WACV). IEEE, 2025.
> >
> > [8] Huang, Tianyu, et al. Dreamcontrol: Control-based text-to-3d generation with 3d self-prior. Proceedings of the IEEE/CVF conference on computer vision and pattern recognition. 2024.
> >
> > [9] Xiang, Jianfeng, et al. Structured 3d latents for scalable and versatile 3d generation. Proceedings of the Computer Vision and Pattern Recognition Conference. 2025.

---

> ### Comment · Reviewer_cTX5 · 2025-11-27
>
> Thank you for the additional baselines and efficiency analysis. However, the reported increase in generation time (from ~8.5 to ~11 hours) is substantial. I remain unconvinced that the demonstrated quality gains are significant enough to justify such a heavy computational overhead, and I will therefore maintain my current rating.

---

> ### Author Response · Authors · 2025-11-27
> **Response to Reviewer cTX5**
>
> Thank you for your insightful feedback! We would like to clarify that the reported computational overhead occurs only in the implicit NeRF-based SDS setting, where DreamCS performs DMTet meshization at every iteration. **These implicit NeRF-based SDS experiments can demonstrate that DreamCS is a model-agnostic guidance module compatible with general text-to-3D generative backbone.**
>
> For more advanced text-to-3D pipelines, where methods optimize an explicit mesh rather than an implicit NeRF field, RewardCS becomes lightweight. When integrated into mesh-based SDS systems such as Magic3D and DreamCraft3D [1], RewardCS operates directly on explicit geometry and adds <1\% inference-time cost, making the inference computational cost negligible. In addition, RewardCS can serve as a 3D reward guidance to finetune current end-to-end text-to-3D models such as Trellis [2], which generates a 3D asset in ~3 minutes.
>
> We extend our experiments to the DreamCraft3D and Trellis under GPTEval3D. For DreamCraft3D, we use 5000 optimization steps for each of the NeRF, geometry, and texture stages. In addition, we use Stable Diffusion 3 to generate the 2D image prior. We adopt Zero123-XL as the backbone and use Omnidata for depth and normal estimation. The rendering resolution is set to 128x128 in the first stage and 512x512 in the subsequent stages. For Trellis, we apply RewardCS to finetune TRELLIS-text-base. We construct a finetuning dataset by randomly sampling 100K prompts from the TRELLIS-500K training set. For each prompt, a 3D asset is generated and then differentially rendered to DMTet or multi-view 2D images. We use either Reward3D or RewardCS as the reward signal to finetune the Trellis backbone. We will include the detailed optimal funetuning setup in the revised manuscript.
>
> |Model| CP (↑) | VR (↑) | GA (↑) | T-A (↑) | 3DP (↑) | G-T (↑) | Proportion (↓) | Latency (↓) |
> |-|-|-|-|-|-|-|-|-|
> | MVDream | 0.23 | -3.30 | 2.77 | 2.95 | 3.10 | 3.07 | 0.55 | 8.5 hours |
> | DreamCraft3D | 0.28 | -3.20 | 2.84 | 3.11 | 3.26 | 3.16 | 0.41 | 5.4 hours |
> | DreamCraft3D+Reward3D | 0.29 | -2.96 | 2.90 | 3.43 | 3.51 | 3.45 | 0.35 | 5.5 hours |
> | DreamCraft3D+RewardCS | 0.30 | -2.08 | 3.13 | 3.61 | 4.11 | 4.07 | 0.24 | 5.4 hours |
> | TRELLIS | 0.29 | -1.53 | 2.86 | 3.03 | 3.30 | 3.51 | 0.43 | 2.9 mins |
> | TRELLIS+Reward3D | 0.28 | -1.47 | 2.90 | 3.10 | 3.35 | 3.58 | 0.40 | 2.9 mins |
> | TRELLIS+RewardCS | 0.30 | -1.39 | 3.02 | 3.15 | 3.43 | 3.60 | 0.35 | 2.9 mins |
>
> These results indicate that advanced mesh-based and end-to-end methods with RewardCS not only outperform implicit SDS-based methods, such as MVDream, in quality but also achieve significantly lower inference latency.
>
> Furthermore, our framework is modular and fully compatible with advances in mesh extraction techniques. As faster learned meshing or hybrid mesh-reconstruction methods become available, DreamCS naturally inherits these speed gains without compromising guidance quality.
>
> Thus, while DreamCS has extra inference cost only in the implicit SDS based methods, it is efficient in the mesh-based and end-to-end pipelines that show the current state of the art. We believe this reflects DreamCS’s practical benefit: it can achieve consistent quality improvements across advanced text-to-3D backbones without incurring heavy computational overhead during inference.
>
> [1] Sun, Jingxiang, et al. Dreamcraft3d: Hierarchical 3d generation with bootstrapped diffusion prior. arXiv preprint arXiv:2310.16818 (2023).
>
> [2] Xiang, Jianfeng, et al. Structured 3d latents for scalable and versatile 3d generation. Proceedings of the Computer Vision and Pattern Recognition Conference. 2025.
>
> ---
> We hope our paper revision and above discussion address your concerns. Please do not hesitate to contact us for any further questions or clarifications.

---

### Official Review · Reviewer_c45f · 2025-11-02

**Soundness:** 2
**Presentation:** 2
**Contribution:** 1
**Rating:** 4
**Confidence:** 4

**Summary:**

This work addresses the high cost of paired multi-view preference datasets and geometric artifacts in text-to-3D generation. The authors create 3D-MeshPref, an unpaired dataset of 30,000+ text-mesh-score triplets labeled via Llama-Mesh. They propose RewardCS, a reward model that encodes 3D geometry and text to predict preference scores, trained with Cauchy-Schwarz divergence to separate preferred and dispreferred meshes in latent space without requiring paired supervision.

**Strengths:**

- The Cauchy-Schwarz divergence training approach for unpaired preference data offers a potentially generalizable framework applicable beyond 3D generation tasks.
- 3D-MeshPref provides a human-verified dataset of diverse unpaired 3D meshes, which may be a good community resource that helps reduces dependence on expensive paired annotations.
- The differentiable meshization pipeline enables end-to-end optimization with geometry-aware supervision, demonstrating technical soundness in integrating 3D reward signals into existing text-to-3D frameworks.

**Weaknesses:**

- The CS divergence receives extensive theoretical treatment (Appendix B) but lacks empirical validation. No ablations demonstrate performance degradation without this loss, no clustering baselines justify its necessity, and Table 4's λ variations don't compare against removing the term entirely. The mathematical formalism appears to add complexity without proven practical benefit. I ask the authors to further elaborate upon this point.
- Table 1 exposes a critical flaw: RewardCS underperforms Reward3D on ImageReward (DreamFusion: -0.21 vs 1.71) and only improves when combined with it (1.89). This pattern repeats across backbones, contradicting the claim that 3D supervision addresses 2D methods' limitations. I believe the method functions as a supplement rather than a standalone solution: please elaborate on this point.
- MeshMAE embeddings are trained purely for geometric reconstruction, encoding shape but not color, material, or style. Cross-attention thus lacks features to evaluate stylistic prompts.
- The experimental results seem to only show marginal improvements over baseline method despite increased computation overhead.

**Questions:**

Please see the Weaknesses section.

---

> ### Author Response · Authors · 2025-11-20
> **Response to Reviewer c45f (1/4)**
>
> Dear Reviewer c45f,
>
> Thank you for the insightful and valuable comments! In the following, we provide our point-by-point response and hope our response helps address your concerns. We also look forward to the subsequent discussion which may further help solve the current issues.
>
> > **W1: The CS divergence receives extensive theoretical treatment (Appendix B) but lacks empirical validation. No ablations demonstrate performance degradation without this loss, no clustering baselines justify its necessity, and Table 4's $\lambda$ variations don't compare against removing the term entirely. The mathematical formalism appears to add complexity without proven practical benefit.**
>
> Thank you for raising this point regarding the empirical validation of the CS divergence. We clarify both the motivation and the supporting experimental evidence below.
>
> As discussed in L202-L207, we explain the motivation to use CS divergence as the latent-space separation loss. Compared to the Kullback-Leibler divergence, the CS divergence offers a tighter generalization bound and is more robust than Jensen Shannon divergence, which lacks a closed-form for Gaussians [1,2,3]. This makes CS divergence well-suited for unpaired 3D reward learning, as maximizing it helps the model capture semantic and geometric cues that differentiate preferred from dispreferred assets without explicit paired supervision for each prompt.
>
> In Appendix D.2, we provide the ablation study on latent-space separation loss $\mathcal{L}\_{div}$ and empirically validate the loss $\mathcal{L}\_{div}$. Table 4 demonstrates the clustering performance under different $\lambda$ values. We conduct ablations with four different $\lambda$ values: 0, 0.5, 1, and 10. The  baseline using $\lambda=0$ corresponds to the RewardCS model without using the latent-space separation loss $\mathcal{L}\_{div}$ (i.e. there is only the MSE loss of the score). For each $\lambda$ setting, we train a separate RewardCS model and evaluate the quality of the learned embeddings using three standard clustering metrics across 10 test batches from the 3D MeshPref dataset. Across Tables 4 and 5, RewardCS trained with $\lambda=0.5$ and $1$ consistently outperforms the $\lambda=0$ variant across all metrics, with $\lambda=1$ yielding the strongest performance overall. This directly demonstrates that removing $\mathcal{L}\_{div}$ leads to degraded clustering performance, and therefore that the CS-based separation term provides practical benefit in addition to its theoretical motivation.
>
> [1] Yin, Wenzhe, et al. Domain adaptation with cauchy-schwarz divergence. arXiv preprint arXiv:2405.19978 (2024).
>
> [2] Fuglede, Bent, and Flemming Topsoe. Jensen-Shannon divergence and Hilbert space embedding. International symposium on Information theory, 2004. ISIT 2004. Proceedings. IEEE, 2004.
>
> [3] Nielsen, Frank. On the Jensen–Shannon symmetrization of distances relying on abstract means. Entropy 21.5 (2019): 485.
>
> ---
>
> > **W2: Table 1 exposes a critical flaw: RewardCS underperforms Reward3D on ImageReward (DreamFusion: -0.21 vs 1.71) and only improves when combined with it (1.89). This pattern repeats across backbones, contradicting the claim that 3D supervision addresses 2D methods' limitations. I believe the method functions as a supplement rather than a standalone solution.**
>
> Thank you for the insightful comment. We agree that on the ImageReward (IR) metric, RewardCS does not surpass Reward3D. As noted in L406 of submission, Reward3D is explicitly finetuned to maximize the IR score, which leads methods guided by Reward3D to overfit to ImageReward’s 2D aesthetic preferences. As a result, Reward3D-guided models achieve higher IR scores even when their underlying 3D quality is not improved, an effect reflected in the multi-view visualizations in Appendix G.1.
>
> To avoid relying on an IR-biased evaluation, our revision add  the CLIP (CP) metric as a 2D metric to complement IR. The image-text similarity metric  is computed by averaging the similarities between each view and the given prompt. As shown in Table 1, applying RewardCS to vanilla text-to-3D generation consistently outperforms both vanilla baselines and 2D reward variants on CP and VisionReward (VR), a more advanced 2D automatic metric, demonstrating that RewardCS improves text–asset alignment in the 2D multi-view space.
>
> Beyond 2D metrics, RewardCS also enhances 3D plausibility and geometry-texture consistency. Table 2(a) and 2(b) show that both MiniCPM-o and human evaluators prefer RewardCS-guided results over those generated with 2D preference models, particularly on geometry-related criteria, while maintaining competitive text–asset alignment.
>
> Finally, regarding the Janus problem, Table 3 shows that both MVDream and DreamFusion with RewardCS yields a lower rate of Janus artifacts (0.30 and 0.41) compared to both the vanilla baseline (0.52 and 0.61) and 2D-guided variants, indicating stronger multi-view consistency of RewardCS.

---

> > ### Author Response · Authors · 2025-11-20
> > **Response to Reviewer c45f (2/4)**
> >
> > > **W3: MeshMAE embeddings are trained purely for geometric reconstruction, encoding shape but not color, material, or style. Cross-attention thus lacks features to evaluate stylistic prompts.**
> >
> > Thank you for the thoughtful comment. DreamCS is primarily motivated by the need to mitigate severe geometric artifacts in current text-to-3D pipelines — particularly the Janus problem, where objects appear plausible from one view but distorted from another. As shown in Figure 1 and discussed in Sections 1 and 3 of our paper, such defects significantly compromise the usability of generated assets. To address this, we propose RewardCS, the first 3D reward model trained on unpaired data, which delivers fine-grained geometry-aware feedback. While our design emphasizes geometry, we design DreamCS as a modular framework that is compatible with both geometry- and appearance-level reward signals. In particular, it supports integration with existing 2D-based models and advanced SDS backbones that are more sensitive to texture, color, and style. As shown in Tables 1, 2, and 3, the DreamCS framework alone achieves competitive or superior performance across all 2D and 3D metrics than vanilla baselines and 2D reward variants, and combining it with 2D reward models further boosts alignment with stylistic prompts.

---

> > > ### Author Response · Authors · 2025-11-20
> > > **Response to Reviewer c45f (3/4)**
> > >
> > > > **W4: The experimental results seem to only show marginal improvements over baseline method despite increased computation overhead.**
> > >
> > > Thank you for the insightful comment. We would like to emphasize that the improvements brought by RewardCS are not marginal; in fact, they exceed the gains achieved by 2D reward variants over their respective vanilla baselines. In addition, we conduct additional experiments comparing RewardCS directly to MVControl [2], one of 3D controllable generation methods. Moreover, we demonstrate that RewardCS can flexibly serve as a reward model for finetuning end-to-end 3D generative models, such as Trellis [5].
> > >
> > > **Performance Superiority of RewardCS**
> > >
> > > As shown in Table 1, for DreamFusion, DreamFusion + RewardCS improves CP by +8.7\% and VR by +29.2\% over DreamFusion + DreamDPO (the 2nd-best variant), whereas DreamFusion + DreamDPO improves over the vanilla DreamFusion by only +4.5\% (CP) and +7.2\% (VR). This demonstrates a substantial improvement from RewardCS.
> > >
> > > For 3D-specific metrics, Table 2(a) shows that MVDream + RewardCS achieves +20.9\% (3DP) and +22.7\% (G-T) over MVDream + Reward3D, while the Reward3D variant improves over vanilla MVDream by only +7.4\% (3DP) and +4.5\% (G-T). Thus, RewardCS yields stronger improvements in geometry and 3D plausibility.
> > >
> > > Regarding multi-view consistency, Table 3 shows that RewardCS reduces the proportion of Janus-affected assets by 31.8\% (MVDream) and 22.6\% (DreamFusion) compared to the Reward3D variant. In contrast, Reward3D reduces Janus artifacts by only 15.4\% and 13.1\% over their respective baselines. This further underscores the effectiveness of RewardCS in improving the 3D quality.
> > >
> > > In addition, we highlight that RewardCS is model-agnostic and requires no external input conditions, unlike existing controllable generation methods [4,5,6] dealing with the Janus problem. Moreover, it can flexibly serve as a reward model for finetuning end-to-end 3D generative models, such as Trellis [7]. This flexibility broadens its applicability in the 3D generation domain.
> > >
> > > **Comparison with 3D Controllable Generation Methods**
> > >
> > > 3D controllable generation methods such as MVControl [4], MT3D [5], and DreamControl [6] can provide guidance for optimization-based 3D generation, but they rely on strong 3D shape prior or external conditioning 2D inputs, making them difficult to apply in real-world text-to-3D settings. Among them, MVControl is the most compatible with the MVDream framework, so we adopt it for a fair comparison.
> > >
> > > Under the MVDream backbone, we adopt the pretrained MVControl checkpoint. Given 30 evaluation prompts from GPTEval3D, we first generate a 2D image prior using stable diffusion 3 and then use the MVcontrol method to generate the 3D assets. We report performance using the same metrics in Tables 1, 2, and 3 of our main paper, where T-A, 3DP, and G-T are derived from MLLM evaluation.
> > >
> > > |Model| CP (↑) | VR (↑) | GA (↑) | T-A (↑) | 3DP (↑) | G-T (↑) | Proportion (↓) |
> > > |-|-|-|-|-|-|-|-|
> > > | MVDream | 0.23 | -3.30 | 2.77 | 2.95 | 3.10 | 3.07 | 0.55 |
> > > | MVControl | 0.27 | -3.09 | 2.86 | 3.42 | 3.47 | 3.51 | 0.40 |
> > > | MVDream+Reward3D | 0.27 | -3.13 | 2.85 | 3.38 | 3.34 | 3.20 | 0.46 |
> > > | MVDream+RewardCS | 0.29 | -2.15 | 2.96 | 3.58 | 4.04 | 3.93 | 0.31 |
> > >
> > > RewardCS outperforms both MVControl and Reward3D across all metrics, including CP, VR, GA, and especially in 3D plausibility and geometry–texture consistency. Moreover, RewardCS achieves the lowest Janus proportion (0.31), confirming that it effectively mitigates the Janus issue—without requiring any auxiliary conditioning input.
> > >
> > > **RewardCS for Advanced End-to-End 3D Generation Method**
> > >
> > > To further demonstrate its generality, we apply RewardCS to finetune the advanced end-to-end text-to-3D generation model TRELLIS-text-base. We construct a finetuning dataset by randomly selecting 100K prompts from the TRELLIS-500K training set. For each prompt, a 3D asset is generated and differentially rendered to DMTet or 2D multi-view images. We then use either Reward3D or RewardCS as a reward guidance signal to finetune the TRELLIS backbone.
> > >
> > > Given 60 evaluation prompts from GPTEval3D, we report performance using the same metrics in Tables 1, 2, and 3 of our main paper, where T-A, 3DP, and G-T are derived from MLLM evaluation.
> > >
> > > |Model| CP (↑) | VR (↑) | GA (↑) | T-A (↑) | 3DP (↑) | G-T (↑) | Proportion (↓) |
> > > |-|-|-|-|-|-|-|-|
> > > | TRELLIS | 0.29 | -1.53 | 2.86 | 3.13 | 3.30 | 3.51 | 0.43 |
> > > | TRELLIS+Reward3D | 0.28 | -1.47 | 2.90 | 3.20 | 3.35 | 3.58 | 0.39 |
> > > | TRELLIS+RewardCS | 0.30 | -1.39 | 3.02 | 3.25 | 3.43 | 3.60 | 0.35 |
> > >
> > > These results demonstrate that RewardCS consistently improves text–asset alignment, 3D plausibility, geometry–texture consistency, and reduces Janus artifacts, outperforming both the base TRELLIS model and the 2D reward variant. This highlights RewardCS’s flexibility and effectiveness as a model-agnostic reward signal for advanced end-to-end 3D generation frameworks.

---

> ### Author Response · Authors · 2025-11-20
> **Response to Reviewer c45f (4/4)**
>
> ### **Follow-up on response to W4**
>
> We would like to clarify that the reported computational overhead occurs only in the implicit NeRF-based SDS setting, where DreamCS performs DMTet meshization at every iteration. **These implicit NeRF-based SDS experiments can demonstrate that DreamCS is a model-agnostic guidance module compatible with general text-to-3D generative backbone.**
>
> For more advanced text-to-3D pipelines, where methods optimize an explicit mesh rather than an implicit NeRF field, RewardCS becomes lightweight. When integrated into mesh-based SDS systems such as Magic3D and DreamCraft3D, RewardCS operates directly on explicit geometry and adds <1\% inference-time cost, making the inference computational cost negligible. In addition, RewardCS can serve as a 3D reward guidance to finetune current end-to-end text-to-3D models such as Trellis, which generates a 3D asset in ~3 minutes.
>
> |Model| CP (↑) | VR (↑) | GA (↑) | T-A (↑) | 3DP (↑) | G-T (↑) | Proportion (↓) | Latency (↓) |
> |-|-|-|-|-|-|-|-|-|
> | MVDream | 0.23 | -3.30 | 2.77 | 2.95 | 3.10 | 3.07 | 0.55 | 8.5 hours |
> | DreamCraft3D | 0.28 | -3.20 | 2.84 | 3.11 | 3.26 | 3.16 | 0.41 | 5.4 hours |
> | DreamCraft3D+Reward3D | 0.29 | -2.96 | 2.90 | 3.43 | 3.51 | 3.45 | 0.35 | 5.5 hours |
> | DreamCraft3D+RewardCS | 0.30 | -2.08 | 3.13 | 3.61 | 4.11 | 4.07 | 0.24 | 5.4 hours |
> | TRELLIS | 0.29 | -1.53 | 2.86 | 3.03 | 3.30 | 3.51 | 0.43 | 2.9 mins |
> | TRELLIS+Reward3D | 0.28 | -1.47 | 2.90 | 3.10 | 3.35 | 3.58 | 0.40 | 2.9 mins |
> | TRELLIS+RewardCS | 0.30 | -1.39 | 3.02 | 3.15 | 3.43 | 3.60 | 0.35 | 2.9 mins |
>
> These results indicate that advanced mesh-based and end-to-end methods with RewardCS not only outperform implicit SDS-based methods, such as MVDream, in quality but also achieve significantly lower inference latency.
>
> Furthermore, our framework is modular and fully compatible with advances in mesh extraction techniques. As faster learned meshing or hybrid mesh-reconstruction methods become available, DreamCS naturally inherits these speed gains without compromising guidance quality.
>
> Thus, while DreamCS has extra inference cost only in the implicit SDS based methods, it is efficient in the mesh-based and end-to-end pipelines that show the current state of the art. We believe this reflects DreamCS’s practical benefit: it can achieve consistent quality improvements across advanced text-to-3D backbones without incurring heavy computational overhead during inference.
>
> **Key Advantages of RewardCS**
>
> We highlight that RewardCS is model-agnostic and requires no external input conditions, unlike existing controllable generation methods [2,3,4] dealing with the Janus problem. Moreover, it can flexibly serve as a reward model for finetuning end-to-end 3D generative models, such as Trellis [5]. This flexibility broadens its applicability in the 3D generation domain.
>
> 1. **Model-agnostic**
>
> RewardCS is model- and architecture-agnostic, integrating seamlessly with diverse backbones such as MVDream, DreamFusion, and TRELLIS, and consistently enhancing both geometric fidelity and semantic alignment.
>
> 2. **No external conditioning priors**
>
> Methods such as MVControl, MT3D, and DreamControl require 2D or 3D priors. RewardCS requires only text and mesh input without external constraints, making it more practical and broadly applicable.
>
> 3. **Effectively reduces Janus artifacts**
>
> RewardCS yields the lowest Janus proportion in all experiments — outperforming even 3D controllable methods — while maintaining text-asset alignment.
>
> These comparisons show that RewardCS not only surpasses existing 3D controllable generation methods, but is also more flexible and model-agnostic to deploy in general text-to-3D pipelines.
>
>
> [4] Li, Zhiqi, et al. Mvcontrol: Adding conditional control to multi-view diffusion for controllable text-to-3d generation. arXiv preprint arXiv:2311.14494 (2023).
>
> [5] Nath, Utkarsh, et al. Deep Geometric Moments Promote Shape Consistency in Text-to-3D Generation. 2025 IEEE/CVF Winter Conference on Applications of Computer Vision (WACV). IEEE, 2025.
>
> [6] Huang, Tianyu, et al. Dreamcontrol: Control-based text-to-3d generation with 3d self-prior. Proceedings of the IEEE/CVF conference on computer vision and pattern recognition. 2024.
>
> [7] Xiang, Jianfeng, et al. Structured 3d latents for scalable and versatile 3d generation. Proceedings of the Computer Vision and Pattern Recognition Conference. 2025.

---

### Author Response · Authors · 2025-11-20
**Summary of Revisions**

We thank all the reviewers for their valuable feedback and insightful suggestions. Based on the reviews, we have made the following revisions to our paper:

1. We include a widely-used 2D text-asset alignment metric CLIP (CP) to complement the model-biased 2D metric IR, and provide an explanation regarding to the metric GA. (Table 1 and Appendix F.1) [`Reviewer c45f, cTX5, 697y`]

2. We validate the Llama-Mesh scoring capability and provide a detailed description of the human refinement process. (Appendix C.2, F) [`Reviewer cTX5, pQXb`]

3. We include additional multi-stage and advanced text-to-3D SDS-based methods in our comparisons and discussion [1-2]. (Section1, Appendix G.2) [`Reviewer c45f, cTX5, pQXb, 697y`]

4. We provide extended results on 3D controllable generation methods, such as MVControl [3]. (Appendix G.2) [`Reviewer c45f, cTX5, pQXb, 697y`]

5. We add an analysis of how RewardCS can adapt to end-to-end 3D generation frameworks, such as Trellis [4]. (Appendix G.2) [`Reviewer c45f, cTX5, pQXb, 697y`]

6. We provide a detailed analysis of wall-clock time and GPU memory usage for implicit NeRF-based SDS-based and advanced text-to-3D generative pipelines. (Appendix G.4) [`Reviewer cTX5, 697y`]

7.  We validate our proposed Adaptive Mesh Fusion module. (Appendix E.2) [`Reviewer 697y`]

8. We have improved the quality of visual illustrations (e.g., Figures 5). [`Reviewer 697y`]

We have also addressed each reviewer’s comments with more detailed, in-depth responses. Once again we appreciate all the suggestions made by reviewers to improve our work. It is our pleasure to hear your feedback, and we look forward to answering your follow-up questions.

[1] Wang, Zhengyi, et al. Prolificdreamer: High-fidelity and diverse text-to-3d generation with variational score distillation. Advances in neural information processing systems 36 (2023): 8406-8441.

[2] Kwak, Min-Seop, et al. "Geometry-Aware Score Distillation via 3D Consistent Noising and Gradient Consistency Modeling." arXiv preprint arXiv:2406.16695 (2024).

[3] Li, Zhiqi, et al. Mvcontrol: Adding conditional control to multi-view diffusion for controllable text-to-3d generation. arXiv preprint arXiv:2311.14494 (2023).

[4] Xiang, Jianfeng, et al. Structured 3d latents for scalable and versatile 3d generation. Proceedings of the Computer Vision and Pattern Recognition Conference. 2025.

---

### Meta-Review · Area_Chair_w87U · 2026-01-07

**Summary:**

The initial scores were mixed. Most of the early concerns were about how efficient the framework is. In particular, cTX5, 697y, and pQXb pointed out the extra runtime from differentiable meshing, about a 27 percent increase. The rebuttal directly addresses this efficiency issue. The authors also show RewardCS working with an end-to-end model called Trellis and mesh-based pipelines like DreamCraft3D, and the inference overhead there is under 1 percent. The 3D-MeshPref dataset is a strong addition, and the new experiments suggest the method can fit into different 3D pipelines, so AC lean toward acceptance. Authors should incorporate the efficiency analysis and Trellis experiments

**Reviewer Concerns:**

Addressed:

Efficiency: Results on Trellis and DreamCraft3D show under 1 percent inference overhead.

Metric validity: They clarified GA is trained on Cap3D and used CLIP and VisionReward to explain the ImageReward gap as baseline overfitting.

Dataset bias: They reported 93.6 percent agreement between Llama-Mesh and human judgments.

Outstanding:

NeRF-SDS overhead: In the NeRF-SDS setting, differentiable meshing still adds about 27 percent runtime, and this remains a real trade-off.

**Reviewer Scores:**

c45f: Likely 4. The extra CLIP metrics and the ImageReward explanation help.

cTX5: Likely 4. Added baselines help, but the NeRF-SDS overhead remains.

pQXb: Likely 6. Trellis integration supports the generality claim.

697y: Likely 6. Adaptive Mesh Fusion concerns look addressed but still with nonnegligible computational overhead.

---

### Decision · Program_Chairs · 2026-01-26

Accept (Poster)